# Reconciling the surface temperature–surface mass balance relationship in models and ice cores in Antarctica over the last two centuries

Marie G.P. Cavitte[1], Quentin Dalaiden[1], Hugues Goosse[1], Jan T.M. Lenaerts[2], and Elizabeth R. Thomas[3]

[1]Georges Lemaître Centre for Earth and Climate Research (TECLIM), Earth and Life Institute (ELI), Université catholique de Louvain (UCL), Louvain-la-Neuve Belgium
[2]Department of Atmospheric and Oceanic Sciences, University of Colorado Boulder, Boulder CO, USA
[3]British Antarctic Survey, Madingley Road, Cambridge, CB3 0ET, UK

**Correspondence:** Marie G.P. Cavitte (marie.cavitte@uclouvain.be)

**Abstract.** Ice cores are an important record of the past surface mass balance (SMB) of ice sheets, with SMB mitigating the ice sheets' sea level impact over the recent decades. For the Antarctic Ice Sheet (AIS), SMB is dominated by large-scale atmospheric circulation, which collects warm moist air from further north and releases it in the form of snow as widespread accumulation or focused atmospheric rivers on the continent. This suggests that the snow deposited at the surface of the AIS should record strongly coupled SMB and surface air temperature (SAT) variations. Ice cores use $\delta^{18}O$ as a proxy for SAT as they do not record SAT directly. Here, using isotope-enabled global climate models and the RACMO2.3 regional climate model, we calculate positive SMB-SAT and SMB-$\delta^{18}O$ annual correlations over ~90% of the AIS. The high spatial resolution of the RACMO2.3 model allows us to highlight a number of areas where SMB and SAT are not correlated, and show that wind-driven processes acting locally, such as foehn and katabatic effects, can overwhelm the large-scale atmospheric contribution in SMB and SAT responsible for the positive SMB-SAT annual correlations. We focus in particular on Dronning Maud Land, East Antarctica, where the ice promontories clearly show these wind-induced effects. However, using the PAGES2k ice core compilations of SMB and $\delta^{18}O$ of Thomas et al. (2017) and Stenni et al. (2017), we obtain a weak annual correlation, on the order of 0.1, between SMB and $\delta^{18}O$ over the past ~150 years. We obtain an equivalently weak annual correlation between ice core SMB and the SAT reconstruction of Nicolas and Bromwich (2014) over the past ~50 years, although the ice core sites are not spatially co-located with the areas displaying a low SMB-SAT annual correlation in the models. To resolve the discrepancy between the measured and modeled signals, we show that averaging the ice core records in close spatial proximity increases their SMB-SAT annual correlation. This increase shows that the weak measured annual correlation partly results from random noise present in the ice core records, but the change is not large enough to match the annual correlation calculated in the models. Our results indicate thus a positive correlation between SAT and SMB in models and ice core reconstructions but with a weaker value in observations that may be due to missing processes in models or some systematic biases in ice core data that are not removed by a simple average.

## 1 Introduction

In the context of current climate change and sea level rise in the last century, it is important to better constrain the future
contributions from the Antarctic Ice Sheet (AIS) to sea level rise. The AIS is projected to be the largest source of sea level
rise over centennial to millenial timescales, with a potential contribution of 58.3 m if the entire ice sheet were to melt (Stocker
et al., 2013; Pörtner et al., in press). To better predict the timing and rate of the AIS contributions to sea level rise, we need
to better constrain its mass balance. The grounded AIS mass balance is the difference between surface mass balance (SMB),
i.e. the addition of mass at the surface of the ice sheet, and ice discharge through basal hydrology and calving (Lenaerts et al.,
2019). The AIS mass balance is currently negative due to increased ice discharge at the grounding lines (Shepherd et al., 2018),
particularly enhanced in the Amundsen Sea Embayment (Mouginot et al., 2014; Rignot et al., 2019), outpacing a positive trend
in SMB (Medley and Thomas, 2019). SMB could play a bigger role in mitigating future sea level rise, but it is still not well
understood. For the AIS, the SMB signal is generally dominated by snow accumulation (Agosta et al., 2019; Van Wessem et al.,
2018). The main SMB sinks for the AIS are sublimation, wind ablation and melt water runoff, although the latter is negligible
for the AIS due to the very low surface temperatures.

Large-scale atmospheric circulation (100s of km) strongly controls SMB in Antarctica. This large-scale atmospheric cir-
culation embeds synoptic-scale cyclones that collect heat and moisture from further north, including the Southern Ocean,
which they can release onto the AIS (Gorodetskaya et al., 2014; Sodemann and Stohl, 2009; Wang et al., 2019). SMB shows
a large coast-to-interior gradient with very low SMB values in the continental interior (Scarchilli et al., 2011; Fujita et al.,
2011; Favier et al., 2013; Frezzotti et al., 2007), although atmospheric rivers can bring mid-latitude moisture very deep into
the interior (Gorodetskaya et al., 2014). Large-scale atmospheric modes dominate snow accumulation variability in Antarctica
(Lenaerts et al., 2019). Notably, the Southern Annular Mode has been shown to have a dominant role in driving snow accu-
mulation variability, particularly over the Antarctic Peninsula (AP, Thomas et al., 2008, 2015, 2017) and to largely explain the
observed SMB trend patterns across the entire AIS during the 20th century (Medley and Thomas, 2019). In addition, based
on the Clausius-Clapeyron relationship, the increasing mid and upper troposphere temperature due the warmer climate implies
that the vapor pressure of the air is higher, and therefore snowfall is increased (Frieler et al., 2015; Fudge et al., 2016). If this
predicted increase in SMB is linked to increasing temperatures in the 21st century, it will be interesting to see if SMB and SAT
are linked in the past centuries too, since an increase in SAT and tropospheric air temperature are strongly correlated (Kirtman
et al., 2013). In this case, SMB records over time will be a helpful tool to constrain past temperature reconstructions (Dalaiden
et al., 2020).

Ice cores provide an important local and regional record of snow accumulation (Thomas et al., 2017), while the $\delta^{18}$O
measured in the ice cores is often used as a proxy of snowfall-weighted SAT (and by extension of SAT, Stenni et al., 2000).
The SMB and SAT records should be correlated, regardless of whether SMB is controlled by the Clausius-Clapeyron law
or by synoptic-scale atmospheric circulation. Several studies have already shown SMB and $\delta^{18}$O to co-vary in ice cores and

SMB and SAT to co-vary in models for specific AIS regions. Medley et al. (2018) calculate a sensitivity between SMB and SAT around 19 mm w.e. $°C^{-1}$ over Queen Maud Land using global atmospheric models, Frezzotti et al. (2004) measure a linear snow accumulation increase of ~15 mm w.e. per $°C$ increase in firn temperature in ice cores over Adélie Land and Frieler et al. (2015) observe and model a linear increase ranging from ~4.5 to 7% $K^{-1}$ over the AIS. However, Medley and Thomas (2019) have recently shown that, despite a continent-wide temperature warming, SMB trends vary strongly spatially at the continent scale. Both the ice core records and the models used in the study point to a complex SMB-SAT relationship. Dalaiden et al. (2020) have described the co-variance of SMB-SAT on millennial and centennial timescales for seven distinct Antarctic regions. However, they already display a large discrepancy in the strength of the co-variance at this continental scale between the ice core records and the model predictions.

Added complexity in snow accumulation records originates from smaller-scale (several kilometers or less) wind-based re-distribution of snow that is superimposed on the patterns originating from the spatial variability in precipitation. Wind speeds that reach a threshold speed (~5 m $s^{-1}$) can pluck snow from the surface and redeposit it up to several kilometers away where winds decelerate or remove it through blowing snow sublimation (Frezzotti et al., 2004; King et al., 2004; Lenaerts et al., 2019; Agosta et al., 2019). Due to the temperature inversion that commonly occurs near the surface of the ice sheet, the surface air is negatively buoyant and flows down-slope through the influence of gravity (van den Broeke et al., 2002). Whillans (1975) shows a strong relationship between slope, wind strength and mass drift transport. With increased distance and steepness of slope, these dry cold air masses originating from the interior accelerate significantly as they flow down towards the coast, the so-called katabatic winds (Bromwich, 1989; Bromwich and Liu, 1996), reaching speeds of 20 m $s^{-1}$ over Adélie Land (van den Broeke et al., 2002), and up to 90 m $s^{-1}$ measured at Cape Denison (Ball, 1957). Katabatics can cause widespread snow redistribution over kilometers and thus affect the local SMB.

Winds can also impact surface accumulation over even shorter spatial scales (10s of meters to kilometers). As surface winds flow over surface topography, they typically erode the windward side and deposit on the leeward side, thus reworking the SMB record spatially (Black and Budd, 1964; Budd, 1971; Dattler et al., 2019). If winds are very persistent in direction and speed, wind redistribution of the snow pack can create fields of dunes (Frezzotti et al., 2002b; Arcone et al., 2012a; Das et al., 2013) or blue ice areas (Spaulding et al., 2012). Winds also have a strong effect on SAT at the local scale and therefore affect the co-variance of SMB-SAT. For instance, Lenaerts et al. (2017) show that persistent katabatic winds on the Dronning Maud Land (DML) coast can strongly warm the air temperature through enhanced air column mixing at the grounding line.

An additional influence on SAT at the local scale, but also at larger spatial scales, is when large-scale (synoptic) winds interact with surface topography that acts as an obstacle to air flow. A thermodynamic effect will occur, known at the foehn effect. This foehn effect can occur across mountain ranges (e.g. the AP mountain chain) or isolated bumps (e.g. ice promontories or ice rises). Cooling and condensation during uplift of the air mass will remove moisture which implies latent heating of the air and results in warmer air descending on the lee side of the surface topography (Elvidge et al., 2015; Elvidge and Renfrew, 2016). This has been well documented for the AP (Datta et al., 2018, 2019).

Despite these processes that render our interpretation of ice core records difficult, we need the ice core observations to go back in time to validate model reconstructions before the period when we have direct measurements.

Our work follows from that of Dalaiden et al. (2020), in which they show that SMB and SAT are strongly interannually correlated at the continental scale over millennial (1000–2005 AD) and historical (1850–2005 AD) timescales. Here, we examine whether the relationship remains strong at the model grid point scale (which we refer to as the regional scale) and the ice core scale (referred to hereafter as the local scale). For this, we use a suite of Global Climate Models (GCM), a Regional Climate Model (RCM) and published ice core compilations available for the AIS. Combining both in-situ measurements and model simulations allows us to characterise the SMB-SAT relationship over these different spatial scales. In summary, we want to understand the following aspects of the link between SAT and SMB:

1. Do the processes that link SAT and SMB at the continental scale also play a dominant role at the regional scale?

2. How does the SMB-SAT link in models at the regional scale compare to that in ice cores at the local scale?

3. Can our improved understanding of processes at the regional scales explain why the SMB-SAT link measured in ice cores is different to the link as measured in models?

In addition to an improvement of our understanding of the dynamics of the system and the processes controlling SMB, answering these questions will help constrain our confidence in using SMB to reconstruct SAT over the entire AIS. With relatively few in-situ surface air temperature (SAT) observations, establishing additional proxies for SAT would be extremely beneficial for Antarctic climate reconstructions. In this manuscript, we constrain the empirical link between SAT and SMB at the regional and local scales.

Because of the resolution of the ice cores, the focus of our analysis will be interannual variations, both for models and observations. We start with a brief description of the data sets and models used in this study. Then we compare the positive SMB-SAT relationship, as well the SMB-$\delta^{18}$O relationship, obtained in the models at regional scale to the continental-scale results of Dalaiden et al. (2020). We hypothesize physical mechanisms that could explain the discrete areas of the AIS where the SMB-SAT relationship is weak, compared to the continent-wide positive SMB-SAT relationship. We then examine the SMB-SAT relationship in the ice core data, and attempt to reconcile the differences between the results at this local scale and those from the models at the regional scale.

## 2 Methods and Data

### 2.1 Isotopic global climate models

To study longer-term regional climate variability, we use modeled SMB and SAT from four isotope-enabled GCMs (iGCMs hereafter) taken from the Coupled Model Intercomparison Project Phase 5 (CMIP5). These were chosen specifically because (1) they have historical simulations for the required variables (SAT, precipitation and sublimation/evaporation for SMB), (2) they simulate water isotope variations ($\delta^{18}$O) explicitly, which can be compared directly to water isotopes from the ice cores and (3) for consistency with those used in the Dalaiden et al. (2020) study (with the additional fourth made available recently,

iCESM1, Brady et al. (2019); Stevenson et al. (2019)). The iGCM characteristics are detailed in Dalaiden et al. (2020) and Brady et al. (2019) but we briefly describe their relevant characteristics to this study:

  – ECHAM5-MPI/OM is a fully coupled GCM that includes atmospheric and oceanic components, and covers the period 800–2000 AD. It is forced by natural and anthropogenic forcing (Sjolte et al., 2018) and has a spatial resolution of $3.75°$ $\times 3.75°$ (for the atmospheric component).

– ECHAM5-wiso is a GCM that only includes an atmospheric component and covers the period 1871–2011 AD driven by sea surface temperatures and sea ice (Rayner et al., 2003) at a spatial resolution of ~1° (Steiger et al., 2017).

  – iHadCM3 is the isotope-enabled version of the fully coupled version of GCM HadCM3 (Turner et al., 2016; Holloway et al., 2016) and covers the period 1851–2003 AD at a spatial resolution of $3.75° \times 2.5°$ (for the atmospheric component).

  – iCESM1 is an isotope-enabled version of CESM1 (Brady et al., 2019). It has active atmosphere, land, ocean, river
transport and sea ice model components, with a spatial resolution of ~2° for the atmospheric model. The simulations cover 850–2005 AD. iCESM1 is an addition to the iGCM model list of the Dalaiden et al. (2020) study.

Dalaiden et al. (2020) provide an evaluation of first three iGCMs used, and we provide an evaluation of the SMB, SAT and the atmospheric circulation (we use sea-level pressure) for all four iGCMs in Supplement S2. SMB and SAT are evaluated against RACMO27, while sea-level pressure is evaluated against ERA-Interim as the ERA-Interim grid extends further north.
The evaluation is done over the 1979–2000 AD time period, the longest period of overlap for all the models examined. We show that all four models produce realistic SMB, SAT and sea-level pressure outputs, with biases likely linked to differences in their physics as well as their spatial resolutions (see supplement S2).

For the iCESM1 and iHadCM3 GCM ensembles that include three and seven simulations, respectively (each has slightly different initial conditions), we first calculate the correlation of the two variables (SMB-SAT or SMB-$\delta^{18}$O) for every grid
point for each ensemble member (3 for iCESM1 and 7 for iHadCM3). Then we obtain the mean of the correlation values per grid point over each ensemble of simulations for both iCESM1 or iHadCM3. These ensemble means are then compared to the correlations calculated for ECHAM5-MPI/OM and ECHAM5-wiso, as well as with observations.

As we conduct a high number of significance tests on gridded maps, we increase the number of false rejections of the null hypothesis (Wilks, 2016). We apply the correction method to gridded maps, as described by Benjamini and Hochberg
(1995), which controls the "False Discovery Rate" (FDR), i.e the fraction of erroneously rejected null hypotheses. For N local hypothesis tests (i.e. number of model grid points), we obtain N p-values ($p_i$) for i = 1 to N. Local null hypotheses are rejected if their p-values are no larger than a threshold level ($p_{FDR}$) calculated from the distribution of the sorted p-values $p_i$, as described by Wilks (2016):

$$p_{(FDR)} = max_{i=1,...N}[p_{(i)} : p(i) <= (i/N)\alpha_{FDR}] \tag{1}$$

In our case, we choose a control $\alpha_{FDR}$ value of 0.05, which allows us to calculate a $p_{FDR}$ value for each correlation map. The threshold p-value $p_{FDR}$ determines if a correlation is significant or not. In the case of the iHadCM3 and iCESM1 models,

we do not average the p-values obtained from the different members as for the correlation values, but rather apply a threshold of at least 50% of p-values $< p_{FDR}$ to define whether a grid point correlation value is significant (which we then simply label as "significant" on the figures).

To compare the iGCM continent-wide correlations to the RCM-derived correlations, we interpolate the iGCM correlation results onto the RCM grid and average over all four iGCMs (Fig. 1 and 2, panels e). P-values are not interpolated but rather a threshold of at least 50% of p-values $< p_{FDR}$ is used to define whether a grid point correlation value is significant, the same way it was done for the iHadCM3 and iCESM1 ensemble members.

In this study, we focus on two time intervals for the iGCMs: a longer (130 years) timescale covering 1871–2000 AD to compare to the measured ice core SMB and $\delta^{18}$O data, and a shorter (40 years) timescale covering 1961–2000 AD for comparison to the RCM simulations and measured SAT. Since the ice core $\delta^{18}$O data in the regional compilation of Stenni et al. (2017) are 5-yearly in resolution, and we wish to build upon the findings of the Dalaiden et al (2020) study which uses 5-yearly data, we also look at 5-yearly correlations between SMB and $\delta^{18}$O for the iGCMs. To ensure that the time period selected does not impact the correlations, we calculate the SMB-SAT and SMB-$\delta^{18}$O correlation over the 1871–2000 AD (annually for SMB-SAT and 5-yearly for SMB-$\delta^{18}$O) and the 1961–2000 AD intervals (annually) for the iGCMs (Fig. S1 and S2). We show that correlation strengths are similar over the two time intervals, both in spatial distribution and in continent-wide average (see Section 3). As expected, with a shorter time interval, p-values tend to increase (e.g. compare Fig. 1 over 1871–2000 AD and Fig. S1 over the 1961–2000 AD interval). Using 5-year averages (Fig. 2) instead of yearly values of SMB and $\delta^{18}$O (Fig. S3) does not impact the correlation strength significantly (iGCM average 0.47±0.13 vs 0.45±0.12), but it increases the p-values (80% vs 95% significant grid cells), due to the time series' effective shortening. We choose to use the 1961–2000 AD time interval instead of starting in 1979 AD as for the RCM simulation, so that all model simulations have a similar length (40 years and 38 years for the iGCMs and the RCM, respectively) to ensure that the correlation results are as meaningful as possible.

## 2.2 RACMO2.3p2 regional climate model

For regional climate variability at a higher spatial resolution, we use the RCM Regional Atmospheric Climate MOdel version 2.3p2 (RACMO2.3 hereafter) which provides SMB, SAT and other relevant atmospheric variables for the entire AIS. RACMO2.3 combines the atmospheric dynamics of the High Resolution Limited Area Model (HIRLAM5, Unden et al., 2002) and the physics package of the European Centre for Medium-range Weather Forecasts (ECMWF, 2009). It is forced at its boundaries by the ERA-Interim reanalysis from ECMWF over 1979–2016 AD. This version of RACMO is coupled to a multi-layer snow model which includes a snow albedo and a drifting snow scheme (Lenaerts et al., 2010), thus making it particularly well-suited to study polar regions. RACMO2.3 has been demonstrated to have the best fit to recent AIS SMB observations compared to other atmospheric and reanalysis models (Wang et al., 2016). A detailed description of RACMO2.3 can be found in Van Wessem et al. (2018). In this study, we use the 27 km horizontal gridding simulations over the entire AIS (Van Wessem et al., 2018), as well as the 5.5 km horizontal gridding simulations focused on the DML region (~25°W–45°E, Lenaerts et al., 2017). The latter configuration at higher resolution allows studying in more details a region with complex surface topography

and SMB records (Lenaerts et al., 2017, 2014). Both RACMO2.3 simulations cover 1979–2016 AD. We will refer to these as RACMO27 and RACMO5 hereafter.

## 2.3 In-situ measurements

The annual correlations obtained from the model simulations are compared to in-situ measurements of SMB, SAT and $\delta^{18}$O from ice cores and weather stations.

Ice cores provide a measurement of SMB based on the measured distance between distinct seasonal depth markers within an ice core on historical timescales. The annual layer thickness is corrected for densification, compaction and ice flow to produce a water equivalent snow accumulation. We use the Thomas et al. (2017) compilation, the most comprehensive SMB compilation to date for the AIS. Thomas et al. (2017) use 79 ice cores over the AIS to obtain a SMB record with an annual resolution that covers the last millenium up to 2010 AD.

Ice cores record variations of the $\delta^{18}$O of the ice with depth. Here we use the Stenni et al. (2017) compilation (a total of 112 ice cores) over the AIS to obtain a $\delta^{18}$O record that covers the past millennium up to 2014 AD with yearly resolution for 0–2014 AD and 5-yearly resolution for the regional averages. The ice core $\delta^{18}$O record can be used as a proxy for snowfall-weighted SAT, but the link between the two variables can be complex and sometimes is quite weak (Dalaiden et al., 2020; Klein et al., 2019; Goursaud et al., 2017, 2019). Dalaiden et al. (2020) have shown that the annual correlation between SMB and $\delta^{18}$O was

weaker (but still significant) than the annual correlation of SMB with SAT, for seven distinct regions at the continental scale.

Because ice cores do not provide a direct measurement of SAT, we rely on the Nicolas and Bromwich (2014) SAT reconstruction. Nicolas and Bromwich (2014) use SAT records from 15 weather stations around the AIS (mostly coastal, Fig. S4c) to produce estimates of monthly gridded mean temperature anomalies for the whole continent over 1958–2012 AD. The reconstruction is based on a kriging technique, originally developed for the reconstruction of Antarctic snowfall (Monaghan et al.,

2006), to interpolate the data between the stations. The interpolated SAT field allows us to correlate SAT and SMB directly at the ice core locations so as to compare to the RACMO2.3 results. To calculate the significance of our correlations, we choose a p-value of 0.05.

In the Thomas et al. (2017) and Stenni et al. (2017) ice core data compilations, most of the ice core records start after 1800 AD, and we choose to correlate the SMB and temperature signals starting in 1871 AD in order to compare to the models.

Furthermore, the spatial distribution of the ice cores over the AIS is not homogeneous. If we divide Antarctica into low elevation areas (i.e. <2200 m) and high elevation areas (i.e. >2200 m), the majority of the ice cores are located in the low elevation areas, and very few in high elevation areas (except for the large number of ice cores all located in DML, see supplementary Fig. S4). This certainly introduces a spatial bias in our ice core-based annual correlation towards coastal signals and processes.

# 3 Results and Discussion

## 3.1 Do the processes that link SMB and SAT at the continental scale also play a dominant role at the regional scale?

### 3.1.1 Strength of the link between SMB and SAT at the regional scale

We calculate the annual correlation between SMB and SAT at the regional scale over 1871–2000 AD, the time interval shared by all iGCMs. We obtain a positive annual relationship between SMB and SAT at the regional scale with a continent-wide average value of 0.57 and a spatial standard deviation of $\pm 0.10$ (hereafter referred to as $\pm 0.10$) over the four iGCMs (the individual iGCMs continent averages range from 0.54-0.60) with a p-value $< p_{FDR}$ for more than 96% of each iGCM's surface area (Fig. 1), and $> 99\%$ on average. We note some spatial differences between models, in particular we focus on five distinct regions of the ice sheet: (1) the AP ("AP" on Fig. 1) shows weak correlations for iCESM1 and ECHAM5-wiso (0.22-0.24, 57-76% significant grid cells) and positive correlations for ECHAM/MPI-OM and iHadCM3 (0.51-0.60, 100% significant), (2) central West Antarctica ("CWA") which shows the same contrast between iCESM1 and ECHAM5-wiso versus ECHAM/MPI-OM and iHadCM3 (0.31-0.48, 78-91% significant versus 0.63 and 100% significant), (3) the East Antarctic Plateau ("EAP") which shows positive and 100% significant correlations (0.58-0.70); the Amery Embayment ("AE") and Adelie Land ("AL") which show positive correlations (0.49-0.57 and 0.49-0.69, 100% significant). Areas of weaker SMB-SAT correlations are only observed in iCESM1 and ECHAM5-wiso, which could arise from the models' finer spatial resolution compared to iHadCM3 and ECHAM5-MPI/OM. The SMB-SAT correlation remains positive overall for all models. We obtain the same result if we take non-overlapping 5-year averages of the SMB and SAT variables to calculate their 5-yearly correlation (supplementary Fig. S5).

We repeat the annual correlation of SAT and SMB using the RACMO27 simulations over 1979–2016 AD (Fig. 1, panel f). At this spatial scale, the annual correlation is also positive over the large majority of the ice sheet with a similar range of correlation values and a continent-wide average value of 0.54±0.22, which is within the range of the iGCM average (0.57±0.10). For the same five distinct regions, we can see that the EAP and AL are also positive and 100% significant. For AP and CWA that had a lower correlation for iCESM1 and ECHAM5-wiso, we see a stronger weakening down to 0.27 with a lower percentage of significant grid cells (48%). The AE area is most distinct with a correlation average of 0.22 and $\sim$30% significant grid cells. The much higher spatial resolution of RACMO27 with respect to the iGCMs is likely behind these differences, including the higher percentage of areas with a weaker correlation value, if we compare to what we observed for the iGCMs.

Moving down to RACMO5 simulations, the highest spatial model resolution available to us, we show that the annual correlation in the DML region is also positive, with a regional average value of 0.48±0.18 (comparable to an average value of 0.52±0.21 for RACMO27 for the same region, Fig. 3). The annual correlation of SMB and SAT is therefore similar over the 1871–2000 AD and the 1979–2016 AD time intervals, and from a spatial resolution of $> 1°$ down to 5.5 km. We observe a positive correlation for the continent as a whole, strongly positive values for the EAIS plateau and WAIS (with always the presence an area of non significant correlation in central West Antarctica), but weaker values with proximity to the coast, along the eastern side of the AP and over the Amery Embayment.

Dalaiden et al. (2020) have shown that the relationship between SMB and SAT is positive on the continental scale for each of their seven Antarctic regions, whether they used the GCMs or RACMO2.3 simulations. The simple concept that Antarctic precipitation originates mainly from lower latitudes, coming from relatively warm and wet air masses, explains the co-variance of SMB and SAT at the continental scale as shown by Dalaiden et al. (2020). This simple concept therefore also applies at the regional scale for interannual variations here. Indeed, for the temporal and spatial scales investigated, the annual correlation between SMB and SAT is positive for a large majority of the model grid points, despite a few regional differences due to the models' differing spatial resolutions.

The correlation between SMB and SAT remains true for summer- or winter-only months (average SMB-SAT correlation of 0.43±0.21 or 0.56±0.21, respectively using RACMO27, supplementary Fig. S6) or on monthly timescales after removing the seasonal cycle (0.54±0.17 on average, supplementary Fig. S7).

### 3.1.2 Wind effects on SMB and SAT signals

There are a few areas, spatially consistent between the RACMO27, RACMO5 simulations and the iGCMs, where the SMB-SAT annual correlation is not as strong. In those areas, shaded in grey on Fig. 1, panel f, for RACMO27, and outlined on subsequent figures, the annual correlation is either insignificant (p-value > $p_{FDR}$) or negative. A weak SMB-SAT annual correlation suggests that physical processes must affect SMB, SAT or both enough locally that they break the relationship between SMB and SAT seen at large scale in other regions. Winds are known to affect SMB and SAT locally, through wind-based redistribution of SMB, turbulent and adiabatic warming from katabatic winds and foehn effects on leeward slopes. The impact of wind on SMB and SAT due to this redistribution can modify the link between the two variables but this can only be resolved by the RACMO2.3 simulations, because it is the only model in this study that includes drifting snow (although according to Agosta et al. (2019), drifting snow is strongly underestimated). To evaluate whether the lack of correlation between SAT and SMB is due to any of these wind effects, we correlate modelled surface winds to our two variables of interest.

Because wind direction with respect to surface slope and wind strength influence SMB and SAT locally (Frezzotti et al., 2004; King et al., 2004; Black and Budd, 1964; Grima et al., 2014), we take both into account in the Mean (surface) Slope in the (mean) Wind Direction (MSWD). MSWD is defined as the dot product between the mean surface slope and the mean wind direction (Scambos et al., 2012; Das et al., 2013; Dattler et al., 2019):

$$MSWD = \overrightarrow{ws}_{10m} \cdot \overrightarrow{slope} \tag{2}$$

with $\overrightarrow{ws}_{10m}$ the wind speed 10 m above the surface, and define $\overrightarrow{slope}$ as pointing down-slope. Since we define a positive MSWD as a wind pointing down-slope, we expect winds coming from the coast up into the interior to have a negative MSWD, and winds flowing from the interior to the coast to have a positive MSWD (Fig. 4, and supplementary Fig. S8).

The surface slope is calculated using the surface topography used in the RACMO2.3 simulations. As the ratio of vertical distance over horizontal distance, slope has units of $m \cdot m^{-1}$, i.e. unitless. We then remove areas of the AIS with a negligible slope (<0.001) as in these areas, MSWD will be close to null and will introduce noise when correlating it with SMB or SAT. We then correlate SAT and SMB to this MSWD.

Large-scale air masses, originating over the Southern Ocean and further north, bring warm moist air towards the interior as they flow up-slope. At the annual time scale, these up-slope winds (negative MSWD) will therefore bring a higher temperature and SMB (positive anomaly), and thus induce a strong and negative annual MSWD-SAT and MSWD-SMB correlation (Fig. 4, scenario 1a). Similarly, winds flowing down-slope (positive MSWD) will bring drier and colder air (negative anomaly) from the interior to the coast, and will also induce a strong and negative annual correlation of MSWD with SAT and SMB (Fig. 4, scenario 2a). Any area of the AIS that does not show this negative annual correlation between wind and SAT or SMB implies that the changes cannot be interpreted simply in terms of large-scale circulation at the annual scale or descending cold air from the interior (so in that case, scenarios 1b and 2b). We are aware that cyclonic activity is punctual in time. However, by correlating MSWD to SMB and SAT over 1979–2016 AD, we make the hypothesis that we capture the first order variations of cyclonic activity at interannual timescales, and therefore the variability in the sources of heat and moisture for the continent.

We calculate the annual correlation of MSWD with both SAT and SMB at the 27 km and 5.5 km scales, using the RACMO27 and RACMO5 simulations (Figs. 5 and 6). Examining the results, MSWD is negatively correlated to SAT over most of the continent, with a continent-wide average of -0.31±0.37 at the 27 km scale, and a DML average of -0.4±0.35 at the 5.5 km scale (Fig. 5). MSWD is also mostly negatively correlated to SMB with a continent-wide average of -0.60±0.30 at the 27 km scale, and a DML average of -0.52±0.30 at the 5.5 km scale (Fig. 6).

However, a number of areas show a strongly positive MSWD-SAT annual correlation, simultaneously with a weaker (less negative) or positive MSWD-SMB annual correlation. In addition, these areas are co-located with the areas where the SMB-SAT annual correlation is weak (outlined with a magenta line). Those areas are at roughly the same locations in the 5.5 km and 27 km simulations for the DML area.

We will first examine the wind-SAT interactions. The areas where the SMB-SAT annual correlation is weak have a strong and positive MSWD-SAT annual correlation. We note that these areas tend to be on the leeward side of surface topography. The clearest example of this is found along the AP (Fig. 5, and supplementary Fig. S9 for a zoomed-in view): the eastern (leeward) side of the AP shows positive MSWD-SAT correlation values around 0.4, with a low SMB-SAT annual correlation of 0.38 on average for the iGCMs and 0.15 for RACMO27. Here, Westerlies are the dominant year-round air mass trajectory (Marshall et al., 2006). The mountain chain along the AP creates a long barrier perpendicular to the air masses' dominant trajectory. The warmer air masses coming from the ocean are forced up the windward side (MSWD < 0, bringing warmer SAT from the ocean), which leads to a negative annual MSWD-SAT correlation as this warm air is pushed-up on the windward side. During this process, the air is depleted of moisture and as the now-drier air masses flow back down on the leeward side, the temperature increases (MSWD > 0 and SAT increase further) due to the foehn effect, thus leading to a positive annual MSWD-SAT correlation on this leeward side (Fig. 4, scenario 1b), as observed on Fig. 5. The AP mountains represent very steep topography but this MSWD-SAT positive/negative polarity is also observed in regions with less dramatic topography. Berkner Island ("BI" on Fig. 1, panel f) shows a very distinctive negative MSWD-SAT annual correlation on its eastern side and positive on its western side, which matches the dominant westward wind direction in this area in the lower level of the atmosphere. Zooming in on the DML coast (Fig. 5, panel b), each ice promontory and ice rise shows a distinctive east-west MSWD-SAT

annual correlation, with the positive annual correlation found on the west sides, matching the dominant year-round westward

wind direction in the area (supplementary Fig. S10).

After examining the link with SAT, we will now focus on the link between MSWD and SMB. We see that the areas with a low SMB-SAT annual correlation tend to have a less negative/weak MSWD-SMB annual correlation. These areas correspond to scenario 1b (Fig. 4), i.e. leeward of surface topography or scenario 2b (Fig. 4) when katabatic winds modify the local SMB. For scenario 1, we observe that the MSWD-SMB annual correlation weakens on the leeward side of surface topography, such

as the coastal ice promontories (Fig. 6). Indeed, during average weather conditions in coastal areas, synoptic air masses carry moist air across the surface topography, releasing a significant fraction of their moisture as snow on the windward side, creating a dry accumulation shadow on the leeward side. This has been observed by Lenaerts et al. (2014) and Kausch et al. (2020) for the ice rises of the DML coast. Small-scale accumulation asymmetries across ice rises have also been observed in other regions (King et al., 2004; Morse et al., 1998) and can occur on larger scales, such as across ice divides (Urbini et al., 2008) or the

AP mountain chain (Datta et al., 2018). However, stronger winds can affect this accumulation asymmetry by bringing more moisture and therefore more snow on the windward side, while the leeward side will either remain dry, or begin accumulating more snow if wind speeds allow for redistribution (Frezzotti et al., 2004). In the presence of strong winds, the MSWD-SMB correlation will therefore become weaker on leeward slopes. This is particularly visible across the ice promontories and ice rises on Fig. 6, panel b, where the western sides of these ice promontories and ice rises correspond to the leeward sides based

on the dominant year-round winds for the area (supplementary Fig. S10).

Now considering scenario 2, winds blowing from the continent interior towards the coast are of moderate strength and tend to flow at some angle to the steepest slope when they initiate near the center of the ice sheet, carrying cold dry air from the interior (Parish and Bromwich, 1987). However, when slopes become sufficiently steep, these winds can pick up a lot of speed and blow almost directly down-slope, forming a katabatic wind regime (Parish and Bromwich, 1987). Katabatic winds

transport cold dense air down-slope which can cause widespread erosion of the snow-pack through surface friction locally (MSWD-SMB correlation < 0) and re-deposition of this snow further down-slope (MSWD-SMB correlation > 0). They can also cause a warming of the SAT locally at the grounding lines through increased turbulence where the surface slope breaks from steep ice to flat shelf ice (Lenaerts et al., 2017), or cooling through ablation/sublimation. The Byrd Glacier outlet into the Ross Ice Shelf (red circle on Figs. 5 and 6, panel a) is an example of an area where katabatics are strong year-round and

affect SMB significantly (Ligtenberg et al., 2014; Bromwich, 1989; Parish and Bromwich, 1987). Due to the channeling of the wind down-slope (MSWD > 0) and high sublimation/erosion of the surface (supplementary Fig. S11), most of Byrd Glacier is completely covered with blue ice (SMB = 0 or negative), which agrees with our observed negative MSWD-SMB annual correlation. It also shows a negative MSWD-SAT annual correlation, which matches with the lowered SAT due to the strong ablation/sublimation.

The Amery Embayment ("AE" on Fig. 1, panel f) is an example where the impact of the strong katabatic winds is mitigated. The Amery Embayment is associated with strong sublimation rates (surface and blowing snow sublimation, supplementary Fig. S11), lowering the local SMB and creating areas of exposed blue ice (Markov et al., 2019), but part of this snow gets redeposited down-slope. The Amery Embayment shows a weaker MSWD-SMB annual correlation and a strongly positive

MSWD-SAT annual correlation on the eastern side of the Amery Embayment where katabatics start further inland and flow directly down-slope (Parish and Bromwich, 1987, and supplementary Fig. S10).

In other cases, large-scale air mass input could be sufficiently high that katabatic winds do not affect the deposited surface snow enough to break the large-scale link between SMB and SAT. For example, Adélie Land ("AL" on Fig. 1, panel f) displays high snowfall rates (above 500 mm .w.e yr−1 based on RACMO2.3 results), but is also known for its record-high katabatics (van den Broeke et al., 2002). This region does not display particularly weak MSWD-SMB or MSWD-SAT annual correlations in the RACMO2.3 results.

We therefore expect that the areas regularly under the influence of strong katabatic winds will show a weaker MSWD-SMB annual correlation due to the episodic decrease and increase of their SMB through wind scouring and redeposition, respectively, (Agosta et al., 2019). In those areas, katabatic winds can affect the SMB sufficiently to overwhelm the original synoptic SMB signal (Scambos et al., 2012; Das et al., 2013), but not systematically. At the annual timescale, which is our timescale of interest to compare to the ice core data, the net effect of katabatic winds on SMB is uncertain. Other studies have shown the strong interactions between surface topography and winds. For instance, Agosta et al. (2019) examine the spatial link of time-averaged values of surface curvature and surface winds and they observe that above a certain threshold, winds will affect SMB locally in a pattern that matches that of drifting snow fluxes as modeled by RACMO2.3. Dattler et al. (2019) also show that, at length scales < 25 km, regions of the West Antarctic Ice Sheet show high spatial variability in accumulation simultaneously to high variability in wind speed and direction.

If we now combine the effects of the wind on SAT and SMB, we see that the areas that have a low SMB-SAT annual correlation (outlined by dashed black lines on Fig. 1, panel f, for RACMO27) correspond to a generally positive MSWD-SAT annual correlation (average of 0.03 versus -0.31 for the whole ice sheet, Fig. 5) and weaker MSWD-SMB annual correlation (average of -0.42 versus -0.6 for the whole ice sheet, Fig. 6). The winds, foehn or katabatics, affect local SAT and SMB in those areas but not to the same extent. The synoptic signal consisting of warm and moist air input, or descending cold dry air input, is therefore overwhelmed by wind-induced local SAT or SMB changes, resulting in a weak SMB-SAT annual correlation. Interestingly, we note that the ice rises present on the ice shelves themselves along the DML coast on Fig. 3 do not show a weak SMB-SAT annual correlation. However, they do show both positive MSWD-SMB and MSWD-SAT annual correlations on their leeward sides, both negative MSWD-SMB and MSWD-SAT annual correlations on their windward sides.

### 3.1.3  Spatial-scale dependence of the modelled SAT-SMB annual correlation

To further analyse the spatial scale dependence of the SAT-SMB annual correlation, we smooth the RACMO5 SMB and SAT fields over a step-wise increasing grid spacing, going up to 27.5 km. We calculate the annual correlation between SMB and SAT for each spatial resolution (Fig. 7). We see that spatial smoothing has little effect on the SMB-SAT annual correlation in the model, whose average value remains between 0.48-0.49. Similarly, the areas where the SMB-SAT annual correlation was already weak do not change spatially, and remain co-located with the areas that have a weak SMB-SAT annual correlation at the 27 km scale (shown as dashed black lines). In conclusion, based on the models, the link between SAT and SMB is positive, and valid at all investigated spatial scales, with the exception of areas where wind-induced processes sabotage the link locally.

## 3.2 Strength of the SMB-SAT link in the ice core data

We next examine in-situ measurements of SMB, SAT and $\delta^{18}O$. Because ice cores provide $\delta^{18}O$ and not SAT directly, we first examine the correlation between SMB and $\delta^{18}O$ in the models to compare to the ice core records.

The iGCM simulations give a SMB-$\delta^{18}O$ continent-wide average 5-yearly correlation of $0.47\pm0.13$ over 1871–2000 AD (Fig. 2), compared to the average SMB-SAT annual correlation of $0.57\pm0.10$ for the same time interval (Fig. 1). The strength of the correlation is weaker for SMB-$\delta^{18}O$ than for SMB-SAT for each iGCM, and the percentage of significant grid cells is greatly reduced (although part that reduction is due to the use of 5-yearly averages, instead of annual, supplementary Fig. S3). If we examine the same five regions as described previously for Fig. 1, we see that the 4 iGCM results are consistent, albeit with differences in percentage of significant grid cells. We draw the same conclusions at the regional scale as Dalaiden et al. (2020) at the continental scale: the SMB-SAT annual correlation is stronger than the SMB-$\delta^{18}O$ annual correlation at the regional scale, with a difference on the order of ~0.1.

We then calculate the annual correlation between SMB-SAT and the 5-yearly correlation between SMB-$\delta^{18}O$ for the ice cores, using the Nicolas and Bromwich (2014) SAT reconstruction over 1958–2010 AD. We showed earlier that the timescale used does not affect our conclusions in terms of annual correlation. We can therefore compare SMB-SAT annual correlations over different time intervals between the models and the observations. We observe a weak-to-null annual correlation between SAT and SMB over 1958–2010 AD in the ice cores (Fig. 8) with an average value of $0.09\pm0.18$ over all the ice cores, versus a continent-wide average value of $0.57\pm0.10$ for the iGCMs (Fig. 1, panel e) and $0.54\pm0.22$ for RACMO27 (Fig. 1, panel f). The same is true for the 5-yearly correlation between SMB and $\delta^{18}O$ over 1871–2010 AD with a continent-wide average value of $0.13\pm0.25$ (Fig. 8), versus a continent-wise average value of $0.47\pm0.13$ for the iGCMs (Fig. 2). We note no observable difference in the annual correlation strength between SAT-SMB and SMB-$\delta^{18}O$ in the ice core records, as opposed to in the models.

## 3.3 Can our improved understanding of processes at regional scales help to explain why the SMB-SAT link measured in ice cores is different to the link as estimated in models?

We list six potential reasons why ice cores might show a weaker relationship between SMB and SAT or between SMB and $\delta^{18}O$ than the models. (1) First, we argue that the difference in annual correlation between the models and the observations could arise because of the representation of large scale processes in the models. In other words, the models (iGCMs and RCMs alike) may not represent the reality well enough because they are missing an important dynamic process, i.e. a process that acts to weaken the annual correlation between SMB and SAT. However, we argue that this is unlikely since the iGCM and RACMO2.3 simulations agree despite their different representations of the physics at work and different resolutions. (2) A second hypothesis is that the models do not represent processes well enough at the scale of a few 10s of kilometers. 5.5 km, the RACMO2.3 spatial scale, is a spatial resolution that is still too coarse to resolve small-scale SAT or SMB-modifying processes. In particular, wind redistribution has been shown to be under-estimated in the polar-focused RCMs (Agosta et al., 2019). Turton et al. (2017) have shown that a spatial resolution of 1.5 km is required to simulate foehn flow accurately over

the AP. In addition, we also know from observations that a lot of local scale snow redistribution effects occur to form sastrugi, dunes, etc, which are not resolved in simulations at the 5.5 km scale (Ligtenberg et al., 2014).

(3) On the data side, we know that the Nicolas and Bromwich (2014) SAT data set is not representative of the entire AIS. With only 15 data points for the entire AIS, mostly located around the coast (supplementary Fig. S4c), this data set has a strong coastal signal bias. We have shown in Fig. 1, panel f, that the coastal regions correspond to weaker SMB-SAT annual correlations. However, it does not explain why the SMB-$\delta^{18}$O annual correlation is also weak. (4) Also on the data side, the ice cores might contain a noisy record of SMB and $\delta^{18}$O, therefore reducing the measured correlation between the two. We have to keep in mind that ice cores are point measurements on the ice sheet, with a surface area of ~31 cm$^2$. The ice cores are affected both by measurement errors due to depth, density and age uncertainties (Parrenin et al., 2012), and surface wind processes which act to redistribute the snow at the surface and therefore reduce or increase SMB very locally (e.g. ice crests, sastrugi, ice crusts, etc) or over large areas (e.g. dune fields, Das et al., 2013), thus inducing a representativeness error. This local noise, sampled in the ice core records, might get averaged out at the 5.5 km scale in models. A large number of ice cores might help reduce the noise contained in individual records, if this noise is random. However, estimating this may be hampered by the relatively low number of historical ice core records (53 ice cores were used here to calculate the SMB-SAT and SMB-$\delta^{18}$O annual correlations using the Stenni et al. (2017) and Thomas et al. (2017) compilations). (5) In addition, many of the ice cores retrieved so far from the AIS, and used in the Stenni et al. (2017) and Thomas et al. (2017) compilations, are clustered in the coastal areas, with a higher sampling in West Antarctica than in East Antarctica. This implies that the SMB-SAT annual correlation calculated based on these ice core compilations is likely not representative of the continent-wide, or even regional, SMB-SAT annual correlation. However, it cannot explain the discrepancy with the model-derived annual correlations as most of the ice cores used are located where the models locally predict a positive SMB-SAT annual correlation (i.e. outside of the grey shading on Fig. 8). (6) Finally, we examine the correlation between SMB and SAT on annual timescales, but we know that SMB (snowfall in this case) is brought to the AIS in very episodic ways (atmospheric rivers, etc, Gorodetskaya et al., 2014). Using annual values might not be representative of the conditions during accumulation. Turner et al. (2019) show that more than 70% of the variance of the annual precipitation is explained by extreme events that have a very short duration (one or more consecutive days).

When comparing model (in particular RCM) and ice core results, it is thus not simple to assess the origin of the differences in interannual variations. The advantage of RCMs is that they are self-consistent: the SMB and SAT values simulated are linked by the physics of the model. However, RCMs also include errors that are difficult to trace (difficulty of representing blowing snow or diamond dust). RACMO2.3 for example includes blowing snow processes but Agosta et al. (2019) have shown that its spatial variability is under-represented and that it is underestimated by a factor of 3. Furthermore, RCMs can only go back in time as far as we have reliable reanalyses to drive them at their boundaries (often stopping around 1979 as previous to that, measurement biases increase, Bromwich et al., 2007). Ice cores on the other hand can have different biases dependent on different variables (ablation, diffusion, measurement errors, location, etc). However, they allow us to go back further in time than direct observations.

To see whether the lack of correlation between SMB and SAT in ice cores versus models is due to the presence of local random noise in the ice core records, we aggregate ice cores onto a regularly spaced grid, similarly to how we smoothed the RACMO2.3 data previously (Fig. 7). Since there are few ice cores (53 in total), we use the RACMO27 model grid and average the ice core records every four grid points, i.e. we aggregate ice cores on a 108 x 108 km regular grid. So that high accumulation sites do not dominate the averages calculated, we first normalize the ice core records (SMB and SAT) by subtracting the mean

value over 1960–1990 AD (a period common to all ice cores) and dividing by the standard deviation over that same period (i.e. we use their z-score value). If the lack of correlation between SMB and SAT is due to random noise that operates below the RACMO5 spatial scale, we should see an increase in the SMB-SAT annual correlation using the aggregated climate records. We observe that both the SMB-SAT and the SMB-$\delta^{18}$O annual correlations remain at 0.09 and 0.13 respectively (not shown). If we average records every eight grid points (216 x 216 km grid), the SMB-SAT annual correlation increases up to 0.12 from

0.09 previously, while the SMB-$\delta^{18}$O annual correlation increases to 0.16 from 0.13 previously (Fig. 9). This suggests that a small part of the discrepancy between ice cores and models could be due to noise at the ice core level. The annual correlation remains very low compared to the models, however.

        We repeat the scaling experiment but increase further the distance over which we aggregate the ice core records, from every four RACMO27 grid points step-wise up to every 24 RACMO27 grid points (i.e. 648 x 648 km grid). We only retain grid

points that aggregate five or more ice cores to have sufficient averaging of the ice core records (Fig. 10). For each retained grid point, we calculate (1) the mean of the SMB-SAT annual correlation of each individual ice core aggregated (= "individual means", in light blue on Fig. 10) and (2) the annual correlation of the averaged SMB and SAT records from the aggregated ice cores (= "aggregate values", in dark blue on Fig. 10). Due to the scarcity of ice core measurements, few grid points satisfy our condition of five or more ice cores (e.g. only three grid points contain at least five ice cores at the 648 x 648 km grid

resolution, corresponding to the spacing of 24 grid points on Fig. 10). Two conclusions emerge from this scaling experiment: (1) on average, the aggregate annual correlation is higher than the individual mean annual correlation, (2) this is true at all investigated spatial scales. Averaging the climate signals in the ice core records before calculating the annual correlations (i.e. the aggregate value) increases the ice core SMB-SAT annual correlations by ~0.1-0.2, reaching a correlation up to 0.3.

        For comparison, we calculate the individual mean and the aggregate value of the SMB-SAT annual correlation for the

RACMO27-simulated SMB and SAT fields, only retaining the RACMO27 grid points where ice cores exist (also shown on Fig. 10 in green). This allows us to compare the model-derived and ice core-derived annual correlations locally. As for the ice cores, the aggregate value is always higher than the individual mean for the RACMO27-derived SMB-SAT annual correlation, but the difference is small (up to a 0.1 correlation increase).

        Examining the ice core-based results, we note that increasing the number of ice core records that are initially averaged

results in a visible increase of the SMB-SAT annual correlation, but the trend is weak. The SMB-SAT annual correlation is consistently lower for the ice cores than for the models, aggregate and individual mean values alike. The gap between the models and the ice cores reduces with spatial scaling but remains large (~0.4-0.5 for the same individual mean or aggregate mean). We conclude that there is some random local noise in the ice core records that can be removed by simple averaging over multiple cores. Aggregating the ice core records increases the signal-to-noise ratio of their climate records. However, the

increase in correlation is low. Models may overestimate the annual correlation at scale of tens of kms but it is also likely that we are not able to increase the signal-to-noise ratio sufficiently in the records and reach the obtained model annual correlation.

Another explanation is that we are not able to quantify some systematic processes and effects that occur between the model scale and the ice core scale that should be taken into account. In that case, retrieving multiple cores in different regions is maybe not the most appropriate option (if only for practical reasons). We need to better understand the spatial representativeness of the

ice core records. The sparse distribution of the ice core measurements impacts their representativeness for the whole ice sheet. Frezzotti et al. (2004) had calculated at the time that the total number of accumulation data point measurements (including stakes and non-ice core measurements) reached 1,860 for the entire AIS, representing one data point every 6,500 km$^2$.

We know that the surface of the ice sheet is incredibly rough at the scale of an ice core which will greatly influence the climate records retrieved through wind-topography dynamic feedbacks (Frezzotti et al., 2002a). Furthermore, the location of

an ice core is often chosen based on surface topography and surface features: smooth surface, absence of dunes or surface erosion, top of a dome or of an ice rise, etc. However, dynamic processes, although not acting today, might have been active in the past, as evidenced from ice-penetrating radar data (e.g. Arcone et al., 2012b; Cavitte et al., 2016; Frezzotti et al., 2002a), and therefore can have affected the ice core records examined.

Both the snow accumulation and the isotopic signal recorded in ice cores contain a signature of the post-depositional pro-

cesses occurring at the surface and the intermittency of precipitation (Casado et al., 2019). Extreme precipitation events can explain 70% of the variance of annual precipitation (Turner et al., 2019), for which the greatest contribution occurs in coastal areas. This would suggest that the ice core climatic signal contains only a snapshot of conditions, rather than a continuous record. This is especially important in the stable water isotope records, which are already precipitation biased (only recording temperature during periods of snowfall) and further complicated by isotopic diffusion. Based solely on physical processes,

the local noise in some regions overwhelms the climatic signal at timescales of less than 1000 years (Casado et al., 2019). Using only high accumulation sites (> 0.5 m yr$^{-1}$ at least once over the 1958–2010 AD interval), the resulting SMB-SAT ice core annual correlation average over the continent increases from 0.09±0.18 initially for all cores to 0.28±0.25. Keeping only high accumulation sites seems thus to reduce the impact of post-depositional effects. We are however left with a 13-ice core compilation, with a strong spatial bias towards West Antarctica and the AP.

**4    Conclusions**

We have shown that there is a positive annual correlation between SMB and SAT over the AIS, particularly in the interior of the ice sheet. We have also shown that the annual correlation between SMB and $\delta^{18}$O is also positive although it is weaker than for SMB and SAT. This confirms what has already been shown at the continental scale (Dalaiden et al., 2020). The main source of accumulation over the AIS comes from further north through large-scale atmospheric circulation that carries warm moist

air to the continent (Gorodetskaya et al., 2014; Wang et al., 2019), therefore resulting in the overall positive SMB-SAT annual correlation observed. There are a few areas of the AIS where the SMB-SAT annual correlation does not hold strong, generally found in the coastal areas. These are areas where wind-driven processes act on the SMB or SAT locally, through foehn and

katabatic warming and erosion. If the winds are sufficiently strong, they overwhelm the synoptic-scale inputs that induce the positive SMB-SAT and SMB-$\delta^{18}$O annual correlations.

In models, the SMB-SAT annual correlation does not seem to be strongly scale-dependent. However, the spatial resolution of the models does influence whether we can resolve small-scale topography where wind processes have a dominant influence (e.g. individual ice promontories) and therefore detect a local reduction in the SMB-SAT link. At the ice core scale, we have shown that the annual correlation between SMB and SAT is much weaker (even though the ice cores are located in regions with a high model SMB-SAT annual correlation), corroborating the observations made at the continental scale by Dalaiden

et al. (2020). Averaging ice core records in close spatial proximity improves their SMB-SAT annual correlation, probably due to random noise averaging. Such an increase of the SMB-SAT annual correlation with averaging indicates that the processes detected in the models can also be detected in ice core data, even if the strength of the SMB-SAT link remains lower than in the models. However, in addition to this random noise, ice cores might be affected by a number of local processes that perturb the measured annual correlation between SMB and SAT systematically and cannot be removed through simple averaging.

Choosing only high accumulation ice core sites helps improve the measured annual correlation between SMB and SAT but reduces the number of ice cores left to a handful.

     This implies that we must correct for the local processes present in each ice core record so that their spatial representativeness is closer to that of the models, or models must increase their spatial resolution to better resolve wind effects, in order to improve our confidence in using SMB as a direct proxy for SAT over the entire AIS.

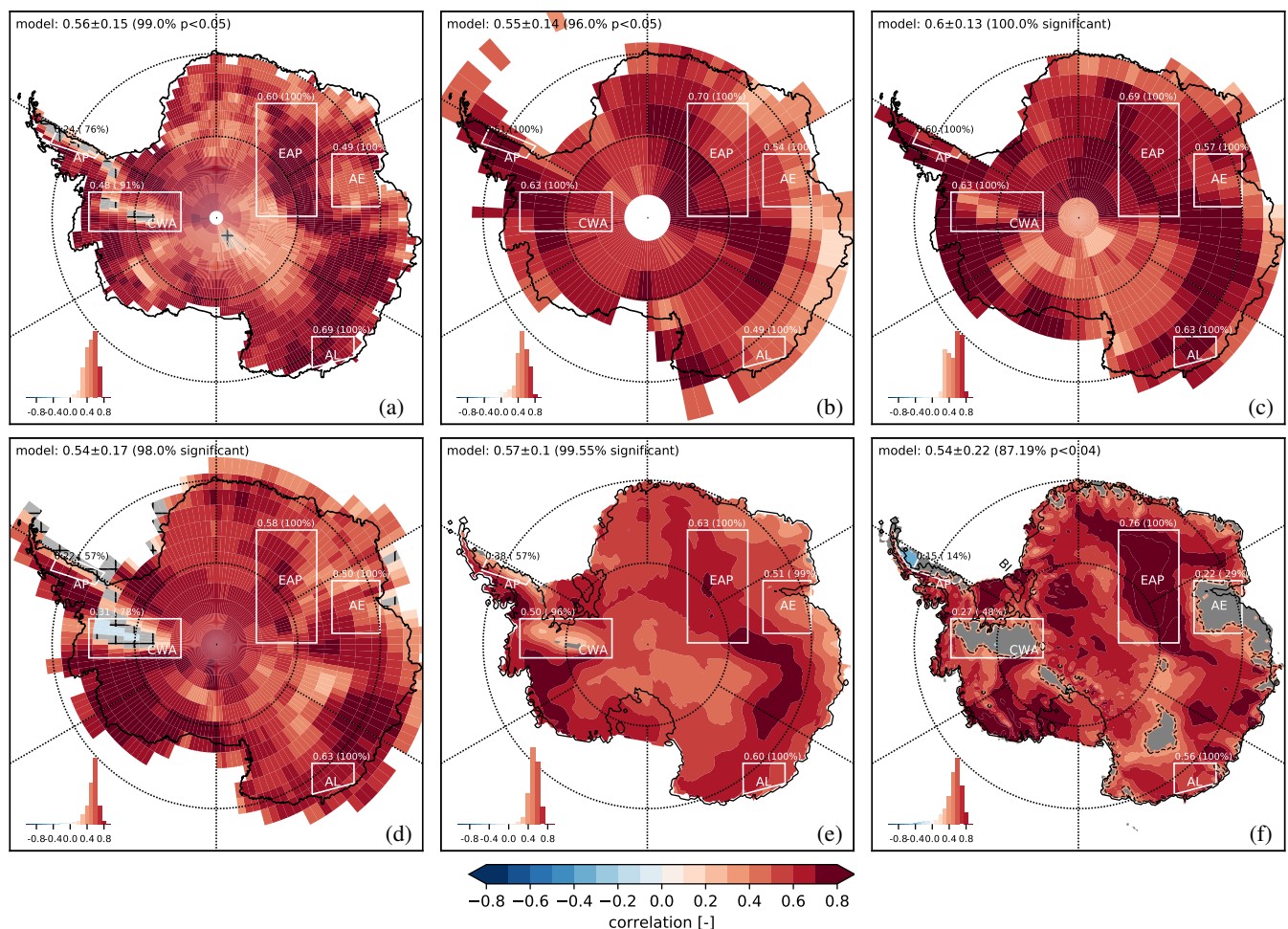

**Figure 1.** Annual correlation between SMB and SAT over 1871–2000 AD for panels a-e: (a) ECHAM5-wiso, (b) ECHAM-MPI/OM, (c) iHadCM3 (7 simulation average), (d) iCESM1 (3 simulation average), (e) averaged over all the isotopic GCMs. Annual correlation between SMB and SAT over 1979–2016 AD for RACMO2.3 at the 27 km spatial resolution for panel (f). Statistically insignificant areas (p > $p_{FDR}$, the threshold p-value calculated) are in grey. The histogram displays the distribution of correlation values. Continent-wide annual correlation mean, standard deviation and percentage of model area with p < $p_{FDR}$ are provided on each panel (for panel e, average over all 4 isotopic GCMs). The five regions mentioned are annotated: 'AE', Amery Embayment - 'AL', Adélie Land - 'EAP', East Antarctic Plateau - 'AP', Antarctic Peninsula, 'CWA', central West Antarctica, as well as 'BI', Berkner Island.

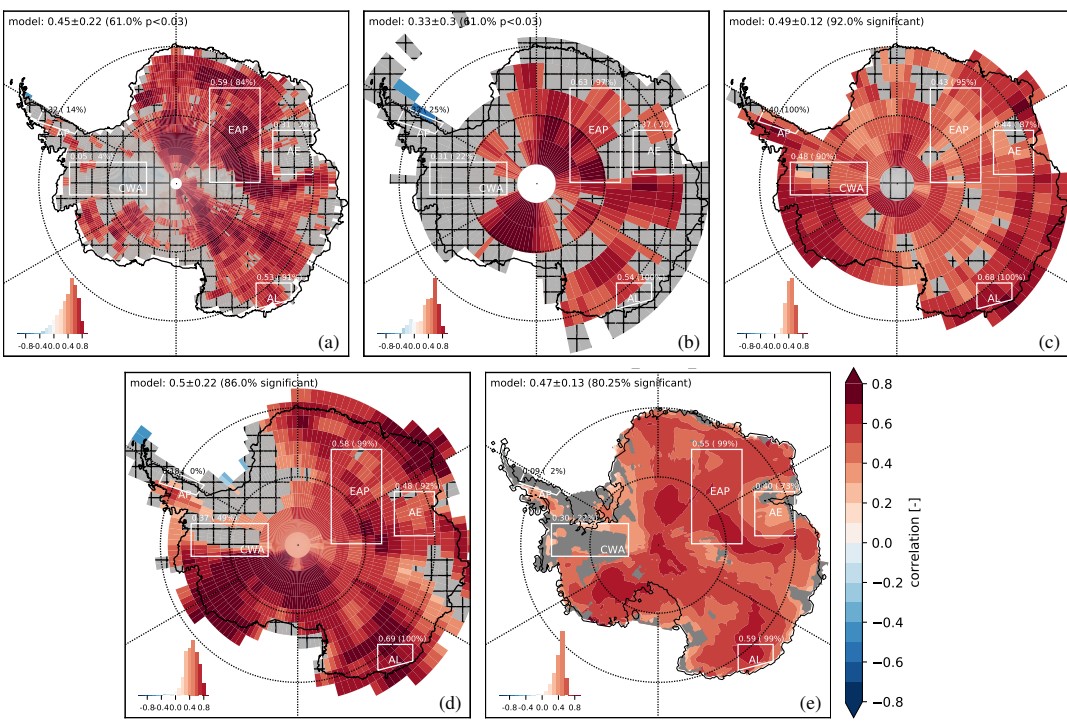

**Figure 2.** 5-yearly correlation between SMB and $\delta^{18}$O over 1871–2000 AD for panels a-e: (a) ECHAM5-wiso, (b) ECHAM-MPI/OM, (c) iHadCM3 (7 simulation average), (d) iCESM1 (3 simulation average), (e) averaged over all the isotopic GCMs. Statistically insignificant areas (p > $p_{FDR}$, the threshold p-value calculated) are in grey. The histogram displays the distribution of correlation values. Continent-wide annual correlation mean, standard deviation and percentage of model area with p < $p_{FDR}$ are provided on each panel (for panel e, average over all 4 isotopic GCMs). The five regions mentioned are annotated as in Fig. 1.

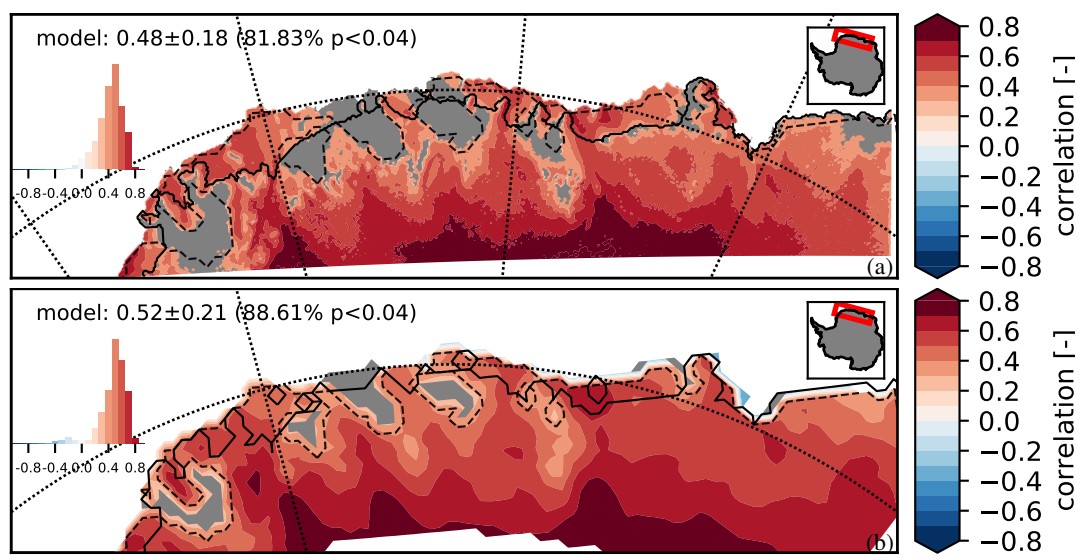

**Figure 3.** Annual correlation between SMB and SAT for RACMO2.3 over 1979–2016 AD at the (a) 5.5 km and (b) 27 km spatial resolution for Dronning Maud Land. Statistically insignificant areas (p > $p_{FDR}$, the threshold p-value calculated) are in grey. The dashed black lines on both panels correspond to the areas with p > $p_{FDR}$ at the 27 km resolution. The histogram displays the distribution of correlation values. Region-wide annual correlation mean, standard deviation and percentage of model area with p < $p_{FDR}$ are provided on each panel.

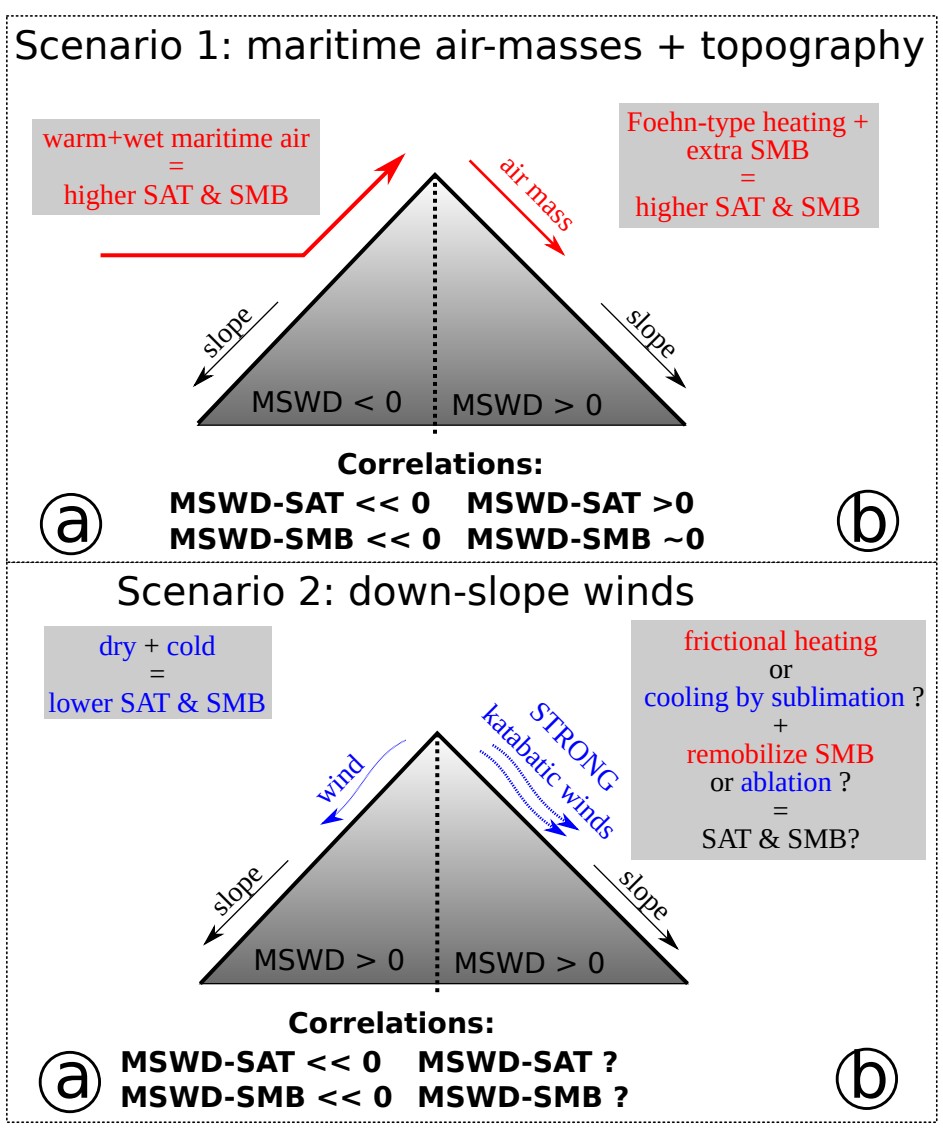

**Figure 4.** Sketch of slope, wind, MSWD and the resulting correlations expected for the two scenarios described: (1) ocean-sourced warm and moist air interacting with topography and (2) down-slope and strong katabatic winds interacting with the surface.

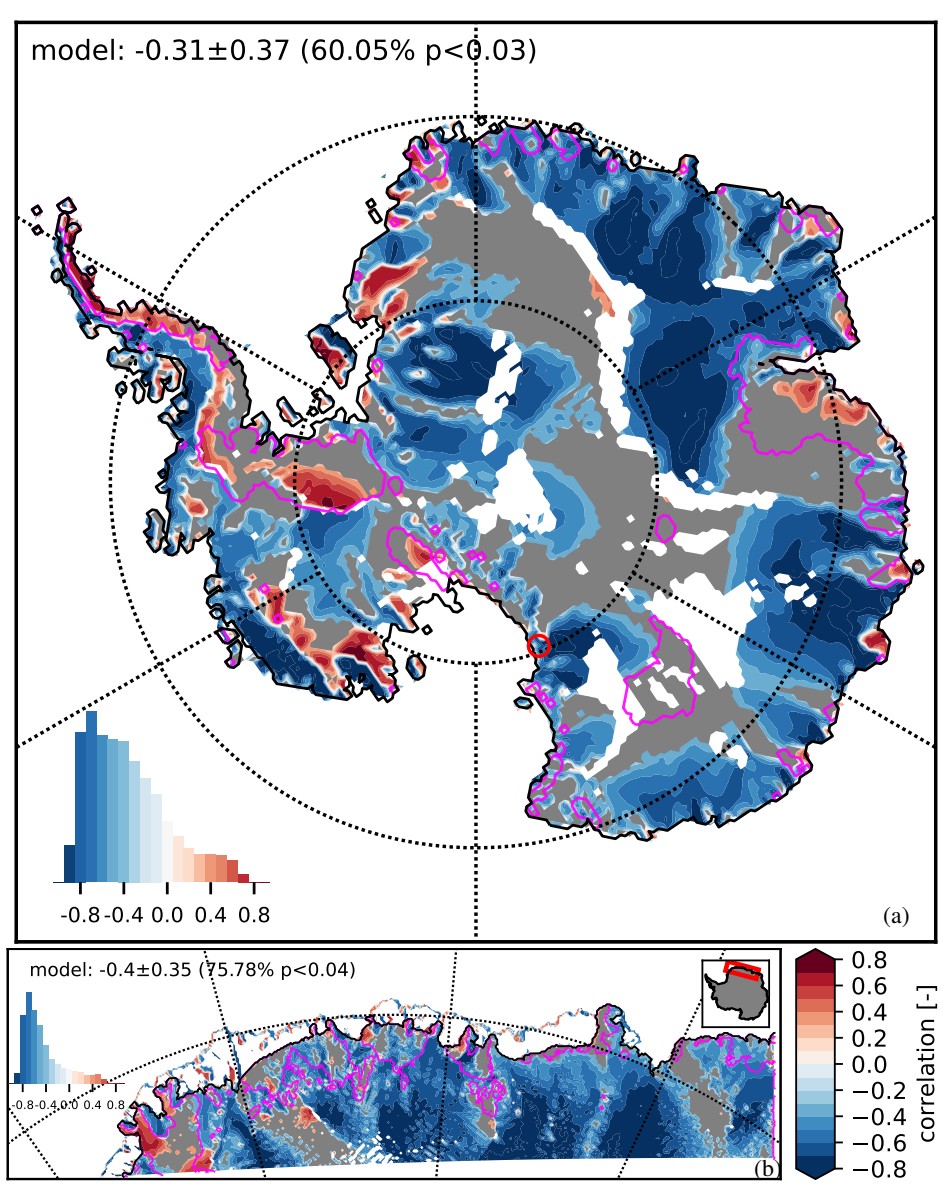

**Figure 5.** Annual correlation between MSWD and SAT using (a) RACMO27 and (b) RACMO5 simulations over 1979–2016 AD. Statistically insignificant areas (p > $p_{FDR}$, the threshold p-value calculated) are in grey. Areas with a slope smaller than 0.1% are removed and appear in white. Magenta lines outline the areas that have a weak SMB-SAT annual correlation in Fig. 1, panel f. The histogram displays the distribution of correlation values. Continent- or region-wide annual correlation mean, standard deviation and percentage of model area with p < $p_{FDR}$ are provided on each panel. A red circle locates the Byrd Glacier outlet discussed in the manuscript.

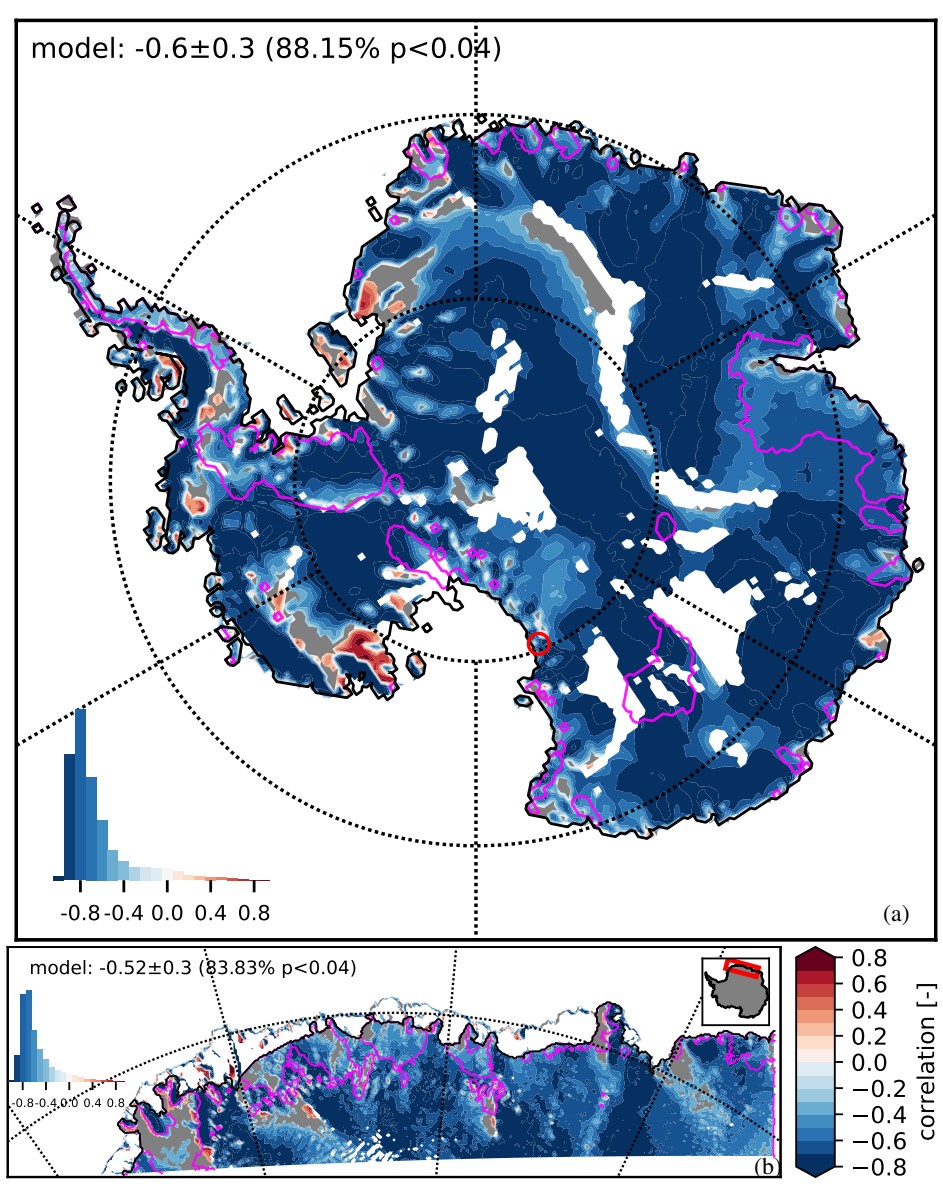

**Figure 6.** Annual correlation between MSWD and SMB using (a) RACMO27 and (b) RACMO5 simulations over 1979–2016 AD. Statistically insignificant areas (p > $p_{FDR}$, the threshold p-value calculated) are in grey. Areas with a slope smaller than 0.1% are removed and appear in white. Magenta lines outline the areas that have a weak SMB-SAT annual correlation in Fig. 1, panel f. The histogram displays the distribution of correlation values. Continent- or region-wide annual correlation mean, standard deviation and percentage of model area with p < $p_{FDR}$ are provided on each panel. A red circle locates the Byrd Glacier outlet discussed in the manuscript.

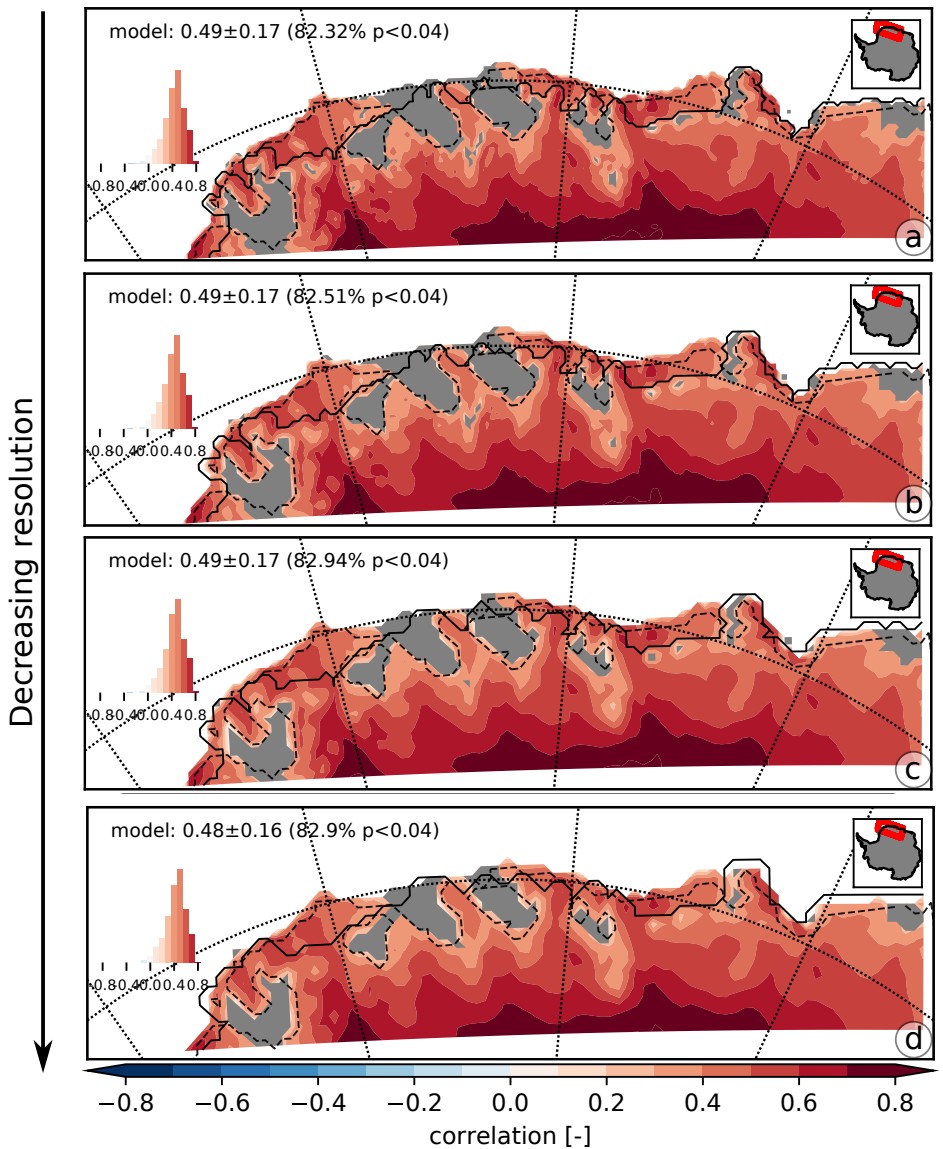

**Figure 7.** Annual correlation between SMB and SAT for RACMO5 simulations over increasing grid scalings from top to bottom: (a) 11 km, (b) 16.5 km, (c) 22 km and (d) 27.5 km over 1979–2016 AD. Statistically insignificant areas (p > $p_{FDR}$, the threshold p-value calculated) for each resolution are hashed in grey. A dashed black line outlines the areas with a low SAT-SMB annual correlation from Fig. 1, panel f, for comparison. The histogram displays the distribution of correlation values for each panel. Region-wide annual correlation mean, standard deviation and percentage of model area with p < $p_{FDR}$ are provided on each panel.

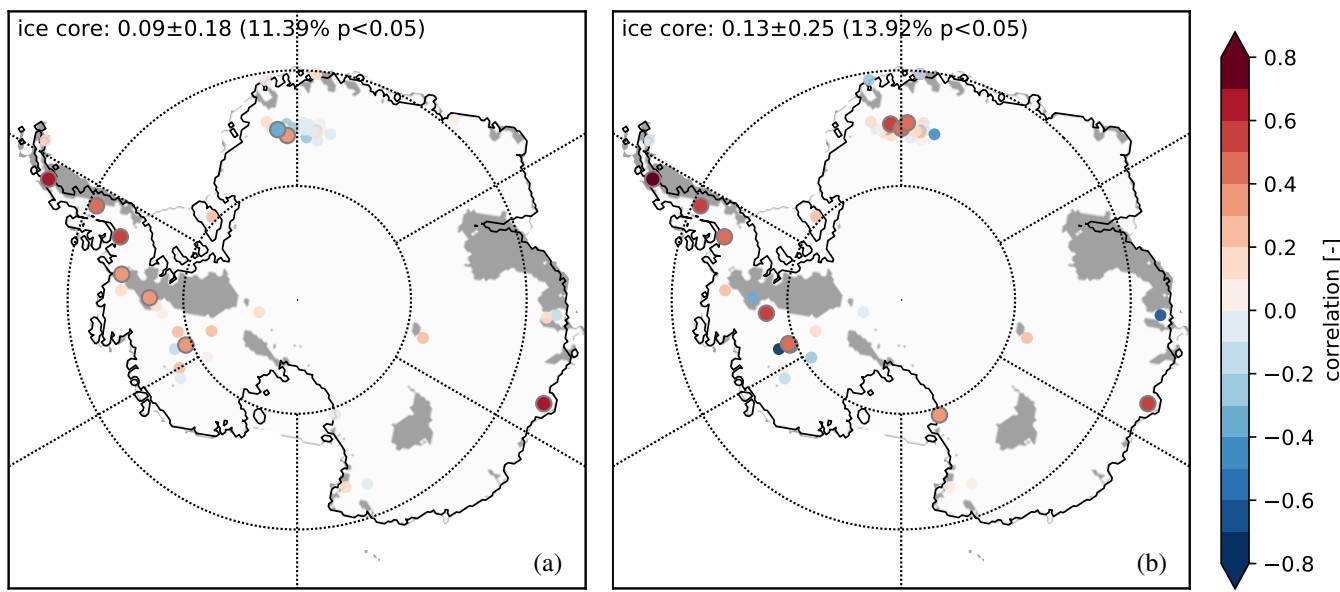

**Figure 8.** Ice core correlations (a) annually for SMB-SAT over 1958–2010 AD and (b) 5-yearly for SMB-$\delta^{18}$O over 1871–2010 AD. Large grey encircled dots indicate that the correlation value is significant, smaller dots indicate a p-value > 0.05. Statistically insignificant areas of the RACMO27 SMB-SAT annual correlation ($p > p_{FDR}$) are hashed in grey for reference. The average annual correlation, standard deviation and percentage of ice cores with p < 0.05 is provided on each panel.

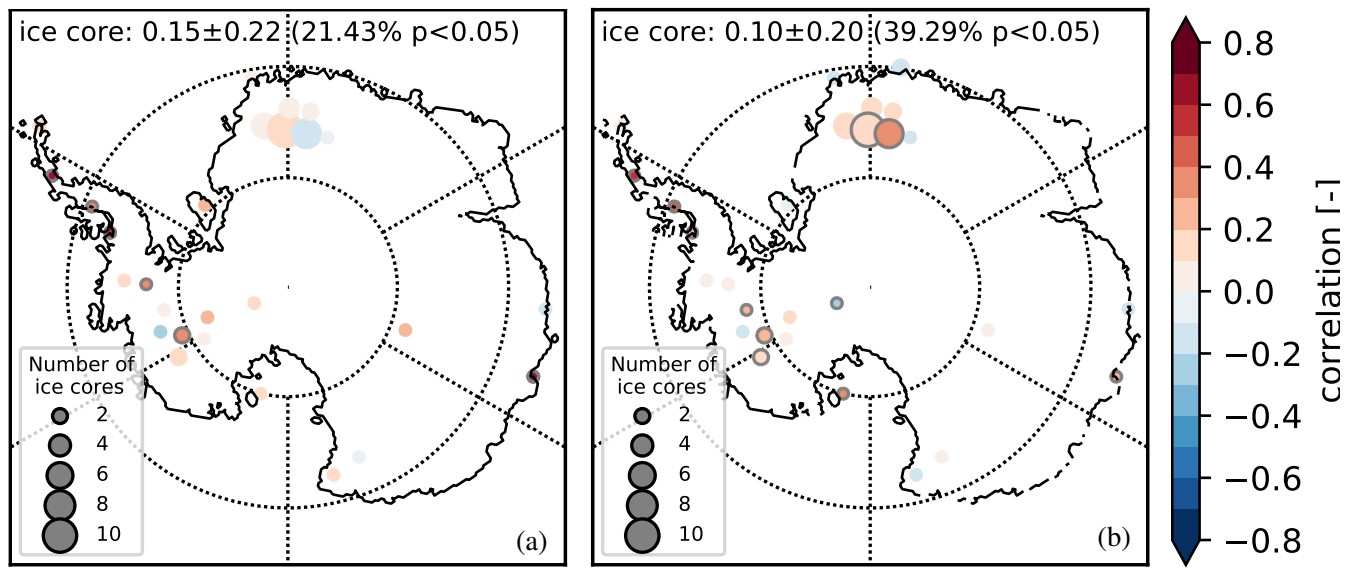

**Figure 9.** Aggregated ice core annual correlations on a 216 x 216 km grid for (a) SMB-SAT annually over 1958–2010 AD and (b) SMB-$\delta^{18}$O 5-yearly over 1871–2010 AD. Grey encircled dots indicate that the correlation value is significant ($p < 0.05$). The size of the dot represents the number of ice cores aggregated, with the legend given on each panel. The average annual correlation, standard deviation and percentage of ice cores with $p < 0.05$ is provided on each panel.

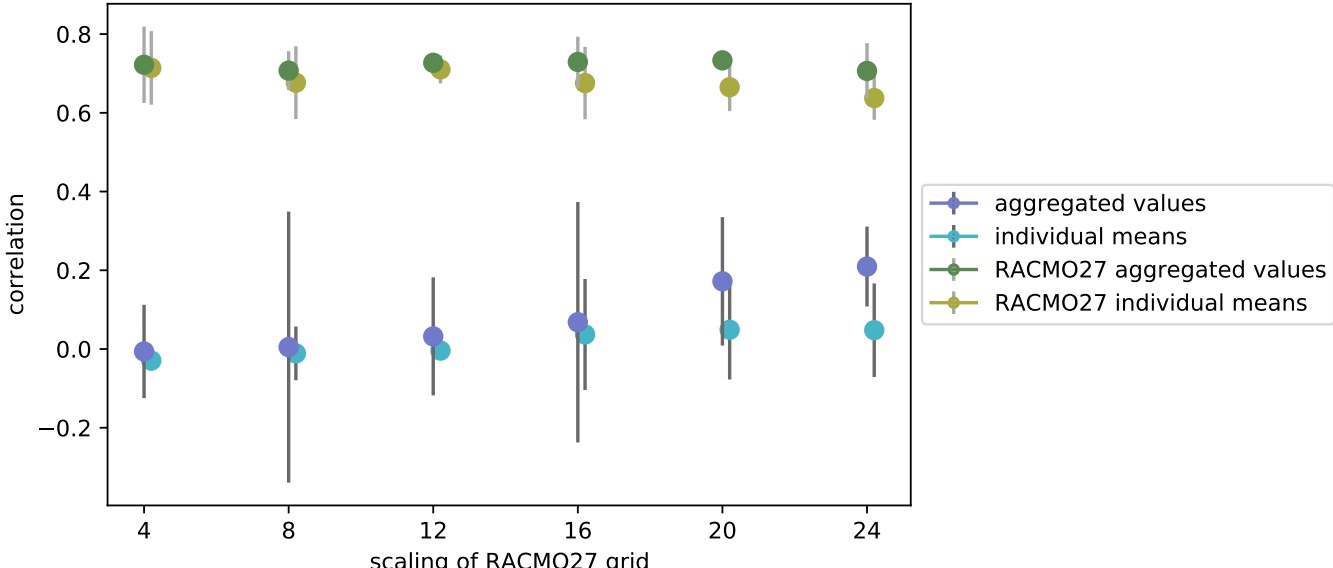

**Figure 10.** SMB-SAT annual correlation as a function of grid spacing for the aggregated records versus mean of the individual annual correlations for the ice cores (dark and light blues, respectively) and RACMO27 simulations (in dark and light green, respectively). The grey bars indicate the full range of correlation values with the mean indicated by the colored dot. For the RACMO27 simulation results, only RACMO27 grid cells where ice cores exist are used. For both ice cores and RACMO27, only grid points with at least five ice cores are kept. Annual correlations are over the 1958–2010 AD time interval for the ice core data, and over 1979–2016 AD for the RACMO27 simulations.

*Data availability.* RACMO2.3 simulations are available by request to J.M. (Melchior) van Wessem (j.m.vanwessem@uu.nl); CMIP5 simulations are available at http://pcmdi9.llnl.gov; iHadCM3 simulations are available by request to Max Holloway (max.holloway@sams.ac.uk); ECHAM5-wiso simulations are available at https://doi.org/10.5281/zenodo.1249604; ECHAM5/MPI-OM simulations are available by request to Jesper Sjolte (jesper.sjolte@geol.lu.se). The $\delta^{18}$O Stenni et al. (2017) compilation is available at https://www.ncdc.noaa.gov/paleo-search/study/22589, the SMB Thomas et al. (2017) compilation is available by a request to Elizabeth R. Thomas (lith@bas.ac.uk), the SAT
Nicolas and Bromwich (2014) compilation is available at http://polarmet.osu.edu/datasets.

*Author contributions.* MGPC and HG designed the experiments and MGPC carried them out. QD provided analysis support, JTML provided RACMO support, ERT provided ice core support. MGPC prepared the manuscript with contributions from all co-authors.

*Competing interests.* The authors declare no competing interests.

*Acknowledgements.* We would like to thank Melchior Van Wessem for the RACMO2.3 model outputs and Jesper Sjolte for the ECHAM5-
MPI/OM model outputs. We would like to thank the editor and the two anonymous reviewers for their helpful and constructive comments which have strongly improved the quality of this manuscript. We acknowledge the CESM1(CAM5) Last Millennium Ensemble Community Project and supercomputing resources provided by NSF/CISL/Yellowstone, the CESM project is supported primarily by the National Science Foundation (NSF). We thank Matthew Brett for his implementation of the FDR method in Python (https://matthew-brett.github.io/teaching/fdr.html). This work was supported by the Belgian Research Action through Interdisciplinary Networks (BRAIN-be) from Belgian Science Policy Office in the framework of the project "East Antarctic surface mass balance in the Anthropocene: observations and multiscale modelling (Mass2Ant)" (Contract n° BR/165/A2/Mass2Ant). Hugues Goosse is the research director within the F.R.S.-FNRS.

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
