# Peer review of "Reconciling the surface temperature—surface mass balance relationship in models and ice cores in Antarctica over the last two centuries"

_The Cryosphere, 2020_

## Referee Comment (RC1) · Anonymous Referee #1 · 1 May 2020

**Review of Cavitte et al. 2020 'Reconciling the surface temperature–surface mass balance relationship in models and ice cores in Antarctica over the last two centuries'**

In this article, Cavitte and co-authors analyse the link between surface mass balance (SMB) and surface air temperature (SAT) over the Antarctic ice sheet, at annual resolution. They focus on the last 200 years (1871–2000).

They use a series of climate model simulations: four global climate models including water stable isotopes (iGCMs) and a regional climate model (RCM) without isotopes. They also use observation-based results : the temperature reconstruction of Nicolas and Bromwich, 2014 (NB14) and ice-core annual to 5-year-mean $\delta$18O (Stenni et al., 2017) and SMB (Thomas et al., 2017).

SAT is supposed to be recorded by the $\delta$18O signal of ice cores. But $\delta$18O and SAT are generally correlated with SMB, as they both result of large scale advection of warm and moist air from lower latitudes.

The aim of the authors is to understand how much SMB and SAT are correlated based on climate simulations and ice core records, and what can explain the strength of correlations at the regional and local scale. They also want to understand what is lacking in our current understanding to explain the observed lower correlation of $\delta$18O-SMB in ice cores than SAT-SMB in models.

It is a very interesting study that I recommend for publication in *The Cryosphere*. However, I pointed out major issues that need to be answered before publication.

**Major (method)**

This is a minor remark, but important for improving the readability of the article. In the article, the main assumption is that $\delta$18O is a proxy of SAT. However, you show 'SMB-SAT' and '$\delta$18O-SMB' correlations. I suggest to write it the same order for both, e.g. 'SMB-SAT' and 'SMB-$\delta$18O', even if it has no effect on correlations.

**iGCM members averaging**

**P5L121** *'we average over their ensemble of simulations to obtain a mean representation of SMB, SAT and 18O for each iGCM.'*

I think this might be a major issue, as when averaging each variable across different simulation, the interannual variability is dampened. The correlation of average is not the average of correlations. It might not change dramatically your results but it should be corrected.

**iGCMs evaluation**

**P5L123** *'Dalaiden et al. (2019) provide an evaluation of the iGCMs used here'*

Model evaluation is too weak. You should show in supplementary key evaluations for the 4 models.

You should add all Dalaiden et al. 2019's evaluations related to the 3 iGCMs you use in your article, and add evaluation of iCESM1, that is not provided in Dalaiden et al. 2019.

As SMB-SAT relationship is driven by atmospheric circulation, it would be good to see how large-scale circulation of these models compare with reanalyses for the satellite era. Showing at least an evaluation of sea level pressure patterns on average for the common period is needed to understand how large the models' biases can be.

**Model selection and averaging.**

**P5L122** *'all four iGCMs show similar spatial variations of the correlation between SMB-SAT and δ18O-SMB on the continent scale [...] To compare the iGCM continent-wide correlations to the RCM-derived correlations, we interpolate the iGCM results onto the RCM grid and average over all four iGCMs.'*

From Fig.S1 we can see that ECHAM5-MIP-OM gives signifanctly different results than the other 3 iGCMS at the ice sheet margins for δ18O-SMB.

From Dalaiden et al. 2019 Fig. S7 and S8, we can see that ECHAM5/MPI-OM have a major issue for modelling SMB. It is of major importance for this article, because as you average the 4 isotopic global climate models (iGCMs) correlations into one single map of correlation, it suppose that you consider equal skills of the 4 models.

Consequently, you should consider disregarding ECAM5-MIP-OM simulations, or at least giving it less weight in the average.

**Major (results and interpretation)**

**Consistencies between models and time scales**

**P6L183** *'Moreover, the maximum and minimum correlations obtained are consistent between iGCMs, in magnitude and spatial distribution (see supplementary Fig.S1-S4).'*

Can you develop the area you think are consistent? Because I see more differences that analogy between the 4 models. ECHAM5-MPI-OM is the only iGCM with a clear loss of correlation at the ice sheet margins in East Antarctica (for both SAT and δ18O). If you exclude ECHAM5-MPI-OM, I see some consistencies between the 3 models left, but still, it's not very clear given the patchy patterns.

**P7L190** *'This implies that the correlation of SMB and SAT is similar over the 1871–2000 AD and the 1979–2016 AD time intervals, and from a spatial resolution of >1 down to 5.5 km.'*

Please clarify what you think is similar between the resolutions and the time periods. I see many differences, so if you want to highlight the consistencies, you should detail them (e.g. high correlations for West AIS?).

**Generally, I am concerned about the too optimistic way of presenting consistencies between simulations.**

**Wind effect**

**P7L196** *'There are a few areas, spatially consistent between the RACMO27, RACMO5 simulations **and the iGCMs**, where the SMB-SAT correlation is not as strong.'*

Again, I am not sure to agree with the authors. If you look at SMB-SAT correlation on Fig.S4 compared to Fig.2, it is not clear that there is a loss of correlation in the iGCMs at the "same areas" than in the RCM. It seems that combining the 4 iGCMs gives by chance the same pattern at in RACMO2?

**P7L204** *'Large-scale air masses, originating over the Southern Ocean and further north, bring warm moist air towards the interior as they flow up-slope, thus inducing a strong and negative correlation'*: You suggest that at the interannual time scale, when you have upslope winds (so more negative) you have higher temperature and SMB, and the more positive is mean slope in mean wind direction (MSWD) the colder and the dryer. It seems reasonable, but it would be really good to explicit this, because I had trouble trying understanding the positive/negative correlations with wind.

Can you show time series of MSWD together with time series of SAT and SMB at some specific locations, so that we can understand what the correlation means? E.g. at locations where it's significantly positively/negatively correlated to SAT/SMB, and on the coast/on the plateau, or at least examples for your cases (1) and (2), and for Adelie Land vs. Amery Embayment.

In addition, I am wondering how much annual MSWD is a good indicator of the mechanism you want to highlight. Advection of warm and moist air by cyclones are punctual whereas surface winds generally flow downslope all year long. Consequently I am not sure how much you capture the cyclone activity with MSWD.

You discuss this with cases (1) and (2), but it seems very speculative. Basing your statement on time series will help developing more robust analyses.

I was lost reading the wind considerations, but after re-reading it a couple of times, I think I agree with most of your conclusions. It would be better to re-write this part before I can give a better feedback. **I do think it is very interesting to analyse the strong correlations between MSWD, SMB and SAT at the interannual time scale.** You should use the different components of SMB available in RACMO to analyse the effect of wind on this components (precipitation, drifting snow fluxes, sublimation, etc.), instead of trying to guess why it works this way.

**Related minor comments:**

**Eq. (1)** You must remove the arrow on MSWD as it is a scalar.

**Fig.S9** Can you show MSWD too? To see where it's negative and positive?

**P7L215** *'We then remove areas of the AIS with a negligible slope (¡0.001) as in these areas, MSWD will be close to null and will introduce a lot of noise when correlating it with SMB or SAT. '*: Shouldn't you remove areas where SMB interannual variability and MSWD interannual variability are small too? I would suspect that all the high (negative) correlation for MSWD-SAT and MSWD-SMB found in the EAIS plateau are because of very small interannual variability in wind and SMB(?).

**P8L223** *'Agosta et al. (2019) also show a strong link between modelled surface topography (surface curvature in their case) with SMB over the continent when wind speeds exceed 5 m s1 .'*: but in Agosta et al. 2019, it is a spatial link of time-averaged values, not a temporal link.

**P8L264** *'We therefore expect that the areas regularly under the influence of strong katabatic winds will show a weaker MSWD-SMB correlation due to the episodic but persistent reduction in their SMB through wind scouring'.* Here is one out of many examples where you can use RACMO outputs to verify your hypothesis, since you have access to all surface mass balance fluxes, including drifting snow fluxes, in this model.

**P10L288** *'Perhaps here snowfall input from further north is so high that it dominates the SMB and SAT records.'* The same: you can use RACMO outputs to clarify what's happening.

**Suspicion of wrong SMB interannual variability in ice cores**

Sections 3.2 and 3.3 study the SMB-SAT link in ice core data, and aims at understanding why it is much weaker in ice cores than in models.

**P10L307** *' We observe a weak-to-null annual correlation between SAT and SMB in the ice cores (Fig.7) with an average value of 0.09±0.18 over all the ice cores, versus a continent-wide average value of 0.57±0.10 for the iGCMs and 0.54±0.22 for RACMO27.'*

I am wondering whether the low correlation between ice cores SMB and NB14 SAT is because of a too low or incorrect interannual variability of SMB in ice cores.

I think this analysis is very interesting, but I cannot evaluate it further if I am not sure that the difference between ice cores and models is not because of wrong SMB interannual variability in ice cores.

Can you show annual time series of ice core SMB vs. RACMO SMB? The results will have a major impact for reviewing the rest of the study.

**Minor comments**

**P2L35** *'temperature'*: heat

**P2L35** remove *'usually'*

**P2L35-36** *'that collect heat and moisture from further north, including the Southern Ocean, which they can release onto the AIS'*

It's the same idea and mechanism than described in the previous sentence:

*'Large-scale atmospheric circulation strongly controls SMB in Antarctica, bringing air masses with a high moisture and temperature content'*

Merge the sentences.

**P2L43-44** *'In addition, based on the Clausius-Clapeyron relationship, the increasing surface air temperature (SAT) due to climate change should induce a greater moisture holding capacity of the air'*: It's the increasing of air temperature (in the mid and upper troposphere) that induces an increase in moisture holding capacity and more precipitation, not the increase in *surface* air temperature. The confusion here comes from the fact that increase in SAT and in tropospheric air temperature are strongly correlated.

**P2L49** *'is often used as a proxy for SAT'*: Is it used as a proxy of SAT or a proxy of snowfall-weighted SAT?

**P3L84** AP is not introduced before

**P3L86** *'over historical timescales'*: give the time period and the resolution.

**P4L91** *'1. Which processes link SAT and SMB at regional scales and how do they scale down from conclusions at the continental scale'*: You don't answer to this question.

**P4L98-99** *'We hypothesize physical mechanisms that could explain the discrete areas of the AIS where the SMB-SAT relationship is weak.'* Here you suppose this relationship is always strong. But in this article, you don't analyse why the SMB-SAT relationship is generally strong.

**P5L122** *'all four iGCMs show similar spatial variations of the correlation between SMB-SAT and δ18O-SMB on the continent scale'*: You didn't give on which time scale you correlate the time series. You should emphasis the time step (annual, from the text after) in the method, by always associating "correlation" with "annual".

**P6L172** *'Furthermore, the spatial distribution of the ice cores over the AIS is not homogeneous, with the majority of the ice cores located in the coastal areas, and very few in the interior (see supplementary Fig.S5). This certainly introduces a spatial bias in our ice core-based correlation towards coastal signals and processes.'*

The lower the accumulation, the lower the ice core resolution. So it is expected that annually resolved ice cores will be at the ice sheet margins.

In addition, you should not write "coastal", as most of the ice cores are not at the coast but inland. You can maybe divide Antarctica in "low elevation" (<2200 m) and "high elevation" (>2200 m).

**P7L185** *'We repeat the annual correlation of SAT and SMB using the RACMO27 simulations over 1979–2016 AD (see Fig.2).'*: Merge Fig. 2 to Fig. 1 so that we can compare patterns between iGCMs and RACMO2.

**P7L200** *'Winds are known to affect SMB and SAT locally, through wind-based redistribution of SMB, turbulent warming from katabatics and Foehn warming effects on leeward slopes. '*: you should specify that the loss of correlation because of wind-based redistribution of SMB can be modelled by RACMO only, because it is the only model in this study that includes drifting snow modelling.

**P7L200** *'turbulent warming from katabatics and Foehn warming effects on leeward slopes'*: katabatics and surface winds in general (e.g. pressure gradient winds superimposed with katabatics) are also directly concerned by adiabatic warming, the same process involved in the foehn warming.

**P11L347** *' Turner et al. (2019) show that more than 70% of the annual accumulation consists of extreme events that have a very short duration (one or more consecutive days).'*: warning. Turner et al. (2019) show that more than 70% **of the variance** of the annual precipitation is explained by extreme precipitation events, meaning the interannual variability, not the mean value.

**P14L427** *' to improve our confidence in using SMB as a direct proxy for SAT over the entire AIS.'* You never mentioned this objective before. Can you develop it in the introduction?

---

## Referee Comment (RC2) · Anonymous Referee #2 · 21 Jun 2020

In this article the authors investigate the relationship between the surface mass balance (SMB), surface air temperature (SAT) and $\delta$18O in models and ice cores. They analyze output of four GCM models and one regional climate model of which the GCMs are aware of isotopes. The dataset of ice cores consists of about 50 ice cores. They report a strong positive correlation between modelled SMB and SAT as well as $\delta$18O and SMB, which is in agreement with previous studies. However, there are regions where the relationships break for which the authors investigate reasons. Mainly they find a dependence on the prevailing winds that can strongly affect the relationship by e.g. leeside warming due to Föhn winds or adiabatic warming of the falling katabatic winds along the coast. Additionally, they report that the correlations found in ice cores are

much lower than in the model and discuss reasons for this discrepancy. Overall, this is a very interesting study, which I recommend for publication in The Cryosphere. Before publishing, some major and a number of minor points should be clarified/improved as listed below.

Major

(1) Correlation testing The authors present a number of correlation maps. When conducting a high number of significance tests on a gridded map, this increases the number of false rejection of the null hypothesis, (e.g. Wilks 2016). There are correction methods for this issue (e.g. Benjamin and Hochberg, 1995), which the authors should apply in their study.

Benjamini, Y. and Hochberg, Y.: Controlling The False Discovery Rate – A Practical And Powerful Approach To Multiple Testing, J. Roy. Stat. Soc. Ser. B, 57, 289–300, 1995.

Wilks, D. S.: "The Stippling Shows Statistically Significant Grid Points": How Research Results are Routinely Overstated and Overinterpreted, and What to Do about It, B. Am. Meteorol. Soc., 97, 2263–2273, https://doi.org/10.1175/bams-d-15-00267.1, 2016.

(2) Gridding of ice core data The authors describe their method to analyze the ice core data on a grid. They mention that it is important to have enough ice cores per grid cell, however, the information about how many ice core measurements used per gridded value is missing (a figure or table in the supplementary material could do the job). First, they present gridded ice core results at 108 km x 108 km and 216 km x 216 km. Finally, they analyze results where they analyze results for different resolution between 108 km x 108 km and 684 km x 648 km (Fig. 9) where they only take into account grid cells with five or more ice cores. Based on figure 9, this criteria is not fulfilled for 108 km x 108 km. Thus, I wonder how many ice cores are available in the 108 km x 108 km and 216 km x 216 km grid cells. Does the analysis presented for 108 km x 108 km and 216 km x 216 km actually remove the general noise? For how many locations do

these grid cells take into account more than one or two ice cores? This needs to be clearly stated as it will have a strong impact on the analysis.

(3) Figure 9 Figure 9 needs strong improvement. In the caption it is stated that the size of the colored dots is a function of the number of ice cores aggregated. There is need for a legend giving information about the actual number of ice cores for the different dot sizes. The dots for the model in the legend are far too small (black and gray dots). The dots in the legend could all have the same size.

Minor

General:

- Avoid "see Figure . . .", "shown on Fig. . . ." just reference the figures. For the figures add a space after Fig., i.e. Fig.1 -> Fig. 1

- Avoid spoken language, i.e. try to reduce the number of non-informative sentences, e.g. "Let us first focus on the link between wind redistribution and SAT." (P.8, L228) or "To explain this, we have to first describe average conditions." (P.8, L247) or "Now that we have a better understanding from the models let us look at . . ." (P.10, L.398). These are some examples but there are more in the text.

Specific:

P.2, L24: sea-level rise -> SLR

P.2, L35: temperature -> heat

P.2, L34-36: The first sentence leaves the reader wondering where the air comes from. Consider reformulation of the two sentences.

P.2, L45: twice mentioned the increased SAT. Is it a positive feedback, i.e. increasing SAT -> increased snowfall -> increase in SAT? If yes, consider reformulation, if no remove one of the increased SAT.

P.3, L80: at -> as

P.3, L85: (Dalaiden et al., 2019) -> Dalaiden et al. (2019)

P.3, L87: Here, for the second time you define the local scale. Redefine one or link them. Also, in the Results and discussion the "local scale" is only used once.

P.4, L91-95: Add "?" to the questions

P.4, L107: "(with the additional fourth made available recently, . . .)" Mention which model is new.

P.4, L108: Add the reference of Brady et al., 2019 as not all the models are described in Dalaiden et al., 2019.

P4. L114: "by a sea surface temperatures . . . " -> "by sea surface temperatures"

P.5, L120-121: The authors use a different number of ensemble members for the different iGCMS. By averaging over several ensemble members variability is lost. It would be interesting to know how this affects the results, please discuss.

P.5, L121: Consider reformulations "we average over their ensemble of simulations to obtain mean representation of SMB, SAT and $\delta$18O for each iGCM." -> "we retrieve the ensemble mean of SMB, SAT and $\delta$18O for each iGCM."

P.5, L123: Mention the most important findings of the evaluation. What about the model that is not evaluated in Dalaiden et al., 2019?

P.5, L126: significative -> significant

P.5, L129-130: Explicitly give the number of years taken into account. Avoid "$\sim$" as the time periods are clearly defined.

P.5, L130/P.5, L141: Please clarify the choice of the time periods. You mention ". . .a shorter () timescale covering 1961-2000 AD for comparison to the RCM simulations and measured SAT." However, the RCM start in 1979. Why don't you take a time period

that matches between the iGCMs and the RCM? I guess this would be 1979-2000?

P.5, L132: Here and in the Figures 1, and S1-S4 you first mention $\delta 18O$-SMB followed by SMB-SAT. However, in the analysis you first analyze SMB-SAT. Flip the order in both the text (here) and the figures.

P.5, L143: to studying -> to study

P.5, L147: remove "," before etc.

P.6, L158: Reformulate: "Ice cores record variations with depth of ice's $\delta 18O$."

P.6, L164-168: How well does this temperature reconstruction method work? Why do you need to rely on this? Could you use SAT from reanalysis instead? Are there weather stations in the close proximity of the ice core locations? For locations where weather stations are available, how well does the SAT reconstruction method agree with the measurements? Could you present correlations of ice cores to station measurements for some locations?

P.6, L177: Consider starting the results and discussion section with your results and discuss the comparison to other studies later.

P.6, L184: Figures S1 and S2 are about $\delta 18O$. Don't mention them here. I think only Fig. S3 is relevant to this section. Reorder the figures in the supplementary material and only mention the related ones.

P.7, L191: Avoid generalizing, as you only have two different resolutions and two different time periods (which are partially overlapping).

P.8, L223-224: How do the findings by Agosta et al. (2019) relate to your findings? Indicate the link more clearly.

P.8, L225: "very positive", reformulate to "strongly positive" or "positive".

P.8, L225: "weaker-to-positive" this needs to be reformulated. What you refer to is a

weaker-negative-to-positive correlation. This needs to be specified.

P.8, L226: "(outlined here with a magenta line)": Why using a magenta line (Figs. 4 and 5) and not the black dashed lines as in Fig. 2? Magenta lines are harder to see. Remove "here", i.e. "(outlined with a magenta line)".

P.8, L231: "very positive" -> "strongly positive"

P.8, L232: To me it looks like the effect is not present over the whole range of the Trans-Antarctic Mountains, i.e. I can't see the effect for the eastern part of the mountain range. Please, clarify.

P.8, L236: "this leeward side" -> "the leeward side"

P.8, L.239: "here" -> "in this area"

P.8, L.241: Be specific about the difference between the low level westward winds and upper level winds bringing the warm air masses from the north.

P.8, L.244: Did you test whether the correlation is significantly less negative? Otherwise, avoid using significantly.

P.9, L.255: "... become weaker or increase". This is not clear, please clarify. Do you mean "become weaker or positive"?

P.9, L.256: "bottom panel" -> use a), b), ... for all the figure panels.

P.9, L.268: Instead of "marked on Fig. 2" use either "AE on Fig. 2" (or "marked as AE on Fig. 2"). Same for BG (P.9, L274) and AL (P.9, L.277).

P.9, L.271: "i.e." instead of "so"

P.9, L.272: Change "very positive" to "strongly positive"

P.9, L.277: Does the model confirm especially high precipitation in the Adélie Land?

P.9, L.282: The lines on Fig. 2 are not magenta but black dashed.

P.10, L.288: "Perhaps here snowfall input from further north is so high that it dominates the SMB and SAT records." Do the model results confirm this? The models might have the different components of the SMB to check this.

P.10, L.290: RACMO05? (= RACMO5?)

P.10, L.295: Try to avoid generalization. Consider changing "all scales" to "all investigated scales".

P.10, L.310: Reference figures for iGCMs and RACMO.

P.11, L.318: ".. an important process, process that . . ." -> ". . . an important process, i.e. a process that.." or ". . . an important process, that . . ."

P.11, L.393: instead of "as discussed below" reference Section.

P.11, L.342ff: You mention that the correlations of the cores with an inhomogeneous distribution over the continent are probably not representing the continental or even regional correlation. Then, you mention that the model shows mainly positive correlations where the ice cores are located. Did you consider calculating the correlation of the model only for grid cells for which ice core data exists? How does the comparison compare in this case? Could this result be used to underline the disagreement?

P.11, L.344: remove "outlined"

P.12, L.356: reference the 108x108 km results as "not shown"

P.12, L.366: "Results are shown in Fig. 9." No need for a whole sentence, just give reference as (Fig. 9).

P.12, L.368: "all spatial scales": Avoid generalization, i.e. "all investigated spatial scales"

P.12, L.375-376: Please clarify this sentence.

P.21, Fig.7: significantive -> significant

P.21, Fig.7: Maybe change "Large dots indicate…" to "Large gray encircled dots indicate…"

P.22, Fig.8: Why don't you show the standard deviation and percentage of p<0.1 on these graphs? What does the size of the dots display? Provide information in the caption and a legend for sizes.

Suppl., P.8, Fig S9: specify on which level wind speed and direction is shown.

---

## Author Comment (AC1) · 16 Jul 2020

We thank the reviewer for their constructive comments. We have responded to all of them and will modify the paper accordingly. Our point-by-point answers follow as a supplement. Please note that review comments are in grey italics while our answers are not. Changes/additions to the original manuscript are indicated in blue.

Please also note the supplement to this comment:
https://tc.copernicus.org/preprints/tc-2020-36/tc-2020-36-AC1-supplement.pdf

[Figure]

**Supplement:**

**Response to RC1 comments on the submitted paper** *Reconciling the surface temperature–surface mass balance relationship in models and ice cores in Antarctica over the last two centuries*

We thank the reviewer for their constructive comments. We have responded to all of them and will modify the paper accordingly. Our point-by-point answers follow.

Please note that review comments are in grey italics while our answers are not. Changes/additions to the original manuscript are indicated in blue.

**Answers to RC1**

*In this article, Cavitte and co-authors analyse the link between surface mass balance (SMB) and surface air temperature (SAT) over the Antarctic ice sheet, at annual resolution. They focus on the last 200 years (1871–2000).*

*They use a series of climate model simulations: four global climate models including water stable isotopes (iGCMs) and a regional climate model (RCM) without isotopes. They also use observation-based results : the temperature reconstruction of Nicolas and Bromwich, 2014 (NB14) and ice-core annual to 5-year-mean $\delta^{18}O$ (Stenni et al., 2017) and SMB (Thomas et al., 2017).*

*SAT is supposed to be recorded by the $\delta^{18}O$ signal of ice cores. But $\delta^{18}O$ and SAT are generally correlated with SMB, as they both result of large scale advection of warm and moist air from lower latitudes. The aim of the authors is to understand how much SMB and SAT are correlated based on climate simulations and ice core records, and what can explain the strength of correlations at the regional and local scale. They also want to understand what is lacking in our current understanding to explain the observed lower correlation of $\delta^{18}O$-SMB in ice cores than SAT-SMB in models.*

*It is a very interesting study that I recommend for publication in The Cryosphere.*

*However, I pointed out major issues that need to be answered before publication.*

**Major (method)**

*This is a minor remark, but important for improving the readability of the article. In the article, the main assumption is that $\delta^{18}O$ is a proxy of SAT. However, you show 'SMB-SAT' and '$\delta^{18}O$-SMB' correlations. I suggest to write it the same order for both, e.g. 'SMB-SAT' and 'SMB-$\delta^{18}O$', even if it has no effect on correlations.*

**Answer:** This has been changed everywhere in the manuscript.

**iGCM members averaging**

**P5L121** *'we average over their ensemble of simulations to obtain a mean representation of SMB, SAT and 18O for each iGCM.' I think this might be a major issue, as when averaging each variable across different simulation, the interannual variability is dampened. The correlation of average is not the average of correlations. It might not change dramatically your results but it should be corrected.*

**Answer:** We are sorry that the wording of this sentence was confusing. What we did was: (1) we calculated the correlation of the two variables (SMB-SAT or SMB-$\delta^{18}O$) for every grid point for each model member (3 members for iCESM1 and 7 members for iHadCM3). (2) We calculated the mean of the correlation values obtained per grid point over all the model members that belong to iCESM1 or iHadCM3 to get the mean SMB-SAT and SMB-$\delta^{18}O$ correlation for each model. (3) We interpolated then all four iGCM correlation results (ensemble means for iCESM1 and iHadCM3, ECHAM5-MPI/OM and ECHAM5-wiso correlation results) onto the RACMO27 grid, for both the SMB-SAT annual correlations and the SMB-$\delta^{18}O$ 5-yearly correlations. (4) We then calculated the mean over all four iGCMs to get the resulting iGCM plots shown in Fig.1. The sentence has now been changed to: "As the iCESM1 and iHadCM3 ensembles include three and seven simulations, respectively (each has slightly different initial conditions), we first calculate the correlation of the two variables (SMB-SAT or SMB-$\delta^{18}O$) for every grid point for each ensemble member (3 for iCESM1 and 7 for iHadCM3). Then we obtain the mean of the correlation values per grid point over each ensemble of simulations for both iCESM1 or iHadCM3. These ensemble means can then be compared to the correlations calculated for ECHAM5-MPI/OM and ECHAM5-wiso."

**iGCMs evaluation**

*P5L123 'Dalaiden et al. (2019) provide an evaluation of the iGCMs used here'. Model evaluation is too weak. You should show in supplementary key evaluations for the 4 models.*

*You should add all Dalaiden et al. 2019's evaluations related to the 3 iGCMs you use in your article, and add evaluation of iCESM1, that is not provided in Dalaiden et al. 2019.*

*As SMB-SAT relationship is driven by atmospheric circulation, it would be good to see how large-scale circulation of these models compare with reanalyses for the satellite era. Showing at least an evaluation of sea level pressure patterns on average for the common period is needed to understand how large the models' biases can be.*

**Answer:** As suggested, we will provide an evaluation of the SMB, SAT and SLP (Sea-Level Pressure) for the 4 iGCMs in the supplement. SMB and SAT will be evaluated against RACMO27, while SLP will be evaluated against ERA-Interim as the RACMO27 grid does not extend far enough north. The evaluation will be done over the 1979–2000 AD time period, the longest period of overlap for all the models examined. We will further discuss the relevance of this evaluation in the manuscript in this section (P5). Note that the Dalaiden et al. (2019) The Cryosphere Discussions paper has now been published in The Cryosphere and all references have been changed to Dalaiden et al. (2020), here and in the manuscript.

*Model selection and averaging*

*P5L122 'all four iGCMs show similar spatial variations of the correlation between SMB-SAT and $\delta^{18}O$-SMB on the continent scale [...] To compare the iGCM continent-wide correlations to the RCM-derived correlations, we interpolate the iGCM results onto the RCM grid and average over all four iGCMs.' From Fig. S1 we can see that ECHAM5-MIP-OM gives significantly different results than the other 3 iGCMS at the ice sheet margins for $\delta^{18}O$-SMB. From Dalaiden et al. 2019 Fig. S7 and S8, we can see that ECHAM5/MPI-OM have a major issue for modelling SMB. It is of major importance for this article, because as you average the 4 isotopic global climate models (iGCMs) correlations into one single map of correlation, it suppose that you consider equal skills of the 4 models. Consequently, you should consider disregarding ECHAM5-MIP-OM simulations, or at least giving it less weight in the average.*

**Answer:** We agree that ECHAM5-MPI/OM differs from the other 3 iGCMs in representing SMB, based on Dalaiden et al. (2020) Fig. S7 and S8. Nevertheless, selecting a criteria to keep or remove a model from our analyses may be difficult to justify objectively as all models have strengths and weaknesses. We propose to keep all 4 iGCMs in this study, but move figures S1 and S3 (that display the SMB-SAT and SMB-$\delta^{18}O$ correlations over the 1871–2000 AD time interval) to the main manuscript. The goal is to discuss the similarities and differences between the models more explicitly and in particular discuss if the conclusions that could be derived from ECHAM5-MPI/OM differ from the ones obtained in the other models. We will also use the iGCM evaluations from the point above to support the discussion in this paragraph of the manuscript.

*Major (results and interpretation)*

*Consistencies between models and time scales*

*P6L183 'Moreover, the maximum and minimum correlations obtained are consistent between iGCMs, in magnitude and spatial distribution (see supplementary Fig.S1-S4).'*

*Can you develop the area you think are consistent? Because I see more differences that analogy between the 4 models. ECHAM5-MPI-OM is the only iGCM with a clear loss of correlation at the ice sheet margins in East Antarctica (for both SAT and $\delta^{18}O$). If you exclude ECHAM5-MPI-OM, I see some consistencies between the 3 models left, but still, it's not very clear given the patchy patterns.*

**Answer:** We will expand the description of the similarities in this paragraph, and also include a description of the areas that are dissimilar. We will define several areas (specific boxes) over the continent, some where we observe a consistently positive correlation (a specific area situated over the Antarctic Plateau),

some where we observe a consistently weak correlation (an Amery Embayment, Peninsula, and a WAIS area), and we will calculate the average/max correlation, and also the % of non-significant values for each of these boxes for each iGCM to compare the iGCM correlation results quantitatively in our description.

*P7L190 'This implies that the correlation of SMB and SAT is similar over the 1871–2000 AD and the 1979–2016 AD time intervals, and from a spatial resolution of >1 down to 5.5 km.' Please clarify what you think is similar between the resolutions and the time periods. I see many differences, so if you want to highlight the consistencies, you should detail them (e.g. high correlations for West AIS?).* **Generally, I am concerned about the too optimistic way of presenting consistencies between simulations.**
**Answer:** We agree that we were too vague in our description and will describe the similarities and dissimilarities in more details in that paragraph, the same way that we propose to do it for the iGCMs (see comment above). We will use the same areas ("boxes") to quantify the average/max correlations and the % of non-significant values for each box.

*Wind effect*
*P7L196 'There are a few areas, spatially consistent between the RACMO27, RACMO5 simulations and the iGCMs, where the SMB-SAT correlation is not as strong.' Again, I am not sure to agree with the authors. If you look at SMB-SAT correlation on Fig.S4 compared to Fig.2, it is not clear that there is a loss of correlation in the iGCMs at the "same areas" than in the RCM. It seems that combining the 4 iGCMs gives by chance the same pattern as in RACMO2?*
**Answer:** We expect that the SMB-SAT correlation won't be identical. We agree with the reviewer that we were too general and optimistic in our description. In light of having changed Fig.1 and Fig.2 to show the individual iGCM correlation results (see a few comments above), we will discuss more specifically the areas that we see as showing a similar loss of SMB-SAT correlation between the iGCMs and RACMO2.3.
In particular, we will stress that the Amery Embayment shows as non-significant in RACMO2.3 as well as ECHAM5-wiso and iCESM1. The loss of correlation along the AP seen in RACMO2.3 is visible also on iCESM1 and ECHAM5-wiso. iCESM1 shows a bit of the coastal variability seen in RACMO2.3, in part due to its finer grid. ECHAM5-wiso also has a fine grid but because we only keep model grid points that are entirely continental using the ECHAM5-wiso mask, we have less constraints along the coast.

*P7L204 (1) 'Large-scale air masses, originating over the Southern Ocean and further north, bring warm moist air towards the interior as they flow up-slope, thus inducing a strong and negative correlation': You suggest that at the interannual time scale, when you have upslope winds (so more negative) you have higher temperature and SMB, and the more positive is mean slope in mean wind direction (MSWD) the colder and the dryer. It seems reasonable, but it would be really good to explicit this, because I had trouble trying understanding the positive/negative correlations with wind.*
*(2) Can you show time series of MSWD together with time series of SAT and SMB at some specific locations, so that we can understand what the correlation means? E.g. at locations where it's significantly positively/negatively correlated to SAT/SMB, and on the coast/on the plateau, or at least examples for your cases (1) and (2), and for Adelie Land vs. Amery Embayment.*
*(3) In addition, I am wondering how much annual MSWD is a good indicator of the mechanism you want to highlight. Advection of warm and moist air by cyclones are punctual whereas surface winds generally flow downslope all year long. Consequently I am not sure how much you capture the cyclone activity with MSWD.*
*(4) You discuss this with cases (1) and (2), but it seems very speculative. Basing your statement on time series will help developing more robust analyses. I was lost reading the wind considerations, but after*

*re-reading it a couple of times, I think I agree with most of your conclusions. It would be better to re-write this part before I can give a better feedback.* **I do think it is very interesting to analyse the strong correlations between MSWD, SMB and SAT at the interannual time scale.** *You should use the different components of SMB available in RACMO to analyse the effect of wind on this components (precipitation, drifting snow fluxes, sublimation, etc.), instead of trying to guess why it works this way.*

**Answer:** Point #1 - We agree that the sign of the positive/negative correlations is perhaps unclear, we now describe the signs explicitly in the text which we have moved below the description of Eq (1) for clarity: "Since we define a positive MSWD as a wind pointing down-slope, we expect winds coming from the coast up into the interior to have a negative MSWD, and winds flowing from the interior to the coast to have a positive MSWD. Large-scale air masses, originating over the Southern Ocean and further north, bring warm moist air towards the interior as they flow up-slope. At the annual time scale, stronger up-slope winds (negative MSWD anomaly) will therefore bring a higher temperature and SMB (positive anomaly), and thus induce a strong and negative annual MSWD-SAT and MSWD-SMB correlation. Similarly, winds flowing down-slope (positive MSWD) will bring drier and colder air (negative anomaly), and will also induce a strong and negative annual correlation of MSWD with SAT and SMB. Any area of the AIS that does not show this negative annual correlation between wind and SAT or SMB implies that the changes cannot be interpreted simply in terms of large-scale circulation variability at the annual scale, and additional processes are affecting the descending cold air from the interior." Furthermore, we will add a small sketch for clarity.

Point #2 - We will show time series of MSWD with time series of SAT and SMB. We will use the AP as a location where MSWD and SAT/SMB are weakly/positively correlated to exemplify the Foehn effect's influence. We will use the Amery Embayment where MSWD and SAT/SMB are weakly/positively correlated and exemplify the influence of the katabatic winds. We will use the plateau region near Dome A (80.3667°S, 77.3005°E) and Adélie land where the MSWD-SMB/SAT correlation is strong and negative. We apologize but we could not figure out what the cases 1 and 2 the reviewer was referring to were.

Point #3 - The hypothesis in this paper is that, even if cyclone activity is punctual, by correlating MSWD to SMB/SAT over 1979–2016 AD, we will be capturing the variations of cyclonic activity at interannual timescales, i.e. the variability in the path of cyclones, and therefore variability in the sources of heat and moisture, whether they are dominant or not over katabatic winds. It would be interesting to compare to the SMB and MSWD at the daily-level, to look at individual cyclonic events more reliably. However, at present, daily RACMO2.3p2 simulations are not freely available. We have added a short sentence to this effect in the manuscript in this paragraph: "We are aware that such cyclonic activity is punctual. However, by correlating MSWD to SMB and SAT over 1979–2016 AD, we make the hypothesis that we capture at the first order the variations of cyclonic activity at interannual timescales, and therefore the variability in the sources of heat and moisture."

Point #4 - Although not shown here, we had looked at the influence of the various SMB components on the total SMB and observed that SMB is strongly dominated by precipitation at the annual timescale. Correlating SMB with its precipitation component over 1979–2016 AD gives an average correlation over the whole continent of 0.96. We will calculate the correlation of MSWD with the various components of SMB (sublimation, blowing snow, precipitation) which we will add to the supplement and describe the results in this paragraph of the manuscript.

*Related minor comments:*
*Eq. (1) You must remove the arrow on MSWD as it is a scalar.*
**Answer:** It has been removed.

*Fig.S9 Can you show MSWD too? To see where it's negative and positive?*
**Answer:** Yes, it has been added as a Figure S10

*P7L215 'We then remove areas of the AIS with a negligible slope (0.001) as in these areas, MSWD will be close to null and will introduce a lot of noise when correlating it with SMB or SAT.': Shouldn't you remove areas where SMB interannual variability and MSWD interannual variability are small too? I would suspect that all the high (negative) correlation for MSWD-SAT and MSWD-SMB found in the EAIS plateau are because of very small interannual variability in wind and SMB(?).*
**Answer:** We argue that areas with a small SMB and MSWD interannual variability are also interesting to look at. We only remove areas that have a negligible slope because correlating such a small number with SMB or SAT could introduce a lot of noise. However, the results are not radically different with and without the threshold applied on the slope (see Fig. 1 and 2). It reduces the average continent-wide correlation strength and increases the number of grid cells with p > 0.1 but not significantly.

*P8L223 'Agosta et al. (2019) also show a strong link between modelled surface topography (surface curvature in their case) with SMB over the continent when wind speeds exceed 5 m s-1.': but in Agosta et al. 2019, it is a spatial link of time-averaged values, not a temporal link.*
**Answer:** We agree that Agosta et al (2019) clearly look at the spatial link of time-averaged values while we look at the temporal link, but we wanted to highlight the parallel between their study and our regarding the impact of the interactions between surface curvature and wind speed. We will change the wording in the text as follows: "Agosta et al. (2019) also show the strong impact of surface topography (surface curvature in their case) - wind interactions on modelled SMB by examining the spatial link of time-averaged values. They observe that above a certain threshold, winds will affect SMB locally in pattern that matches that of drifting snow fluxes as modeled by RACMO2.3.". We plan to also cite the work of Dattler et al. (2019) who show that, in their study area, the variability in accumulation is correlated with MSWD variability also.

*P8L264 'We therefore expect that the areas regularly under the influence of strong katabatic winds will show a weaker MSWD-SMB correlation due to the episodic but persistent reduction in their SMB through wind scouring'. Here is one out of many examples where you can use RACMO outputs to verify your hypothesis, since you have access to all surface mass balance fluxes, including drifting snow fluxes, in this model.*
**Answer:** We agree, and related to a comment we responded to further up, we will show the correlation of MSWD with the various components of SMB in the supplement and provide discussion of these correlations in this section of the manuscript.

*P10L288 'Perhaps here snowfall input from further north is so high that it dominates the SMB and SAT records.' The same: you can use RACMO outputs to clarify what's happening.*
**Answer:** We agree, same comment as above.

*Suspicion of wrong SMB interannual variability in ice cores*
*Sections 3.2 and 3.3 study the SMB-SAT link in ice core data, and aims at understanding why it is much weaker in ice cores than in models.*
*10L307 'We observe a weak-to-null annual correlation between SAT and SMB in the ice cores (Fig.7) with an average value of 0.09±0.18 over all the ice cores, versus a continent-wide average value of 0.57±0.10 for the iGCMs and 0.54±0.22 for RACMO27.' I am wondering whether the low correlation between ice cores SMB and NB14 SAT is because of a too low or incorrect interannual variability of*

*SMB in ice cores. I think this analysis is very interesting, but I cannot evaluate it further if I am not sure that the difference between ice cores and models is not because of wrong SMB interannual variability in ice cores. Can you show annual time series of ice core SMB vs. RACMO SMB? The results will have a major impact for reviewing the rest of the study.*

**Answer:** We agree with the reviewer that it is difficult to quantify whether the low correlation between ice cores SMB-SAT is because of a too low or incorrect SMB interannual variability of SMB in the ice cores. Ice core SMB errors are difficult to constrain (measurement errors, representativity errors due to processes such as a wind ablation that removes a part of the deposited annual snowfall). In addition, missing one year in the dating of the ice cores, for example, can shift the time series, and then the correlation between ice core-derived SMB and the independent SAT time series would be lost. However, we show (Figs. 7 and 8 of the manuscript) that the correlation between SMB and $\delta^{18}$O, both ice core-derived records, is also low. It suggest that the low correlation is perhaps unrelated to a bad synchronization of the SMB and SAT variables. RCMs also include errors that are difficult to trace (difficulty of representing blowing snow or diamond dust). RACMO2.3 for example includes blowing snow processes but Agosta et al. (2019) have shown that its spatial variability is under-represented and that it is underestimated by a factor or 3. When comparing RCM and ice core results, it is thus not simple to assess the origin of the differences in interannual variations. The advantage of RCMs is that they are self-consistent: the SMB and SAT values simulated will be linked by the physics of the model. Ice cores on the other hand can have different biases dependent on different variables (ablation, diffusion, measurement errors, location, etc). However, RCMs can only go back as far as we have direct measurements (i.e. 1979 as previous to that, measurement biases increase, e.g. Bromwich et al., 2007). We need the ice core observations to go back further in time. We will add a sentence to that effect in the introduction and in these sections to provide more context to the readers as to why we compare model correlations to the ice core correlations, which we had perhaps not stated clearly enough.

***Minor comments***
***P2L35*** *'temperature' : heat*
**Answer:** Replaced.

***P2L35*** *remove 'usually'*
**Answer:** Removed.

***P2L35-36*** *'that collect heat and moisture from further north, including the Southern Ocean, which they can release onto the AIS'. It's the same idea and mechanism than described in the previous sentence: 'Large-scale atmospheric circulation strongly controls SMB in Antarctica, bringing air masses with a high moisture and temperature content'. Merge the sentences.*

**Answer:** The two sentences have been merged as: "Large-scale atmospheric circulation (100s of km) strongly controls SMB in Antarctica. This large-scale atmospheric circulation embeds synoptic-scale cyclones that collect heat and moisture from further north, including the Southern Ocean, which they can release onto the AIS (Gorodetskaya et al., 2014; Sodemann and Stohl, 2009; Wang et al., 2019; Lenaerts et al., 2019)."

***P2L43-44*** *'In addition, based on the Clausius-Clapeyron relationship, the increasing surface air temperature (SAT) due to climate change should induce a greater moisture holding capacity of the air': It's the increasing of air temperature (in the mid and upper troposphere) that induces an increase in moisture holding capacity and more precipitation, not the increase in \*surface\* air temperature. The confusion here comes from the fact that increase in SAT and in tropospheric air temperature are strongly correlated.*

**Answer:** We agree and the sentence has been appropriately reworded: "In addition, based on the Clausius-Clapeyron relationship, the increasing mid and upper troposphere temperature due to climate change implies that the air can include more moisture before saturation, and therefore snowfall is increased (Frieler et al., 2015; Fudge et al., 2016). If this predicted increase in SMB is linked to increasing temperatures in the 21st century, it will be interesting to see if SMB and SAT are linked in the past too since an increase in SAT and tropospheric air temperature are strongly correlated. In which case, SMB records over time will be a helpful tool to constrain past climates (Dalaiden et al., 2020)."

*P2L49* *'is often used as a proxy for SAT': Is it used as a proxy of SAT or a proxy of snowfall-weighted SAT?*
**Answer:** We have changed this to "is often used as a proxy of snowfall-weighted SAT (and by extension of SAT, Stenni et al., 2000[...]".

*P3L84* *AP is not introduced before*
**Answer:** We noticed that the first instance that "Antarctic Peninsular appears is P2L42, and have therefore defined the acryonym there: "Antarctic Peninsula (AP,...)"

*P3L86* *'over historical timescales': give the time period and the resolution.*
**Answer:** We have modified the sentence as follows: "Our work follows from that of (Dalaiden et al., 2020), in which they show that SMB and SAT are strongly correlated interannually at the continental scale over millennial (1000–2005 AD) and historical (1850–2005 AD) timescales"

*P4L91* *'1. Which processes link SAT and SMB at regional scales and how do they scale down from conclusions at the continental scale': You don't answer to this question.*
**Answer:** We have reformulated the question as "Do the processes that link SAT and SMB at the continental scale also play a dominant role at the regional scale?". The title of section 3.1 has also been changed accordingly.

*P4L98-99* *'We hypothesize physical mechanisms that could explain the discrete areas of the AIS where the SMB-SAT relationship is weak.' Here you suppose this relationship is always strong. But in this article, you don't analyse why the SMB-SAT relationship is generally strong.*
**Answer:** This was indeed omitted because P3L86, we briefly summarized Dalaiden et al. (2020)'s conclusions on the SMB-SAT relationship at the continental scale, but this was likely too succinct. We propose to not change this paragraph much (except adding that it is a "positive SMB-SAT relationship" that we compare at the regional and continental scale. But provide more discussion in section 3.1.1 where the strength of the SMB-SAT relationship is described and discussed.
P4L98-99 is now: We start with a brief description of the data sets and models used in this study. Then we compare the positive SMB-SAT relationship, as well the SMB-$\delta^{18}$O relationship, obtained in the models at regional scale to the continental-scale results of Dalaiden et al. (2020).
And we have added the following at the end of Section 3.1.1: "[...] Whatever the temporal and spatial scale, the correlation between SMB and SAT is positive for a large majority of the model grid points. The simple concept that Antarctic precipitation originates mainly from lower latitudes, in the form of warm and wet air masses, that explained the covariance of SMB and SAT at the continental scale for Dalaiden et al. (2020), therefore also applies at the regional scale here. Indeed, for the temporal and spatial scales investigated, the annual correlation between SMB and SAT is positive for a large majority of the model grid points. The correlation between SMB and SAT remains true for summer- or winter-only months [...]"

***P5L122*** *'all four iGCMs show similar spatial variations of the correlation between SMB-SAT and $\delta^{18}O$-SMB on the continent scale' : You didn't give on which time scale you correlate the time series. You should emphasis the time step (annual, from the text after) in the method, by always associating "correlation" with "annual".*

**Answer:** We agree and have added "annual" before correlation in this paragraph, and all the relevant places in the manuscript where we are talking specifically about the annual correlation of two variables, and not correlation values in general.

***P6L172*** *'Furthermore, the spatial distribution of the ice cores over the AIS is not homogeneous, with the majority of the ice cores located in the coastal areas, and very few in the interior (see supplementary Fig.S5). This certainly introduces a spatial bias in our ice core-based correlation towards coastal signals and processes.'*
*The lower the accumulation, the lower the ice core resolution. So it is expected that annually resolved ice cores will be at the ice sheet margins. In addition, you should not write "coastal", as most of the ice cores are not at the coast but inland. You can maybe divide Antarctica in "low elevation" (<2200 m) and "high elevation" (>2200 m).*

**Answer:** We agree with both comments, and have reformulated the paragraph as: "Furthermore, the spatial distribution of the ice cores over the AIS is not homogeneous. If we divide Antarctica into low elevation areas (i.e.<2200 m) and high elevation areas (i.e. >2200 m), the majority of the ice cores are located in the low elevation areas, and very few in high elevation areas (see supplementary Fig.S5). This certainly introduces a spatial bias in our ice core-based annual correlation towards coastal signals and processes."
We will also add a line to mark altitude 2200 m on Fig. S5.

***P7L185*** *'We repeat the annual correlation of SAT and SMB using the RACMO27 simulations over 1979–2016 AD (see Fig.2).': Merge Fig. 2 to Fig. 1 so that we can compare patterns between iGCMs and RACMO2.*

**Answer:** These will be merged. Following earlier comments, we propose that Fig. 1 will now show the SMB-SAT correlation for each of the 4 iGCMs (previously supplementary Fig. S3), the iGCM average (previously Fig. 1, panel c) and the RACMO27 SMB-SAT correlation (previously Fig. 2). And Fig. 2 will now include the SMB-$\delta^{18}$O correlation for each of the 4 iGCMs (previously Fig. S1) and the iGCM average (previously Fig. 1, panel a).

***P7L200*** *'Winds are known to affect SMB and SAT locally, through wind-based redistribution of SMB, turbulent warming from katabatics and Foehn warming effects on leeward slopes.': you should specify that the loss of correlation because of wind-based redistribution of SMB can be modelled by RACMO only, because it is the only model in this study that includes drifting snow modelling.*

**Answer:** We agree and have modified the text adapting your input as follows: "[...]turbulent warming from katabatics and Foehn warming effects on leeward slopes. Additionally, the impact of wind on SMB and SAT due to wind-based redistribution can modify the link between the two variables but this can only be resolved by the RACMO2.3 simulations, because it is the only model in this study that includes drifting snow (although according to Agosta et al. (2019), drifting snow is strongly underestimated). To evaluate whether the lack of correlation between SAT and SMB is due to any of these wind effects, we correlate[...]"

***P7L200*** *'turbulent warming from katabatics and Foehn warming effects on leeward slopes': katabatics*

*and surface winds in general (e.g. pressure gradient winds superimposed with katabatics) are also directly concerned by adiabatic warming, the same process involved in the foehn warming.*

**Answer:** We agree that in both cases, there is a warming effect, so we have modified this line as follows: "turbulent and adiabatic warming from katabatic winds and Foehn effects on leeward slopes"

*P11L347 ' Turner et al. (2019) show that more than 70% of the annual accumulation consists of extreme events that have a very short duration (one or more consecutive days).': warning. Turner et al. (2019) show that more than 70% **of the variance** of the annual precipitation is explained by extreme precipitation events, meaning the interannual variability, not the mean value.*

**Answer:** We have modified the text accordingly: "Turner et al. (2019) show that more than 70% of the variance of the annual precipitation is explained by extreme events that have a very short duration (one or more consecutive days)."

*P14L427 'to improve our confidence in using SMB as a direct proxy for SAT over the entire AIS.' You never mentioned this objective before. Can you develop it in the introduction?*

**Answer:** This was mentioned briefly in the introduction already P2L46 "If this predicted increase in SMB is linked to increasing SAT in the 21st century, it will be interesting to see if SMB and SAT are linked in the past too, in which case SMB records over time will be a helpful tool to constrain past climates." but perhaps the goal "to improve our confidence in using SMB as a direct proxy for SAT over the entire AIS." should be stated more explicitly. We have added this more explicitly at the end of our Introduction section, P4L97: "Answering these questions will help constrain our confidence in using SMB as a direct proxy for SAT over the entire AIS. With relatively few in-situ observations, additional SAT proxies would be extremely beneficial for Antarctic climate reconstructions."

[Figure]

Figure 1: Annual correlation between MSWD and SAT (top) with and (bottom) without the slope >0.1% threshold, using RACMO27 simulations over 1979–2016 AD.

[Figure]

Figure 2: Annual correlation between MSWD and SMB (top) with and (bottom) without the slope >0.1% threshold, using RACMO27 simulations over 1979–2016 AD.

---

## Author Comment (AC2) · 16 Jul 2020

We thank the reviewer for their constructive comments. We have responded to all of them and will modify the paper accordingly. Our point-by-point answers follow as a supplement. Please note that review comments are in grey italics while our answers are not. Changes/additions to the original manuscript are indicated in blue.

Please also note the supplement to this comment:
https://tc.copernicus.org/preprints/tc-2020-36/tc-2020-36-AC2-supplement.pdf

[Figure]

**Supplement:**

**Response to RC2 comments on the submitted paper** *Reconciling the surface temperature–surface mass balance relationship in models and ice cores in Antarctica over the last two centuries**

We thank the reviewer for their constructive comments. We have responded to all of them and will modify the paper accordingly. Our point-by-point answers follow.

Please note that review comments are in grey italics while our answers are not. Changes/additions to the original manuscript are indicated in blue.

**Answers to RC2**

In this article the authors investigate the relationship between the surface mass balance (SMB), surface air temperature (SAT) and d180 in models and ice cores. They analyze output of four GCM models and one regional climate model of which the GCMs are aware of isotopes. The dataset of ice cores consists of about 50 ice cores. They report a strong positive correlation between modelled SMB and SAT as well as d180 and SMB, which is in agreement with previous studies. However, there are regions where the relationships break for which the authors investigate reasons. Mainly they find a dependence on the prevailing winds that can strongly affect the relationship by e.g. leeside warming due to Föhn winds or adiabatic warming of the falling katabatic winds along the coast. Additionally, they report that the correlations found in ice cores are much lower than in the model and discuss reasons for this discrepancy. Overall, this is a very interesting study, which I recommend for publication in The Cryosphere. Before publishing, some major and a number of minor points should be clarified/improved as listed below.

**Major**

(1) Correlation testing The authors present a number of correlation maps. When conducting a high number of significance tests on a gridded map, this increases the number of false rejection of the null hypothesis, (e.g. Wilks 2016). There are correction methods for this issue (e.g. Benjamin and Hochberg, 1995), which the authors should apply in their study.

Benjamini, Y. and Hochberg, Y.: Controlling The False Discovery Rate – A Practical And Powerful Approach To Multiple Testing, J. Roy. Stat. Soc. Ser. B, 57, 289–300, 1995.

Wilks, D. S.: "The Stippling Shows Statistically Significant Grid Points": How Research Results are Routinely Overstated and Overinterpreted, and What to Do about It, B. Am. Meteorol. Soc., 97, 2263–2273, https://doi.org/10.1175/bams-d-15-00267.1, 2016.

**Answer:** Thank you for bringing these papers to our attention. We will test the FDR method to our correlations and modify the text and figures where necessary and describe this method in the manuscript, citing these two papers as references.

(2) Gridding of ice core data The authors describe their method to analyze the ice core data on a grid. They mention that it is important to have enough ice cores per grid cell, however, the information about how many ice core measurements used per gridded value is missing (a figure or table in the supplementary material could do the job). First, they present gridded ice core results at 108 km x 108 km and 216 km x 216 km. Finally, they analyze results where they analyze results for different resolution between 108 km x 108 km and 684 km x 648 km (Fig. 9) where they only take into account grid cells with five or more ice cores. Based on figure 9, this criteria is not fulfilled for 108 km x 108 km. Thus, I wonder how many ice cores are available in the 108 km x 108 km and 216 km x 216 km grid cells. Does the analysis presented for 108 km x 108 km and 216 km x 216 km actually remove the general noise? For how many locations do these grid cells take into account more than one or two ice cores? This needs to be clearly stated as it will have a strong impact on the analysis.

**Answer:** Fig. 8 presents the gridded ice core results at the 216 km x 216 km resolution only (for both the SMB-SAT correlation and the SMB- $\delta^{18}$ O correlation). We only describe the results obtained for the gridding at the 108x108 km resolution in the text, stating that "both the SMB-SAT and the SMB- $\delta^{18}$ O correlations remain at 0.09 and 0.13 respectively". Aggregation at this resolution does not improve our results, which is why we do not show it. We will add "(not shown here)" to that effect to be as clear as possible.

When setting a minimum threshold 1 ice core per grid point, the correlation values are much more noisy (see Fig. 3 at the end of this document; note that if Fig. 3 is added to the supplement, it will of course

modified in the same way that Fig. 9 will be modified). However, if we look at the average calculated for both the individual ice core and the aggregate ice cores ("average of individual" and "average of aggregated", i.e. the dark orange and dark blue dots), the same slightly increasing trend in correlation strength is observed. But of course, there is a lot more "noise" in the individual signals, seen from the widely varying correlation values. In the manuscript, we propose to keep using the same two panels for Fig. 9, but doubling them, with the extra two panels providing information on the number of ice cores per dot. Fig. 9 shows the comparison, for a specific scaling of the RACMO27 grid cells, between the average of the ice core correlations in that augmented grid cell versus the average of the RACMO27 correlation over the augmented grid cell. We propose to, on the model side, also calculate the average in the augmented grid cell of only the original RACMO27 grid cells that contained ice cores (see Fig. 4 for a sketch). We will therefore be able to show a direct comparison between the ice cores and the RACMO grid cell correlations.

(3) Figure 9 Figure 9 needs strong improvement. In the caption it is stated that the size of the colored dots is a function of the number of ice cores aggregated. There is need for a legend giving information about the actual number of ice cores for the different dot sizes. The dots for the model in the legend are far too small (black and gray dots). The dots in the legend could all have the same size.

**Answer:** We agree that this figure is too dense and should be clarified. We propose to make all dots the same size, and to double each panel: each of the existing panels will have a first panel where the dots show the correlation values, and a second panel where the dots show the number of ice cores used per dot. In addition, the dots for the model will be made the same size as that of the ice cores, and we will modify the legend accordingly.

Minor

General:

Avoid "see Figure...", "shown on Fig..." just reference the figures. For the figures add a space after Fig., i.e. Fig.  $1 \rightarrow$  Fig. 1

**Answer: We have applied your suggestions.**

Avoid spoken language, i.e. try to reduce the number of non-informative sentences, e.g. "Let us first focus on the link between wind redistribution and SAT." (P.8, L228) or "To explain this, we have to first describe average conditions." (P.8, L247) or "Now that we have a better understanding from the models let us look at..." (P.10, L.398). These are some examples but there are more in the text.

Answer: We have reworded some of these: P8L228 has become "We will first examine[...]"; P8L247 has become "During average weather conditions in coastal areas,[...]; P10L398 has become "We will next examine in-situ measurements of SMB, SAT and  $\delta^{18}$ O.". As we modify the manuscript in the next phase, we will pay attention to any further sentences such as these.

**P.2, L24:** sea-level rise $\rightarrow$ SLR**

**Answer:** We have opted to explicitly write "sea level rise" every time as it it is the only place in the manuscript that we mention sea level rise.

*P.2, L35:* temperature  $\rightarrow$  heat **Answer:** Replaced.

**P.2, L34-36:** The first sentence leaves the reader wondering where the air comes from. Consider reformulation of the two sentences.

**Answer:** The first two sentences have been reworded as follows: "Large-scale atmospheric circulation (100s of km) strongly controls SMB in Antarctica. This large-scale atmospheric circulation embeds synoptic-scale cyclones that collect heat and moisture from further north, including the Southern Ocean, which they can release onto the AIS[...]"

**P.2, L45:** twice mentioned the increased SAT. Is it a positive feedback, i.e. increasing SAT  $\rightarrow$  increased snowfall  $\rightarrow$  increase in SAT? If yes, consider reformulation, if no remove one of the increased SAT.

**Answer:** We thank the reviewer for noticing this confusing formulation. Also taking reviewer #1's comment, this paragraph has been changed as follows: "In addition, based on the Clausius-Clapeyron relationship, the increasing mid and upper troposphere temperature due to climate change implies that the air can include more moisture before saturation, and therefore snowfall is increased (Frieler et al., 2015). If this predicted increase in SMB is linked to increasing temperatures in the 21st century, it will be interesting to see if SMB and SAT are linked in the past too, since an increase in SAT and tropospheric air temperature are strongly correlated. In this case, SMB records over time will be a helpful tool to constrain past climates."

**P.3, L80:**  $at \rightarrow as$ **Answer:** Changed.

**P.3, L85:** (Dalaiden et al., 2019)  $\rightarrow$  Dalaiden et al. (2019)

**Answer:** Changed. Also, note that the Dalaiden et al. (2019) The Cryosphere Discussions paper has now been published in The Cryosphere and all references have been changed to Dalaiden et al. (2020), here and in the manuscript.

P.3, L87: Here, for the second time you define the local scale. Redefine one

**Answer:** We have chosen to remove the definition of local scale earlier ('*We will refer to this wind-based redistribution as the "local scale"*) and keep it defined here P3L87 as that is spatial scale that we use in the whole manuscript.

*P.4, L91-95:* Add "?" to the questions **Answer:** Added.

*P.4, L107:* "(with the additional fourth made available recently, . . .)" Mention which model is new. **Answer:** This is now mentioned explicitly: "with the additional fourth made available recently, iCESM1, "

*P.4, L108:* Add the reference of Brady et al., 2019 as not all the models are described in Dalaiden et al., 2020.

Answer: Added, as well as Stevenson et al. (2019).

**P4.** L114: "by a sea surface temperatures..."  $\rightarrow$  "by sea surface temperatures" **Answer:** Modified.

**P.5, L120-121:** The authors use a different number of ensemble members for the different iGCMS. By averaging over several ensemble members variability is lost. It would be interesting to know how this affects the results, please discuss.

Answer: We are sorry that the wording of this sentence was confusing. What we did was: (1) we calculated the correlation of the two variables (SMB-SAT or SMB- $\delta^{18}$ O) for every grid point for each model member (3 members for iCESM1 and 7 members for iHadCM3). (2) We calculated the mean of the correlation values obtained per grid point over all the model members that belong to iCESM1 or iHadCM3 to get the mean SMB-SAT and SMB- $\delta^{18}$ O correlation for each model. (3) We interpolated then all four iGCM correlation results (ensemble means for iCESM1 and iHadCM3, ECHAM5-MPI/OM and ECHAM5-wiso correlation results) onto the RACMO27 grid, for both the SMB-SAT annual correlations and the SMB- $\delta^{18}$ O 5-yearly correlations. (4) We then calculated the mean over all four iGCMs to get the resulting iGCM plots shown in Fig.1. The sentence has now been changed to: "As the iCESM1 and iHadCM3 ensembles include three and seven simulations, respectively (each has slightly different initial conditions), we first calculate the correlation of the two variables (SMB-SAT or SMB- $\delta^{18}$ O) for every grid point for each ensemble member (3 for iCESM1 and 7 for iHadCM3). Then we obtain the mean of the correlation values per grid point over each ensemble of simulations for both iCESM1 or iHadCM3. These ensemble means can then be compared to the correlations calculated for ECHAM5-MPI/OM and ECHAM5-wiso."

**P.5, L121:** Consider reformulations "we average over their ensemble of simulations to obtain mean representation of SMB, SAT and  $\delta^{18}O$  for each iGCM."  $\rightarrow$  "we retrieve the ensemble mean of SMB, SAT and  $\delta^{18}O$  for each iGCM."

Answer: Same comment as above.

**P.5**, *L123*: Mention the most important findings of the evaluation. What about the model that is not evaluated in Dalaiden et al., 2019?

**Answer:** Based on the comments of both reviews, we have chosen to re-do the evaluation provided by Dalaiden et al. (2020) for SMB for three of the iGCMs, and add iCESM1 to it, as well as evaluate the SAT and sea-level pressure (SLP) fields. SMB and SAT will be evaluated against RACMO27 and SLP will be evaluated against ERA-Interim (to provide sufficient northerly coverage), over the 1979–2000 AD time interval. We will further discuss the relevance of this evaluation in the manuscript in this section (P5).

**P.5, L126:** significative  $\rightarrow$  significant **Answer:** Changed.

**P.5, L129-130:** Explicitly give the number of years taken into account. Avoid " $\sim$ " as the time periods are clearly defined.

Answer: These have been changed to 130 years and 40 years.

**P.5**, L130/P.5, L141: Please clarify the choice of the time periods. You mention "...a shorter () timescale covering 1961-2000 AD for comparison to the RCM simulations and measured SAT." However, the RCM start in 1979. Why don't you take a time period that matches between the iGCMs and the RCM? I guess this would be 1979-2000?

Answer: The reason for this is that we wanted to keep as many years as possible for correlating the climate variables (SMB, SAT and  $\delta^{18}O$ . RACMO covers 1979-2016 = 38 years). Truncating the iGCM data to start in 1979, would have meant we only had 22 years of data, which is very short to have meaningful correlation results. We thus prefer to have a similar length for all models. This is now specified at the end of this paragraph as: "We choose to use the 1961–2000 AD time interval instead of starting in 1979 AD as for the RCM simulation, so that all model simulations have a similar length (40 years and 38 years for the iGCMs and the RCM, respectively) and their correlation results are meaningful."

In addition, we have compared the SMB-SAT correlations over the 1961–2000 AD and the 1979–2000 AD time intervals and the correlation strengths are unchanged, although the correlations over the shorter 1979–2000 AD time interval lose in significance (more areas with p > 0.1, e.g. see iCESM1, shown at the end of this document in Fig. 5). For the SMB- $\delta^{18}$ O correlations, the 1979–2000 AD interval is very short to calculate a 5-yearly correlation.

**P.5, L132:** Here and in the Figures 1, and S1-S4 you first mention  $\delta^{18}O$ -SMB followed by SMB-SAT. However, in the analysis you first analyze SMB-SAT. Flip the order in both the text (here) and the figures. **Answer:** We agree, and will flip the order in the text here and this will be taken into account when modifying Fig. 1 and 2 as described P9 of this review.

**P.5, L143:** to studying  $\rightarrow$  to study **Answer:** Changed.

*P.5, L147: remove "," before etc.* Answer: Removed.

**P.6, L158:** Reformulate: "Ice cores record variations with depth of ice's  $\delta^{18}O$ ." **Answer:** Reworded as: "Ice cores record variations of the  $\delta^{18}O$  of the ice with depth"

**P.6, L164-168:** How well does this temperature reconstruction method work? Why do you need to rely on this? Could you use SAT from reanalysis instead? Are there weather stations in the close proximity of the ice core locations? For locations where weather stations are available, how well does the SAT reconstruction method agree with the measurements? Could you present correlations of ice cores to station measurements for some locations?

**Answer:** We use this SAT reconstruction as they are based on the only direct observations of temperatures from weather stations over the last 50 years. Prior to 1979, reanalysis data have well known biases (e.g. Bromwich et al., 2007). This SAT reconstruction uses 15 weather stations (many along the coast), based on which they produce estimates of monthly gridded mean temperature anomalies for the whole continent over 1958–2012 AD. They use a kriging technique, originally developed for the reconstruction of Antarctic snowfall (Monaghan et al. 2006), to then interpolate the data between the stations. We will provide a map of the station locations used in the reconstruction the supplementary material, and provide correlations of ice cores to station measurements for some locations where they are in close proximity.

**P.6**, L177: Consider starting the results and discussion section with your results and discuss the comparison to other studies later.

**Answer:** We have moved this sentence that compares our results to those of Dalaiden et al (2020) towards the end of this section 3.1.1, reworded a little as: "Dalaiden et al (2020) have shown that the relationship between SMB and SAT is positive on the continental scale for each of their seven Antarctic regions, whether they used the GCMs or RACMO2.3 simulations. The simple concept that Antarctic precipitation originates mainly from lower latitudes, in the form of warm and wet air masses, explains the co-variance of SMB and SAT at the continental scale as shown by Dalaiden et al. (2020). This simple concept therefore also applies at the regional scale here."

**P.6, L184:** Figures S1 and S2 are about  $\delta^{18}$ O. Don't mention them here. I think only Fig. S3 is relevant to this section. Reorder the figures in the supplementary material and only mention the related ones. **Answer:** Based also on the other review, we will move part of the supplementary figures (SMB-SAT correlation and SMB- $\delta^{18}$ O over the 1871-2000 AD time interval) for each iGCM and the iGCM average to the main manuscript. But we will only refer them in the relevant sections of the manuscript, as suggested here.

P.7, L191: Avoid generalizing, as you only have two different resolutions and two different time periods (which are partially overlapping).

**Answer:** We have modified "*Whatever the temporal and spatial scale*, to: "For the temporal and spatial scales investigated," and moved it further down in the paragraph to link to the Dalaiden et al. (2020) study.

**P.8, L223-224:** How do the findings by Agosta et al. (2019) relate to your findings? Indicate the link more clearly.

**Answer:** We have changed these lines to: "Agosta et al. (2019) also show the strong impact of surface topography (surface curvature in their case) - wind interactions on modelled SMB by examining the spatial link of time-averaged values. They observe that above a certain threshold, winds will affect SMB locally in pattern that matches that of drifting snow fluxes as modeled by RACMO2.3."

*P.8, L225:* "very positive", reformulate to "strongly positive" or "positive". Answer: Changed.

**P.8**, L225: "weaker-to-positive" this needs to be reformulated. What you refer to is a weaker-negative-to-positive correlation. This needs to be specified.

Answer: "weaker-to-positive" has been changed to "weaker (less negative) or positive".

**P.8, L226:** "(outlined here with a magenta line)": Why using a magenta line (Figs. 4 and 5) and not the black dashed lines as in Fig. 2? Magenta lines are harder to see. Remove "here", i.e. "(outlined with a magenta line)".

**Answer:** We chose to plot the lines in magenta because they were difficult to see due to the blue–white-red color scheme of the maps (versus red tones in Fig.2). "Here" is now removed.

**P.8, L231:** "very positive"  $\rightarrow$  "strongly positive" **Answer:** Changed.

**P.8**, L232: To me it looks like the effect is not present over the whole range of the Trans-Antarctic Mountains, i.e. I can't see the effect for the eastern part of the mountain range. Please, clarify.

**Answer:** This is due to the fact the Antarctic Peninsula is very thin and the eastern side if more difficult to see, especially since we do not show the ice shelf values. We will, depending on which is more visual, either increase the size of this figure panel (for MSWD-SAT and MSWD-SMB both), or provide a zoomed-in view of the AP (see Fig. 1 at the end of this document).

**P.8, L236:** "this leeward side"  $\rightarrow$  "the leeward side" **Answer:** Changed.

**P.8, L.239:** "here"  $\rightarrow$  "in this area" **Answer:** Changed.

P.8, L.241: Be specific about the difference between the low level westward winds and upper level winds

**bringing the warm air masses from the north.**

**Answer:** The distinction between low level westward winds and upper level winds bringing heat and moisture from further north will be made in the revised manuscript, and we will use the opportunity to reorganize and clarify the discussion in this section.

**P.8, L.244:** Did you test whether the correlation is significantly less negative? Otherwise, avoid using significantly.

Answer: We have removed "significantly", since we meant it more as "visibly".

**P.9, L.255:** "...become weaker or increase". This is not clear, please clarify. Do you mean "become weaker or positive"?

Answer: Yes, we meant "become weaker or positive", we have now changed it in the text.

**P.9, L.256:** "bottom panel"  $\rightarrow$  use a), b),... for all the figure panels. **Answer:** This will be changed.

**P.9, L.268:** Instead of "marked on Fig. 2" use either "AE on Fig. 2" (or "marked as AE on Fig. 2"). Same for BG (P.9, L274) and AL (P.9, L.277).

Answer: They have all been changed to: 'yy' on Fig.xx.

*P.9, L.271:* "*i.e.*" *instead of* "*so*" **Answer:** Changed.

*P.9, L.272:* Change "very positive" to "strongly positive" **Answer:** Changed.

**P.9**, L.277: Does the model confirm especially high precipitation in the Adélie Land?

**Answer:** Adélie Land is in a coastal regime and therefore show a high snowfall rate, above 500 mm w.e.  $yr^{-1}$  and up to 2000 mm w.e.  $yr^{-1}$ . See Fig 2, top panel. We have modified this paragraph as follows: "For example, Adélie Land ('AL' on Fig. 2) displays high snowfall rates (above 500 mm w.e.  $yr^{-1}$  based on RACMO2.3 results), but is also known for its record-high katabatics (van den Broeke et al., 2002). This region does not display particularly weak MSWD-SMB or MSWD-SAT annual correlations in RACMO2.3 results."

*P.9, L.282: The lines on Fig. 2 are not magenta but black dashed.* Answer: Thank you for spotting this, it has been changed.

**P.10, L.288:** "Perhaps here snowfall input from further north is so high that it dominates the SMB and SAT records." Do the model results confirm this? The models might have the different components of the SMB to check this.

**Answer:** This was a hypothesis but it is difficult to verify it in this case as we would need daily data to look into the local variability. So we prefer to simply remove this sentence and keep the description factual.

**P.10, L.290: RACMO05?* (= *RACMO5?*)**

Answer: Yes, it was a typo and has been corrected everywhere it appeared in the manuscript.

**P.10**, **L.295**: Try to avoid generalization. Consider changing "all scales" to "all investigated scales". **Answer:** We agree and have added "investigated" in that sentence.

**P.10, L.310: Reference figures for iGCMs and RACMO.**

**Answer:** The figures will be referenced accordingly, once Figs. 1 and 2 have been modified as described in an earlier review comment.

**P.11, L.318:** "...an important process, process that..."  $\rightarrow$  "...an important process, i.e. a process that..." or "...an important process, that..."

Answer: We have opted for "...an important process, i.e. a process that...", thank you.

**P.11, L.339:** *instead of "as discussed below" reference Section.* **Answer:** It's in the same section, so perhaps this phrasing is unnecessary. We have removed it.

**P.11, L.342:** You mention that the correlations of the cores with an inhomogeneous distribution over the continent are probably not representing the continental or even regional correlation. Then, you mention that the model shows mainly positive correlations where the ice cores are located. Did you consider calculating the correlation of the model only for grid cells for which ice core data exists? How does the comparison compare in this case? Could this result be used to underline the disagreement?

Answer: We have not compared the correlation values between the ice cores and the model grid cells for which ice core data exist directly. We will make this comparison and add this information to Fig. 9 and the discussion. What Fig. 9 shows is: for each regular grid spacing, we retain grid cells that contain at least 5 or 10 ice cores. The correlation of the SMB and SAT time series of each of these ice cores is represented by the colored dots in the plot. For any grid cell that meets these ice core criteria (thus for any colored dot), we average the correlation values of the matching RACMO27 grid cell as for the ice cores (we calculate the individual mean and the aggregate value). So any colored dot representing ice core-based correlations has a matching gray dot representing model-based correlations (based on the RACMO27 simulation). The correlation strength difference between the ice core-based and model-based correlations is clearly visible. However, following earlier comments on this figure (see second comment of this review), Fig. 9 will be clarified and the text in the revised manuscript adapted accordingly. We could show this using a threshold of 1 ice core as stated before (see second comment of this review and Fig. 3 at the end of this document), but the fewer ice core records are averaged over, the weaker their mean correlation, and the larger the gap between the ice core-based correlations and the model-based correlation (the latter remains more or less constant). See also response to major comment = 2nd comment of this document.

*P.11, L.344: remove "outlined"* Answer: Removed.

*P.12, L.356:* reference the 108x108 km results as "not shown" Answer: We have added "not shown" after "[...] remain at 0.09 and 0.13 respectively".

**P.12, L.366:** "Results are shown in Fig. 9." No need for a whole sentence, just give reference as (Fig. 9). **Answer:** This has been changed to "(only three grid points contain at least five ice cores at the 648 x 648 km grid resolution, Fig. 9)".

P.12, L.368: "all spatial scales": Avoid generalization, i.e. "all investigated spatial scales"

**Answer: Modified.**

P.12, L.375-376: Please clarify this sentence.

**Answer:** We have reworded it to: "Examining the ice core-based results, we note that increasing the number of ice core records that are initially averaged, results in a visible increase of the SMB-SAT annual correlation, but the trend is weak."

**P.21, Fig.7:** significantive  $\rightarrow$  significant **Answer:** Changed.

*P.21, Fig.7: Maybe change "Large dots indicate..." to "Large gray encircled dots indicate..."* **Answer:** This has been changed as suggested to "Large gray encircled dots indicate [...]".

**P.22, Fig.8:** Why don't you show the standard deviation and percentage of p

---

## Author Response (AR1)

**Response the Editor regarding the submitted paper** *Reconciling the surface temperature–surface mass balance relationship in models and ice cores in Antarctica over the last two centuries*

We thank the Editor for their positive comments and constructive suggestions. We have responded to all Editor and reviewer comments, and have modified the paper accordingly. And we would like to thank the Editor and the two anonymous reviewers for their helpful and constructive comments which have strongly improved the quality of this manuscript. Our point-by-point answers follow, as well as the marked up manuscript and supplement.

**Response to RC1 comments on the submitted paper** *Reconciling the surface temperature–surface mass balance relationship in models and ice cores in Antarctica over the last two centuries*

We thank the reviewer for their constructive comments. We have responded to all of them and have modified the paper accordingly. Our point-by-point answers follow.

Please note that review comments are in grey italics while our answers are not. Changes/additions to the original manuscript are indicated in blue.

**Answers to RC1**

*In this article, Cavitte and co-authors analyse the link between surface mass balance (SMB) and surface air temperature (SAT) over the Antarctic ice sheet, at annual resolution. They focus on the last 200 years (1871–2000).*

*They use a series of climate model simulations: four global climate models including water stable isotopes (iGCMs) and a regional climate model (RCM) without isotopes. They also use observation-based results : the temperature reconstruction of Nicolas and Bromwich, 2014 (NB14) and ice-core annual to 5-year-mean $\delta^{18}O$ (Stenni et al., 2017) and SMB (Thomas et al., 2017).*

*SAT is supposed to be recorded by the $\delta^{18}O$ signal of ice cores. But $\delta^{18}O$ and SAT are generally correlated with SMB, as they both result of large scale advection of warm and moist air from lower latitudes. The aim of the authors is to understand how much SMB and SAT are correlated based on climate simulations and ice core records, and what can explain the strength of correlations at the regional and local scale. They also want to understand what is lacking in our current understanding to explain the observed lower correlation of $\delta^{18}O$-SMB in ice cores than SAT-SMB in models.*

*It is a very interesting study that I recommend for publication in The Cryosphere.*

*However, I pointed out major issues that need to be answered before publication.*

**Major (method)**

*This is a minor remark, but important for improving the readability of the article. In the article, the main assumption is that $\delta^{18}O$ is a proxy of SAT. However, you show 'SMB-SAT' and '$\delta^{18}O$-SMB' correlations. I suggest to write it the same order for both, e.g. 'SMB-SAT' and 'SMB-$\delta^{18}O$', even if it has no effect on correlations.*

**Answer:** This has been changed everywhere in the manuscript.

**iGCM members averaging**

**P5L121** *'we average over their ensemble of simulations to obtain a mean representation of SMB, SAT and 18O for each iGCM.' I think this might be a major issue, as when averaging each variable across different simulation, the interannual variability is dampened. The correlation of average is not the average of correlations. It might not change dramatically your results but it should be corrected.*

**Answer:** We are sorry that the wording of this sentence was confusing. What we did was: (1) we calculated the correlation of the two variables (SMB-SAT or SMB-$\delta^{18}O$) for every grid point for each model member (3 members for iCESM1 and 7 members for iHadCM3). (2) We calculated the mean of the correlation values obtained per grid point over all the model members that belong to iCESM1 or iHadCM3 to get the mean SMB-SAT and SMB-$\delta^{18}O$ correlation for each model. (3) We interpolated then all four iGCM correlation results (ensemble means for iCESM1 and iHadCM3, ECHAM5-MPI/OM and ECHAM5-wiso correlation results) onto the RACMO27 grid, for both the SMB-SAT annual correlations and the SMB-$\delta^{18}O$ 5-yearly correlations. (4) We then calculated the mean over all four iGCMs to get the resulting iGCM plots shown in Fig.1. The sentence has now been changed to: "For the iCESM1 and iHadCM3 GCMs ensembles that include three and seven simulations, respectively (each has slightly different initial conditions), we first calculate the correlation of the two variables (SMB-SAT or SMB-$\delta^{18}O$) for every grid point for each ensemble member (3 for iCESM1 and 7 for iHadCM3). Then we obtain the mean of the correlation values per grid point over each ensemble of simulations for both iCESM1 or iHadCM3. These ensemble means can then be compared to the correlations calculated for ECHAM5-MPI/OM and ECHAM5-wiso."

**iGCMs evaluation**

*P5L123* *'Dalaiden et al. (2019) provide an evaluation of the iGCMs used here'. Model evaluation is too weak. You should show in supplementary key evaluations for the 4 models.*
*You should add all Dalaiden et al. 2019's evaluations related to the 3 iGCMs you use in your article, and add evaluation of iCESM1, that is not provided in Dalaiden et al. 2019.*
*As SMB-SAT relationship is driven by atmospheric circulation, it would be good to see how large-scale circulation of these models compare with reanalyses for the satellite era. Showing at least an evaluation of sea level pressure patterns on average for the common period is needed to understand how large the models' biases can be.*

**Answer:** As suggested, we now provide an evaluation of the SMB, SAT and SLP (Sea-Level Pressure) for the 4 iGCMs, described at length in supplement S2. SMB and SAT are evaluated against RACMO27, while SLP is evaluated against ERA-Interim as its grid does extends further north. The evaluation is done over the 1979–2000 AD time period, the longest period of overlap for all the models examined. In addition to the evaluation in S2, we add the following short paragraph page 5 of the manuscript after the description of the models: "Dalaiden et al. (2020) provide an evaluation of first three iGCMs used, and we provide an evaluation of the SMB, SAT and the atmospheric circulation (we use sea-level pressure) for all four iGCMs in Supplement S2. SMB and SAT are evaluated against RACMO27, while sea-level pressure is evaluated against ERA-Interim as the ERA-Interim grid extends further north. The evaluation is done over the 1979–2000 AD time period, the longest period of overlap for all the models examined. We show that all four models produce realistic SMB, SAT and sea-level pressure outputs, with biases likely linked to differences in their physics as well as their spatial resolutions (supplement S2)."

Note that the Dalaiden et al. (2019) The Cryosphere Discussions paper has now been published in The Cryosphere and all references have been changed to Dalaiden et al. (2020), here and in the manuscript.

*Model selection and averaging*
*P5L122* *'all four iGCMs show similar spatial variations of the correlation between SMB-SAT and $\delta^{18}O$-SMB on the continent scale [...] To compare the iGCM continent-wide correlations to the RCM-derived correlations, we interpolate the iGCM results onto the RCM grid and average over all four iGCMs.' From Fig. S1 we can see that ECHAM5-MIP-OM gives significantly different results than the other 3 iGCMS at the ice sheet margins for $\delta^{18}O$-SMB. From Dalaiden et al. 2019 Fig. S7 and S8, we can see that ECHAM5/MPI-OM have a major issue for modelling SMB. It is of major importance for this article, because as you average the 4 isotopic global climate models (iGCMs) correlations into one single map of correlation, it suppose that you consider equal skills of the 4 models. Consequently, you should consider disregarding ECHAM5-MIP-OM simulations, or at least giving it less weight in the average.*

**Answer:** We agree that ECHAM5-MPI/OM differs from the other 3 iGCMs in representing SMB, based on Dalaiden et al. (2020) Fig. S7 and S8 and our new supplementary material S2. Nevertheless, selecting a criteria to keep or remove a model from our analyses may be difficult to justify objectively as all models have strengths and weaknesses. We propose to keep all 4 iGCMs in this study, but we have moved figures S1 and S3 (that display the SMB-SAT and SMB-$\delta^{18}O$ correlations over the 1871–2000 AD time interval) to the main manuscript. We now discuss the similarities and differences between the models more explicitly in the manuscript Section 3.1.1 (P7-8) and in the supplement S2. We show in supplement S2 that although differences are clearly visible for ECHAM5-MPI/OM in terms of SMB representation, its SAT and SLP fields do not seem to differ much more than the other models' compared to the reference models (RACMO27 and ERA-Interim).

*Major (results and interpretation)*
*Consistencies between models and time scales*
*P6L183* *'Moreover, the maximum and minimum correlations obtained are consistent between iGCMs, in*

*magnitude and spatial distribution (see supplementary Fig.S1-S4).'*
*Can you develop the area you think are consistent? Because I see more differences that analogy between the 4 models. ECHAM5-MPI-OM is the only iGCM with a clear loss of correlation at the ice sheet margins in East Antarctica (for both SAT and $\delta^{18}O$). If you exclude ECHAM5-MPI-OM, I see some consistencies between the 3 models left, but still, it's not very clear given the patchy patterns.*

**Answer:** We have now expanded the description of the similarities and differences in this paragraph by defining several regions (specific boxes) over the continent where we observe consistently positive and negative correlations. These areas are shown on Fig. 1 and 2 and include: central West Antarctica, the eastern Antarctic Peninsula, the Plateau, the Amery Embayment and Adélie Land. For each of these five regions, we calculate the average correlation and percentage of significant grid cells for each iGCM and now compare the iGCM correlation results quantitatively in our description. We have modified section 3.1.1 as: "We calculate the annual correlation between SMB and SAT at the regional scale over 1871–2000 AD, the time interval shared by all iGCMs. We obtain a positive annual relationship between SMB and SAT at the regional scale with a continent-wide average value of 0.57 and a spatial standard deviation of ±0.10 (hereafter referred to as ±0.10) over the four iGCMs (the individual iGCMs continent averages range from 0.54-0.60) with a p-value $< p_{FDR}$ for more than 96% of each iGCM's surface area (Fig. 1), and $> 99\%$ on average. We note some spatial differences between models, in particular we focus on five distinct regions of the ice sheet: (1) the AP (also "AP" on Fig. 1) shows weak correlations for iCESM1 and ECHAM5-wiso (0.22-0.24, 57-76% significant grid cells) and positive correlations for ECHAM/MPI-OM and iHadCM3 (0.51-0.60, 100% significant), (2) central West Antarctica ("CWA") which shows the same contrast between iCESM1 and ECHAM5-wiso versus ECHAM/MPI-OM and iHadCM3 (0.31-0.48, 78-91% significant versus 0.63 and 100% significant), (3) the East Antarctic Plateau ("EAP") which shows positive and 100 % significant correlations (0.58-0.70); the Amery Embayment ("AE") and Adelie Land ("AL") which show positive correlations (0.49-0.57 and 0.49-0.69, 100% significant). Areas of weaker SMB-SAT correlations are only observed in iCESM1 and ECHAM5-wiso, which could arise from the models' finer spatial resolution compared to iHadCM3 and ECHAM5-MPI/OM. The SMB-SAT correlation remains positive overall for all models. We obtain the same result if we take non-overlapping 5-year averages of the SMB and SAT variables to calculate their 5-yearly correlation (supplementary Fig. S5)."

*P7L190 'This implies that the correlation of SMB and SAT is similar over the 1871–2000 AD and the 1979–2016 AD time intervals, and from a spatial resolution of >1 down to 5.5 km.' Please clarify what you think is similar between the resolutions and the time periods. I see many differences, so if you want to highlight the consistencies, you should detail them (e.g. high correlations for West AIS?). **Generally, I am concerned about the too optimistic way of presenting consistencies between simulations.***

**Answer:** We agree that we were too vague in our description and now describe the similarities and dissimilarities in more details in that paragraph, the same way that we have done it for the iGCMs (see comment above). We use the same 5 Antarctic regions, shown on the corresponding figure (now Fig. 1, panel f) to quantify the average correlation and the percentage of significant grid cells per region. Section 3.1.1. L224-236 has now been changed to: "We repeat the annual correlation of SAT and SMB using the RACMO27 simulations over 1979–2016 AD (Fig. 1, panel f). At this spatial scale, the annual correlation is also positive in the large majority of the ice sheet with a similar range of correlation values and a continent-wide average value of 0.54±0.22, which is within the range of the iGCM average (0.57±0.10). For the same five distinct regions, we can see that the EAP and AL are also positive and 100% significant. For AP and CWA that had a lower correlation for iCESM1 and ECHAM5-wiso, we see a stronger weakening down to 0.27 with a lower percentage of significant grid cells (48%). The AE area is most distinct with a correlation average of 0.22 and ~30% significant grid cells. The much higher spatial resolution of RACMO27 with respect to the iGCMs is likely behind these differences and the higher percentage of areas with a weaker correlation value, if we compare to what we observed for the iGCMs."

*Wind effect*
**P7L196** *'There are a few areas, spatially consistent between the RACMO27, RACMO5 simulations and the iGCMs, where the SMB-SAT correlation is not as strong.' Again, I am not sure to agree with the authors. If you look at SMB-SAT correlation on Fig.S4 compared to Fig.2, it is not clear that there is a loss of correlation in the iGCMs at the "same areas" than in the RCM. It seems that combining the 4 iGCMs gives by chance the same pattern as in RACMO2?*

**Answer:** We expect that the SMB-SAT correlations won't be identical. We agree with the reviewer that we were too general and optimistic in our description. In light of having changed Figs. 1 and 2 to show the individual iGCM correlation results (see a few comments above), and having defined the 5 regions of fous, we now discuss the similarities and differences in light of these regions specifically between the iGCMs and RACMO2.3 (see comment above). Also, we stress that the increased percentage of non-significant grid cells is likely linked to the finer spatial resolution of RACMO27 with respect to the iGCMs. This is already seen in the different SMB-SAT correlation patterns of iCESM1 and ECHAM5-wiso versus iHadCM3 and ECHAM5-MPI/OM, iCESM1 and ECHAM5-wiso having finest spatial resolution of the iGCMs used here. We have added at the end of the paragraph (P8 L230-232): "The much higher spatial resolution of RACMO27 with respect to the iGCMs is likely behind these differences and the higher percentage of areas with a weaker correlation value, if we compare to what we observed for the iGCMs."

**P7L204** *(1) 'Large-scale air masses, originating over the Southern Ocean and further north, bring warm moist air towards the interior as they flow up-slope, thus inducing a strong and negative correlation': You suggest that at the interannual time scale, when you have upslope winds (so more negative) you have higher temperature and SMB, and the more positive is mean slope in mean wind direction (MSWD) the colder and the dryer. It seems reasonable, but it would be really good to explicit this, because I had trouble trying understanding the positive/negative correlations with wind.*
*(2) Can you show time series of MSWD together with time series of SAT and SMB at some specific locations, so that we can understand what the correlation means? E.g. at locations where it's significantly positively/negatively correlated to SAT/SMB, and on the coast/on the plateau, or at least examples for your cases (1) and (2), and for Adelie Land vs. Amery Embayment.*
*(3) In addition, I am wondering how much annual MSWD is a good indicator of the mechanism you want to highlight. Advection of warm and moist air by cyclones are punctual whereas surface winds generally flow downslope all year long. Consequently I am not sure how much you capture the cyclone activity with MSWD.*
*(4) You discuss this with cases (1) and (2), but it seems very speculative. Basing your statement on time series will help developing more robust analyses. I was lost reading the wind considerations, but after re-reading it a couple of times, I think I agree with most of your conclusions. It would be better to re-write this part before I can give a better feedback.* **I do think it is very interesting to analyse the strong correlations between MSWD, SMB and SAT at the interannual time scale.** *You should use the different components of SMB available in RACMO to analyse the effect of wind on this components (precipitation, drifting snow fluxes, sublimation, etc.), instead of trying to guess why it works this way.*

**Answer:** Point #1 - We agree that the sign of the positive/negative correlations is perhaps unclear, we have now added a sketch (now Fig. 4 in the manuscript) to help with the sign of MSWD with respect to slope and wind direction, for the different scenarios that we describe (synoptic air masses and katabatic

wind interactions). The following sentence is also added just below the description of Eq (1) for clarity: "Since we define a positive MSWD as a wind pointing down-slope, we expect winds coming from the coast up into the interior to have a negative MSWD, and winds flowing from the interior to the coast to have a positive MSWD (Fig. 4, and supplementary Fig. S8)." and (P9-10 L272-279): "Large-scale air masses, originating over the Southern Ocean and further north, bring warm moist air towards the interior as they flow up-slope. At the annual time scale, these up-slope winds (negative MSWD) will therefore bring a higher temperature and SMB (positive anomaly), and thus induce a strong and negative annual MSWD-SAT and MSWD-SMB correlation (Fig. 4, scenario 1a). Similarly, winds flowing down-slope (positive MSWD) will bring drier and colder air (negative anomaly) from the interior to the coast, and will also induce a strong and negative annual correlation of MSWD with SAT and SMB (Fig. 4, scenario 2a). Any area of the AIS that does not show this negative annual correlation between wind and SAT or SMB implies that the changes cannot be interpreted simply in terms of large-scale circulation at the annual scale or descending cold air from the interior (so in that case, scenarios 1b and 2b)."

Point #2 - We have plotted time series of MSWD with time series of SAT and SMB (Fig. 1 at the end of this document). We have chosen, as asked, four locations where MSWD is significantly positively or negatively correlated to SAT/SMB. We show one location on the Amery Embayment (first row of Fig. 1) and one on the eastern side of the Antarctic Peninsula (2nd row of Fig. 1) where MSWD and SAT/SMB are positively/weakly correlated. We show one location in the middle of the plateau region and one in Adélie land where the MSWD-SMB/SAT correlation is strong and negative (third and last rows of Fig. 1). The black cross on the inset maps show the exact position of the location chosen. These plots clearly show that for the negative MSWD-SAT/SMB correlations described in Figs. 5 and 6 of the manuscript (case of Adélie Land and the Plateau), MSWD and SAT/SMB are out of phase. While for the locations where MSWD-SAT is positive and MSWD-SMB is weak, MSWD are clearly in phase with SAT and almost in phase with SMB (case of the Antarctic Peninsula and Amery Embayment locations). However, we do not find that these curves add much information to the information already present in the correlation figures (Figs. 5 and 6 of the manuscript) and so do not include them in our revised submission.

Point #3 - The hypothesis in this paper is that, even if cyclone activity is punctual, by correlating MSWD to SMB/SAT over 1979–2016 AD, we will be capturing the variations of cyclonic activity at interannual timescales, i.e. the variability in the path or intensity of cyclones, and therefore the variability in the sources of heat and moisture. It would be interesting to compare to the SMB and MSWD at the daily timescale, to look at individual cyclonic events more reliably. However, at present, daily RACMO2.3p2 simulations are not freely available, and in any case this could not be done for the ice core records. We have added a short sentence to this effect in the manuscript in this paragraph: "We are aware that such cyclonic activity is punctual in time. However, by correlating MSWD to SMB and SAT over 1979–2016 AD, we make the hypothesis that we capture the first order variations of cyclonic activity at interannual timescales, and therefore the variability in the sources of heat and moisture."

Point #4 - Although not shown, we had looked at the influence of the various SMB components on the total SMB and observed that SMB variability is strongly dominated by precipitation at the annual timescale. Correlating SMB with its precipitation component over 1979–2016 AD gives an average correlation over the whole continent of 0.96. We have now calculated the correlation of MSWD with the various components of SMB (sublimation, snowdrift, snowfall, snowmelt and snowdrift sublimation), however there is no one to one relationship with any of these components. The global effects of the wind are not clear at the annual timescale. It would certainly be insightful to look at their influence at the daily timescale, however, the focus of this study is the annual timescale as we want to be able to compare results with ice core data. We include the results of these correlations at the end of this document (Fig. 2). Note that for the correlation of MSWD and snowmelt, the $p_{FDR}$ value used implies that none of the map is significant (the Bonferroni threshold has to be used for this correlation, Wilkes et al. (2016)).

*Related minor comments:*
*Eq. (1) You must remove the arrow on MSWD as it is a scalar.*
**Answer:** It has been removed.

*Fig.S9 Can you show MSWD too? To see where it's negative and positive?*
**Answer:** Yes, it has been added as Fig. S8.

*P7L215 'We then remove areas of the AIS with a negligible slope (0.001) as in these areas, MSWD will be close to null and will introduce a lot of noise when correlating it with SMB or SAT.': Shouldn't you remove areas where SMB interannual variability and MSWD interannual variability are small too? I would suspect that all the high (negative) correlation for MSWD-SAT and MSWD-SMB found in the EAIS plateau are because of very small interannual variability in wind and SMB(?).*
**Answer:** We argue that areas with a small SMB and MSWD interannual variability are also interesting to look at. We only remove areas that have a negligible slope because correlating such a small number with SMB or SAT could introduce a lot of noise. The results are not radically different with and without the threshold applied on the slope (see Fig. 3). It reduces the average continent-wide correlation strength and increases the number of grid cells with $p > p_{FDR}$, but not significantly. We therefore choose to keep the threshold.

*P8L223 'Agosta et al. (2019) also show a strong link between modelled surface topography (surface curvature in their case) with SMB over the continent when wind speeds exceed 5 m s-1.': but in Agosta et al. 2019, it is a spatial link of time-averaged values, not a temporal link.*
**Answer:** We agree that Agosta et al (2019) clearly look at the spatial link of time-averaged values while we look at the temporal link. We have removed this citation P8 but still cite it later, in our discussion of MSWD and SMB page 11: "Other studies have shown the strong interactions between surface surface topography and winds. For instance, Agosta et al. (2019) examine the spatial link of time-averaged values of surface curvature and surface winds and they observe that above a certain threshold, winds will affect SMB locally in pattern that matches that of drifting snow fluxes as modeled by RACMO2.3.". We now also cite the work of Dattler et al. (2019) in our introduction (page 3) who show that, in their study area, the variability in accumulation is correlated with MSWD variability also. And have added a sentence to that effect page 12 L353-355: "Dattler et al. (2019) also show that, at length scales $< 25$ km, regions of the West Antarctic Ice Sheet show high spatial variability in accumulation simultaneously to high variability in wind speed and direction."

*P8L264 'We therefore expect that the areas regularly under the influence of strong katabatic winds will show a weaker MSWD-SMB correlation due to the episodic but persistent reduction in their SMB through wind scouring'. Here is one out of many examples where you can use RACMO outputs to verify your hypothesis, since you have access to all surface mass balance fluxes, including drifting snow fluxes, in this model.*
**Answer:** See response to a previous comment. However, a little further up page 12 (in the revised version), when we discuss the Amery Embayment and Adélie Land specifically, we refer to the added supplementary Fig. S11 which shows the average sublimation and snowdrift ablation fields for the ice sheet over 1979–2016 AD.

*P10L288 'Perhaps here snowfall input from further north is so high that it dominates the SMB and SAT records.' The same: you can use RACMO outputs to clarify what's happening.*

**Answer:** Same comment as a little further up.

*Suspicion of wrong SMB interannual variability in ice cores*
*Sections 3.2 and 3.3 study the SMB-SAT link in ice core data, and aims at understanding why it is much weaker in ice cores than in models.*
*10L307 'We observe a weak-to-null annual correlation between SAT and SMB in the ice cores (Fig.7) with an average value of 0.09±0.18 over all the ice cores, versus a continent-wide average value of 0.57±0.10 for the iGCMs and 0.54±0.22 for RACMO27.' I am wondering whether the low correlation between ice cores SMB and NB14 SAT is because of a too low or incorrect interannual variability of SMB in ice cores. I think this analysis is very interesting, but I cannot evaluate it further if I am not sure that the difference between ice cores and models is not because of wrong SMB interannual variability in ice cores. Can you show annual time series of ice core SMB vs. RACMO SMB? The results will have a major impact for reviewing the rest of the study.*

**Answer:** We agree with the reviewer that it is difficult to quantify whether the low correlation between ice cores SMB-SAT is because of a too low or incorrect SMB interannual variability of SMB in the ice cores. Ice core SMB errors are difficult to constrain (measurement errors, representativity errors due to processes such as a wind ablation that removes a part of the deposited annual snowfall). In addition, missing one year in the dating of the ice cores, for example, can shift the time series, and then the correlation between ice core-derived SMB and the independent SAT time series would be lost. However, we show (Figs. 7 and 8 of the manuscript) that the correlation between SMB and $\delta^{18}$O, both ice core-derived records, is also low. However, in that case, SMB and $\delta^{18}$O are both measured on the same core. It suggests that the low correlation is perhaps unrelated to a bad synchronization of the SMB and SAT variables. RCMs also include errors that are difficult to trace (difficulty of representing blowing snow or diamond dust). RACMO2.3 for example includes blowing snow processes but Agosta et al. (2019) have shown that its spatial variability is under-represented and that it is underestimated by a factor of 3. When comparing RCM and ice core results, it is thus not simple to assess the origin of the differences in interannual variations. The advantage of RCMs is that they are self-consistent: the SMB and SAT values simulated are linked by the physics of the model. Ice cores on the other hand can have different biases dependent on different variables (ablation, diffusion, measurement errors, location, etc). However, RCMs can only go back as far as we have direct satellite measurements to constrain the reanalyses that drive them at their boundaries (i.e. 1979 as previous to that, measurement biases increase, e.g. Bromwich et al., 2007). We need the ice core observations to go back further in time. To state more clearly why we compare model correlations to the ice core correlations, we have added the following sentence to this effect in the introduction: "Despite these processes that render our interpretation of ice core records difficult, we need the ice core observations to go back in time to validate model reconstructions before the period when we have direct measurements.". We have also added the following in section 3.3: "When comparing model (in particular RCM) and ice core results, it is thus not simple to assess the origin of the differences in interannual variations. The advantage of RCMs is that they are self-consistent: the SMB and SAT values simulated are linked by the physics of the model. However, RCMs also include errors that are difficult to trace (difficulty of representing blowing snow or diamond dust). RACMO2.3 for example includes blowing snow processes but Agosta et al. (2019) have shown that its spatial variability is under-represented and that it is underestimated by a factor of 3. Furthermore, RCMs can only go back in time as far as we have reliable reanalyses to drive them at their boundaries (often stopping around 1979 as previous to that, measurement biases increase, Bromwich et al., 2007)). Ice cores on the other hand can have different biases dependent on different variables (ablation, diffusion, measurement errors, location, etc). However, they allow us to go back further in time than direct observations.".

*Minor comments*
*P2L35 'temperature' : heat*
**Answer:** Replaced.

*P2L35 remove 'usually'*
**Answer:** Removed.

*P2L35-36 'that collect heat and moisture from further north, including the Southern Ocean, which they can release onto the AIS'. It's the same idea and mechanism than described in the previous sentence: 'Large-scale atmospheric circulation strongly controls SMB in Antarctica, bringing air masses with a high moisture and temperature content'. Merge the sentences.*
**Answer:** The two sentences have been merged as: "Large-scale atmospheric circulation (100s of km) strongly controls SMB in Antarctica. This large-scale atmospheric circulation embeds synoptic-scale cyclones that collect heat and moisture from further north, including the Southern Ocean, which they can release onto the AIS (Gorodetskaya et al., 2014; Sodemann and Stohl, 2009; Wang et al., 2019)."

*P2L43-44 'In addition, based on the Clausius-Clapeyron relationship, the increasing surface air temperature (SAT) due to climate change should induce a greater moisture holding capacity of the air': It's the increasing of air temperature (in the mid and upper troposphere) that induces an increase in moisture holding capacity and more precipitation, not the increase in \*surface\* air temperature. The confusion here comes from the fact that increase in SAT and in tropospheric air temperature are strongly correlated.*
**Answer:** We agree and the sentence has been appropriately reworded: "In addition, based on the Clausius-Clapeyron relationship, the increasing mid and upper troposphere temperature due the warmer climate implies that the vapor pressure of the air is higher, and therefore snowfall is increased (Frieler et al., 2015; Fudge et al., 2016). If this predicted increase in SMB is linked to increasing temperatures in the 21st century, it will be interesting to see if SMB and SAT are linked in the past centuries too, since an increase in SAT and tropospheric air temperature are strongly correlated. In which case, SMB records over time will be a helpful tool to constrain past climates (Dalaiden et al., 2020)."

*P2L49 'is often used as a proxy for SAT': Is it used as a proxy of SAT or a proxy of snowfall-weighted SAT?*
**Answer:** We have changed this to "is often used as a proxy of snowfall-weighted SAT (and by extension of SAT, Stenni et al., (2000) [...]".

*P3L84 AP is not introduced before*
**Answer:** We noticed that the first instance that "Antarctic Peninsula appears is P2L42, and have therefore defined the acronym there: "Antarctic Peninsula (AP,...)"

*P3L86 'over historical timescales': give the time period and the resolution.*
**Answer:** We have modified the sentence as follows: "Our work follows from that of Dalaiden et al. (2020), in which they show that SMB and SAT are strongly correlated interannually at the continental scale over millennial (1000–2005 AD) and historical (1850–2005 AD) timescales"

*P4L91 '1. Which processes link SAT and SMB at regional scales and how do they scale down from conclusions at the continental scale': You don't answer to this question.*
**Answer:** We have reformulated the question to: "Do the processes that link SAT and SMB at the continental scale also play a dominant role at the regional scale?". The title of section 3.1 has also been

changed accordingly.

*P4L98-99* *'We hypothesize physical mechanisms that could explain the discrete areas of the AIS where the SMB-SAT relationship is weak.' Here you suppose this relationship is always strong. But in this article, you don't analyse why the SMB-SAT relationship is generally strong.*
**Answer:** This was indeed omitted because P3L86, we briefly summarized Dalaiden et al. (2020)'s conclusions on the SMB-SAT relationship at the continental scale, but we agree that it was too succinct. We propose to not change this paragraph much, but just modified the sentence to: "We hypothesize physical mechanisms that could explain the discrete areas of the AIS where the SMB-SAT relationship is weak, compared to the continent-wide positive SMB-SAT relationship.". However, we provide more discussion in section 3.1.1 where the strength of the SMB-SAT relationship is described and discussed.

P4L98-99 is now: We start with a brief description of the data sets and models used in this study. Then we compare the positive SMB-SAT relationship, as well the SMB-$\delta^{18}$O relationship, obtained in the models at regional scale to the continental-scale results of Dalaiden et al. (2020).

Furthermore, we have re-organized section 3.1.1 a little and discuss our results in light of those of Dalaiden et al (2020) at the end as follow: " Dalaiden et al (2020) have shown that the relationship between SMB and SAT is positive on the continental scale for each of their seven Antarctic regions, whether they used the GCMs or RACMO2.3 simulations. The simple concept that Antarctic precipitation originates mainly from lower latitudes, coming from relatively warm and wet air masses, explains the co-variance of SMB and SAT at the continental scale as shown by Dalaiden et al (2020). This simple concept therefore also applies at the regional scale for interannual variations here. Indeed, for the temporal and spatial scales investigated, the annual correlation between SMB and SAT is positive for a large majority of the model grid points, despite a few regional differences due to the models' differing spatial resolutions."

*P5L122* *'all four iGCMs show similar spatial variations of the correlation between SMB-SAT and $\delta^{18}$O-SMB on the continent scale' : You didn't give on which time scale you correlate the time series. You should emphasis the time step (annual, from the text after) in the method, by always associating "correlation" with "annual".*
**Answer:** We agree and have added "annual" before correlation in this paragraph, and all the relevant places in the manuscript where we are talking specifically about the annual correlation of two variables, and not correlation values in general. And we have added "5-yearly" everywhere we are looking at 5-year averages of the variables correlated.

*P6L172* *'Furthermore, the spatial distribution of the ice cores over the AIS is not homogeneous, with the majority of the ice cores located in the coastal areas, and very few in the interior (see supplementary Fig.S5). This certainly introduces a spatial bias in our ice core-based correlation towards coastal signals and processes.'*
*The lower the accumulation, the lower the ice core resolution. So it is expected that annually resolved ice cores will be at the ice sheet margins. In addition, you should not write "coastal", as most of the ice cores are not at the coast but inland. You can maybe divide Antarctica in "low elevation" (<2200 m) and "high elevation" (>2200 m).*
**Answer:** We agree with both comments, and have reformulated the paragraph as: "Furthermore, the spatial distribution of the ice cores over the AIS is not homogeneous. If we divide Antarctica into low elevation areas (i.e. <2200 m) and high elevation areas (i.e. >2200 m), the majority of the ice cores are located in the low elevation areas, and very few in high elevation areas (except for the large number of ice cores all located in DML, see supplementary Fig. S4). This certainly introduces a spatial bias in our

ice core-based annual correlation towards coastal signals and processes."
We have added the contour line = 2200 m of altitude on Fig. S4.

*P7L185 'We repeat the annual correlation of SAT and SMB using the RACMO27 simulations over 1979–2016 AD (see Fig.2).': Merge Fig. 2 to Fig. 1 so that we can compare patterns between iGCMs and RACMO2.*
**Answer:** These have been merged (now Fig. 1). Following earlier comments, Fig. 1 now shows the annual SMB-SAT correlation for each of the 4 iGCMs (previously supplementary Fig. S3), the iGCM average (previously Fig. 1, panel c) and the RACMO27 SMB-SAT correlation (previously Fig. 2). And Fig. 2 now shows the 5-yearly SMB-$\delta^{18}$O correlation for each of the 4 iGCMs (previously Fig. S1) and the iGCM average (previously Fig. 1, panel a).

*P7L200 'Winds are known to affect SMB and SAT locally, through wind-based redistribution of SMB, turbulent warming from katabatics and Foehn warming effects on leeward slopes.': you should specify that the loss of correlation because of wind-based redistribution of SMB can be modelled by RACMO only, because it is the only model in this study that includes drifting snow modelling.*
**Answer:** We agree and have modified the text adapting your input as follows: "[...] turbulent warming from katabatics and foehn warming effects on leeward slopes. Additionally, the impact of wind on SMB and SAT due to wind-based redistribution can modify the link between the two variables but this can only be resolved by the RACMO2.3 simulations, because it is the only model in this study that includes drifting snow (although according to Agosta et al. (2019), drifting snow is strongly underestimated). To evaluate whether the lack of correlation between SAT and SMB is due to any of these wind effects, we correlate [...]"

*P7L200 'turbulent warming from katabatics and Foehn warming effects on leeward slopes': katabatics and surface winds in general (e.g. pressure gradient winds superimposed with katabatics) are also directly concerned by adiabatic warming, the same process involved in the foehn warming.*
**Answer:** We agree that in both cases, there is a warming effect, so we have modified this line as follows: "turbulent and adiabatic warming from katabatic winds and foehn effects on leeward slopes"

*P11L347 ' Turner et al. (2019) show that more than 70% of the annual accumulation consists of extreme events that have a very short duration (one or more consecutive days).': warning. Turner et al. (2019) show that more than 70% **of the variance** of the annual precipitation is explained by extreme precipitation events, meaning the interannual variability, not the mean value.*
**Answer:** We have modified the text accordingly: "Turner et al. (2019) show that more than 70% of the variance of the annual precipitation is explained by extreme events that have a very short duration (one or more consecutive days)."

*P14L427 'to improve our confidence in using SMB as a direct proxy for SAT over the entire AIS.' You never mentioned this objective before. Can you develop it in the introduction?*
**Answer:** This was mentioned briefly in the introduction already P2L46 "If this predicted increase in SMB is linked to increasing SAT in the 21st century, it will be interesting to see if SMB and SAT are linked in the past too, in which case SMB records over time will be a helpful tool to constrain past climates." but perhaps the goal "to improve our confidence in using SMB as a direct proxy for SAT over the entire AIS." should be stated more explicitly. We have added this more explicitly at the end of our introduction section, P4L97: "Answering these questions will help constrain our confidence in using SMB as a direct proxy for SAT over the entire AIS. With relatively few in-situ observations, additional SAT

proxies would be extremely beneficial for Antarctic climate reconstructions."

[Figure]

Figure 1: Timeseries of MSWD and SAT for the left panels, timeseries of MSWD and SMB for the right panels. These are provided for two locations that have weak-to-positive MSWD-SAT and MSWD-SMB correlations (top four panels) and two locations that have strongly negative MSWD-SAT and MSWD-SMB correlations (bottom four panels). Locations are indicated on the insets of each panel.

[Figure]

Figure 2: Annual correlation between MSWD and the components of SMB: (top left) MSWD-snowfall, (top right) MSWD-snowmelt, (middle left) MSWD-snowdrift, (middle right) MSWD-sublimation from snowdrift, (bottom) MSWD-sublimation, over the 1979–2016 AD time interval. Statistically insignificant areas (p ¿ $p_{FDR}$, the threshold p-value calculated) for each resolution are hashed in grey. The histogram displays the distribution of correlation values for each panel. Region-wide annual correlation mean, standard deviation and percentage of model area with p ¡ $p_{FDR}$ are provided on each panel.

[Figure]

Figure 3: Annual correlation between (top) MSWD and SAT, (bottom) MSWD and SMB using RACMO27 simulations over 1979–2016 AD. Left panels show the results with the >0.1% slope threshold, right panels show the results without the slope threshold. Statistically insignificant areas (p ¿ $p_{FDR}$, the threshold p-value calculated) for each resolution are hashed in grey. The histogram displays the distribution of correlation values for each panel. Region-wide annual correlation mean, standard deviation and percentage of model area with p ¡ $p_{FDR}$ are provided on each panel.

**Response to RC2 comments on the submitted paper** *Reconciling the surface temperature–surface mass balance relationship in models and ice cores in Antarctica over the last two centuries*

We thank the reviewer for their constructive comments. We have responded to all of them and have modified the paper accordingly. Our point-by-point answers follow.

Please note that review comments are in grey italics while our answers are not. Changes/additions to the original manuscript are indicated in blue.

**Answers to RC2**

*In this article the authors investigate the relationship between the surface mass balance (SMB), surface air temperature (SAT) and d18O in models and ice cores. They analyze output of four GCM models and one regional climate model of which the GCMs are aware of isotopes. The dataset of ice cores consists of about 50 ice cores. They report a strong positive correlation between modelled SMB and SAT as well as d18O and SMB, which is in agreement with previous studies. However, there are regions where the relationships break for which the authors investigate reasons. Mainly they find a dependence on the prevailing winds that can strongly affect the relationship by e.g. leeside warming due to Föhn winds or adiabatic warming of the falling katabatic winds along the coast. Additionally, they report that the correlations found in ice cores are much lower than in the model and discuss reasons for this discrepancy. Overall, this is a very interesting study, which I recommend for publication in The Cryosphere. Before publishing, some major and a number of minor points should be clarified/improved as listed below.*

*Major*

*(1) Correlation testing The authors present a number of correlation maps. When conducting a high number of significance tests on a gridded map, this increases the number of false rejection of the null hypothesis, (e.g. Wilks 2016). There are correction methods for this issue (e.g. Benjamin and Hochberg, 1995), which the authors should apply in their study.*
*Benjamini, Y. and Hochberg, Y.: Controlling The False Discovery Rate – A Practical And Powerful Approach To Multiple Testing, J. Roy. Stat. Soc. Ser. B, 57, 289–300, 1995.*
*Wilks, D. S.: "The Stippling Shows Statistically Significant Grid Points": How Research Results are Routinely Overstated and Overinterpreted, and What to Do about It, B. Am. Meteorol. Soc., 97, 2263–2273, https://doi.org/10.1175/bams-d-15-00267.1, 2016.*

**Answer:** Thank you for bringing these papers to our attention. We have now applied the FDR correction to the correlations calculated for our gridded and modified the text and figures as necessary. We describe the FDR correction in the methods' section pages 4-5, citing these two papers as references. Even choosing a control $\alpha_{FDR}$ value of 0.05 (stricter than our previous p-value of 0.1), we show that our results are robust and our conclusions are not changed significantly.

*(2) Gridding of ice core data The authors describe their method to analyze the ice core data on a grid. They mention that it is important to have enough ice cores per grid cell, however, the information about how many ice core measurements used per gridded value is missing (a figure or table in the supplementary material could do the job). First, they present gridded ice core results at 108 km x 108 km and 216 km x 216 km. Finally, they analyze results where they analyze results for different resolution between 108 km x 108 km and 684 km x 648 km (Fig. 9) where they only take into account grid cells with five or more ice cores. Based on figure 9, this criteria is not fulfilled for 108 km x 108 km. Thus, I wonder how many ice cores are available in the 108 km x 108 km and 216 km x 216 km grid cells. Does the analysis presented for 108 km x 108 km and 216 km x 216 km actually remove the general noise? For how many locations do these grid cells take into account more than one or two ice cores? This needs to be clearly stated as it will have a strong impact on the analysis.*

**Answer:** Fig. 8 presents the gridded ice core results at the 216 km x 216 km resolution only (for both the SMB-SAT correlation and the SMB-$\delta^{18}$O correlation). We only describe the results obtained for the gridding at the 108x108 km resolution in the text, stating that "both the SMB-SAT and the SMB-$\delta^{18}$O correlations remain at 0.09 and 0.13 respectively". Aggregation at this resolution does not improve our results, as too few series are averaged, which is why we do not show it. We have added "(not shown)" to that effect to be as clear as possible. And we specify now that Fig. 9 in the manuscript (now Fig. 10)

shows a scaling of 8 grid points (216 x 216 km grid).

When setting a minimum threshold of one ice core per box, it implies that we can retain cases that only use one ice core record, which implies no averaging has been effectively applied. The resulting correlation values are much more noisy (see the very large range of correlation values indicated by the grey bars on Fig. 1, top panel, at the end of this document). The threshold of 5 ice cores is one that allows us to retain several grid points for the scaling based on the RACMO27 grid. A table has been added, as suggested, in the supplement (Table S1) which provides the number of ice cores used per scaling of the RACMO27 grid. Each scaling of the RACMO27 grid retains at least two grid points that match the $> 5$ ice cores threshold. We tested a higher threshold than 5 ice cores, but there are too few ice core records, the results are not meaningful. Fig. 9 (now Fig. 10) of the main manuscript has been simplified as shown in Fig. 7 at the end of this document.

(Answering also to a later comment: "Did you consider calculating the correlation of the model only for grid cells for which ice core data exists? How does the comparison compare in this case? Could this result be used to underline the disagreement?") Fig. 10 shows the comparison, for a specific scaling of the RACMO27 grid cells, between the average of the ice core correlations in that grid box including n by n model grid cells, versus the average of the RACMO27 correlations over the same grid boxes. Until now, the up-scaling of the model correlations was done using model values over the whole n by n grid cells (Fig. 6, panel a). We now calculate, for the model results, correlations averages in the n by n model grid cells using only the RACMO27 grid points where ice cores exist (Fig. 6, panel b). The conclusions are unchanged (a much higher correlation for RACMO27 versus the ice cores in both cases). However, we modify Fig. 10 in the manuscript to reflect this change, as well as its legend. And we have modified the text accordingly p15 L471-475: "For comparison, we calculate the individual mean and the aggregate value of the SMB-SAT annual correlation for the RACMO27-simulated SMB and SAT fields, only retaining the RACMO27 grid points where ice cores exist (also shown on Fig. 10 in green). This to compare the model-derived and ice core-derived annual correlations locally. As for the ice cores, the aggregate value is always higher than the individual mean for the RACMO27-derived SMB-SAT annual correlation, but the difference is small (up to a 0.1 correlation increase)."

*(3) Figure 9 Figure 9 needs strong improvement. In the caption it is stated that the size of the colored dots is a function of the number of ice cores aggregated. There is need for a legend giving information about the actual number of ice cores for the different dot sizes. The dots for the model in the legend are far too small (black and gray dots). The dots in the legend could all have the same size.*

**Answer:** We agree that this figure is too dense and needs to be simplified (now Fig. 10 in the manuscript). The dots are now all the same size, and the information about how many ice cores are used for each dot is now given in Table S1 of the supplementary material. We have decided, for clarity, to remove the individual grid points that were shown in light colors, and show their max and min correlation values as a range on the plot (shown by the grey bars). Only the averages are shown by the colored dots (which used to be the dark dots). The same simplification has been applied to model results.

*Minor*
*General:*
*Avoid "see Figure...", "shown on Fig..." just reference the figures. For the figures add a space after Fig., i.e. Fig.1 $\rightarrow$ Fig. 1*

**Answer:** We have applied your suggestions.

*Avoid spoken language, i.e. try to reduce the number of non-informative sentences, e.g. "Let us first focus on the link between wind redistribution and SAT." (P.8, L228) or "To explain this, we have to first*

*describe average conditions." (P.8, L247) or "Now that we have a better understanding from the models let us look at..." (P.10, L.398). These are some examples but there are more in the text.*

**Answer:** We have reworded some of these: P8L228 has become "We will first examine[...]"; P8L247 has become "During average weather conditions in coastal areas,[...]; P10L398 has become "We will next examine in-situ measurements of SMB, SAT and $\delta^{18}$O.". We have not found any other obvious examples of this.

*P.2, L24: sea-level rise $\rightarrow$ SLR*

**Answer:** We have opted to explicitly write "sea level rise" every time as it it is the only place in the manuscript that we mention sea level rise.

*P.2, L35: temperature $\rightarrow$ heat*

**Answer:** Replaced.

*P.2, L34-36: The first sentence leaves the reader wondering where the air comes from. Consider reformulation of the two sentences.*

**Answer:** The first two sentences have been reworded as follows: "Large-scale atmospheric circulation (100s of km) strongly controls SMB in Antarctica. This large-scale atmospheric circulation embeds synoptic-scale cyclones that collect heat and moisture from further north, including the Southern Ocean, which they can release onto the AIS [...]"

*P.2, L45: twice mentioned the increased SAT. Is it a positive feedback, i.e. increasing SAT $\rightarrow$ increased snowfall $\rightarrow$ increase in SAT? If yes, consider reformulation, if no remove one of the increased SAT.*

**Answer:** We thank the reviewer for noticing this confusing formulation. Also taking the other reviewer's comment into consideration, this paragraph has been changed as follows: "In addition, based on the Clausius-Clapeyron relationship, the increasing mid and upper troposphere temperature due to climate change implies that the vapor pressure of the air is higher, and therefore snowfall is increased (Frieler et al., 2015). If this predicted increase in SMB is linked to increasing temperatures in the 21st century, it will be interesting to see if SMB and SAT are linked in the past too, since an increase in SAT and tropospheric air temperature are strongly correlated. In this case, SMB records over time will be a helpful tool to constrain past climates."

*P.3, L80: at $\rightarrow$ as*

**Answer:** Changed.

*P.3, L85: (Dalaiden et al., 2019) $\rightarrow$ Dalaiden et al. (2019)*

**Answer:** Changed. Also, note that the Dalaiden et al. (2019) The Cryosphere Discussions paper has now been published in The Cryosphere and all references have been changed to Dalaiden et al. (2020), here and in the manuscript.

*P.3, L87: Here, for the second time you define the local scale. Redefine one*

**Answer:** We have chosen to remove the definition of local scale earlier ('*We will refer to this wind-based redistribution as the "local scale"*') and keep it defined here P3L87 as that is spatial scale that we use in the whole manuscript.

*P.4, L91-95: Add "?" to the questions*

**Answer:** Added.

*P.4, L107:* *"(with the additional fourth made available recently, . . .)" Mention which model is new.*
**Answer:** This is now mentioned explicitly: "with the additional fourth made available recently, iCESM1,
"

*P.4, L108:* *Add the reference of Brady et al., 2019 as not all the models are described in Dalaiden et al., 2020.*
**Answer:** Added, as well as Stevenson et al. (2019).

*P4. L114:* *"by a sea surface temperatures..." → "by sea surface temperatures"*
**Answer:** Modified.

*P.5, L120-121:* *The authors use a different number of ensemble members for the different iGCMS. By averaging over several ensemble members variability is lost. It would be interesting to know how this affects the results, please discuss.*
**Answer:** We are sorry that the wording of this sentence was confusing. What we did was: (1) we calculated the correlation of the two variables (SMB-SAT or SMB-$\delta^{18}$O) for every grid point for each model member (3 members for iCESM1 and 7 members for iHadCM3). (2) We calculated the mean of the correlation values obtained per grid point over all the model members that belong to iCESM1 or iHadCM3 to get the mean SMB-SAT and SMB-$\delta^{18}$O correlation for each model. (3) We interpolated then all four iGCM correlation results (ensemble means for iCESM1 and iHadCM3, ECHAM5-MPI/OM and ECHAM5-wiso correlation results) onto the RACMO27 grid, for both the SMB-SAT annual correlations and the SMB-$\delta^{18}$O 5-yearly correlations. (4) We then calculated the mean over all four iGCMs to get the resulting iGCM plots shown in Fig.1. The sentence has now been changed to: "For the iCESM1 and iHadCM3 GCMs ensembles that include three and seven simulations, respectively (each has slightly different initial conditions), we first calculate the correlation of the two variables (SMB-SAT or SMB-$\delta^{18}$O) for every grid point for each ensemble member (3 for iCESM1 and 7 for iHadCM3). Then we obtain the mean of the correlation values per grid point over each ensemble of simulations for both iCESM1 or iHadCM3. These ensemble means can then be compared to the correlations calculated for ECHAM5-MPI/OM and ECHAM5-wiso."

*P.5, L121:* *Consider reformulations "we average over their ensemble of simulations to obtain mean representation of SMB, SAT and $\delta^{18}$O for each iGCM." → "we retrieve the ensemble mean of SMB, SAT and $\delta^{18}$O for each iGCM."*
**Answer:** Same comment as above.

*P.5, L123:* *Mention the most important findings of the evaluation. What about the model that is not evaluated in Dalaiden et al., 2019?*
**Answer:** Based on the comments of both reviews, we have chosen to provide an evaluation for SMB for all four iGCMs, as well as of the SAT and sea-level pressure (SLP) fields. SMB and SAT are evaluated against RACMO27 and SLP is evaluated against ERA-Interim (to provide sufficient northerly coverage), over the 1979–2000 AD time interval (an interval common to all models). The evaluation is described at length in the supplement (S2). In addition to the evaluation in S2, we add the following short paragraph P5 after the description of the models: "Dalaiden et al. (2020) provide an evaluation of first three iGCMs used, and we provide an evaluation of the SMB, SAT and the atmospheric circulation (we use sea-level pressure) for all four iGCMs in Supplement S2. SMB and SAT are evaluated against RACMO27, while sea-level pressure is evaluated against ERA-Interim as the ERA-Interim grid extends further north. The

evaluation is done over the 1979–2000 AD time period, the longest period of overlap for all the models examined. We show that all four models produce realistic SMB, SAT and sea-level pressure outputs, with biases likely linked to differences in their physics as well as their spatial resolutions."

***P.5, L126:*** *significative → significant*
**Answer:** Changed.

***P.5, L129-130:*** *Explicitly give the number of years taken into account. Avoid "∼" as the time periods are clearly defined.*
**Answer:** These have been changed to 130 years and 40 years.

***P.5, L130/P.5, L141:*** *Please clarify the choice of the time periods. You mention "...a shorter () timescale covering 1961-2000 AD for comparison to the RCM simulations and measured SAT." However, the RCM start in 1979. Why don't you take a time period that matches between the iGCMs and the RCM? I guess this would be 1979-2000?*
**Answer:** The reason for this is that we wanted to keep as many years as possible for correlating the climate variables (SMB, SAT and $\delta^{18}$O). RACMO covers 1979-2016 = 38 years. Truncating the iGCM data to start in 1979, would have meant we only had 22 years of data, which is very short to have meaningful correlation results. We thus prefer to have a similar length for all models. This is now specified at the end of this paragraph as: "We choose to use the 1961–2000 AD time interval instead of starting in 1979 AD as for the RCM simulation, so that all model simulations have a similar length (40 years and 38 years for the iGCMs and the RCM, respectively) and their correlation results are meaningful." In addition, we have compared the SMB-SAT correlations over the 1961–2000 AD and the 1979–2000 AD time intervals and the correlation strengths are unchanged, although the correlations over the shorter 1979–2000 AD time interval lose in significance (more areas with p > $p_{FDR}$, e.g. see iCESM1 in Fig. 2 at the end of this document). For the SMB-$\delta^{18}$O correlations, the 1979–2000 AD interval is too short to calculate a 5-yearly correlation.

***P.5, L132:*** *Here and in the Figures 1, and S1-S4 you first mention $\delta^{18}$O-SMB followed by SMB-SAT. However, in the analysis you first analyze SMB-SAT. Flip the order in both the text (here) and the figures.*
**Answer:** We agree, and have flipped the order in the text here and this has been taken into account in modifying Figs. 1 and 2 of the manuscript as described later in this review.

***P.5, L143:*** *to studying → to study*
**Answer:** Changed.

***P.5, L147:*** *remove "," before etc.*
**Answer:** Removed.

***P.6, L158:*** *Reformulate: "Ice cores record variations with depth of ice's $\delta^{18}$O."*
**Answer:** Reworded as: "Ice cores record variations of the $\delta^{18}$O of the ice with depth".

***P.6, L164-168:*** *How well does this temperature reconstruction method work? Why do you need to rely on this? Could you use SAT from reanalysis instead? Are there weather stations in the close proximity of the ice core locations? For locations where weather stations are available, how well does the SAT reconstruction method agree with the measurements? Could you present correlations of ice cores to station measurements for some locations?*

**Answer:** We use this SAT reconstruction as it is based on the only direct observations of temperatures from weather stations over the last 50 years. Prior to 1979, reanalysis data have well known biases (e.g. Bromwich et al., 2007). This SAT reconstruction uses 15 weather stations (many along the coast), based on which they produce estimates of monthly gridded mean temperature anomalies for the whole continent over the 1958–2012 AD time period. They use a kriging technique, originally developed for the reconstruction of Antarctic snowfall (Monaghan et al. 2006), to then interpolate the data between the stations. We now provide a map of the station locations used in the reconstruction in the supplementary material (Fig. S4, panel c). And we have added a more detailed description of this SAT dataset in section 2.3: "Because ice cores do not provide a direct measurement of SAT, we rely on the Nicolas and Bromwich (2014) SAT reconstruction. Nicolas and Bromwich (2014) use SAT records from 15 weather stations around the AIS (mostly coastal, Fig. S4c) to produce estimates of monthly gridded mean temperature anomalies for the whole continent over 1958–2012 AD. The reconstruction is based on a kriging technique, originally developed for the reconstruction of Antarctic snowfall (Monaghan et al., 2006), to interpolate the data between the stations. The interpolated SAT field allows us to correlate SAT and SMB directly at the ice core locations so as to compare to the RACMO2.3 results."

At the end of this document, we show the correlation of three ice core $\delta^{18}$O records with the weather station SAT records used in the Nicolas and Bromwich (2014) reconstruction. We choose these three sites for their proximity to a weather station (Fig. 3 at the end of this document), over the 1960–2010 AD time interval. We can see that for the three locations randomly selected, the $\delta^{18}$O-SAT correlation is positive, as expected, since $\delta^{18}$O is a proxy for snowfall-weighted SAT. However, although positive, the SAT-$\delta^{18}$O correlation is not necessarily as robust everywhere on the continent. This agrees with e.g. Klein et al. (2019) who show that the 5-yearly SAT-$\delta^{18}$O correlation can already be quite varied over the much longer 0–2000 AD time interval. Reference: Klein, F., Abram, N. J., Curran, M. A. J., Goosse, H., Goursaud, S., Masson-Delmotte, V., Moy, A., Neukom, R., Orsi, A., Sjolte, J., Steiger, N., Stenni, B., and Werner, M.: Assessing the robustness of Antarctic temperature reconstructions over the past 2 millennia using pseudoproxy and data assimilation experiments, Clim. Past, 15, 661–684, https://doi.org/10.5194/cp-15-661-2019, 2019.

*P.6, L177:* *Consider starting the results and discussion section with your results and discuss the comparison to other studies later.*

**Answer:** We have moved this sentence that compares our results to those of Dalaiden et al (2020) towards the end of this section 3.1.1, reworded as: "Dalaiden et al (2020) have shown that the relationship between SMB and SAT is positive on the continental scale for each of their seven Antarctic regions, whether they used the GCMs or RACMO2.3 simulations. The simple concept that Antarctic precipitation originates mainly from lower latitudes, coming from relatively warm and wet air masses, explains the co-variance of SMB and SAT at the continental scale as shown by Dalaiden et al (2020). This simple concept therefore also applies at the regional scale for interannual variations here. Indeed, for the temporal and spatial scales investigated, the annual correlation between SMB and SAT is positive for a large majority of the model grid points, despite a few regional differences due to the models' differing spatial resolutions."

*P.6, L184:* *Figures S1 and S2 are about $\delta^{18}$O. Don't mention them here. I think only Fig. S3 is relevant to this section. Reorder the figures in the supplementary material and only mention the related ones.*

**Answer:** We have now moved part of the supplementary figures to the main manuscript. Fig. 1 now shows the annual SMB-SAT correlation for each of the 4 iGCMs (previously supplementary Fig. S3), the iGCM average (previously Fig. 1, panel c) and the RACMO27 SMB-SAT correlation (previously Fig. 2). And Fig. 2 now shows the 5-yearly SMB-$\delta^{18}$O correlation for each of the 4 iGCMs (previously Fig.

S1) and the iGCM average (previously Fig. 1, panel a). The other correlations (different time resolutions and time intervals) are kept in the supplement as appropriate.

P.7, L191: Avoid generalizing, as you only have two different resolutions and two different time periods (which are partially overlapping).
**Answer:** We have modified *"Whatever the temporal and spatial scale,* to: *"For the temporal and spatial scales investigated,"* and moved it further down in the paragraph to link to the Dalaiden et al. (2020) study.

*P.8, L223-224: How do the findings by Agosta et al. (2019) relate to your findings? Indicate the link more clearly.*
**Answer:** We now discuss Agosta et al. (2019) results further down in this section (p12 L352-354): : "For instance, Agosta et al. (2019) examine the spatial link of time-averaged values of surface curvature and surface winds and they observe that above a certain threshold, winds will affect SMB locally in pattern that matches that of drifting snow fluxes as modeled by RACMO2.3." We also reference the Dattler et al. (2019) paper which discusses similar findings: "Dattler et al (2019) also show that, at length scales < 25 km, regions of the West Antarctic Ice Sheet show high spatial variability in accumulation simultaneously to high variability in wind speed and direction."

*P.8, L225: "very positive", reformulate to "strongly positive" or "positive".*
**Answer:** Changed.

*P.8, L225: "weaker-to-positive" this needs to be reformulated. What you refer to is a weaker-negative-to-positive correlation. This needs to be specified.*
**Answer:** For clarity, we have changed the sentence to: "However, a number of areas show a strongly positive MSWD-SAT annual correlation, simultaneously with a weaker (less negative) or positive MSWD-SMB annual correlation."

*P.8, L226: "(outlined here with a magenta line)": Why using a magenta line (Figs. 4 and 5) and not the black dashed lines as in Fig. 2? Magenta lines are harder to see. Remove "here", i.e. "(outlined with a magenta line)".*
**Answer:** We chose to plot the lines in magenta because they were difficult to see in black due to the blue–white-red color scheme of the maps (while Fig. 2 was only in red tones). "Here" is now removed.

*P.8, L231: "very positive" → "strongly positive"*
**Answer:** Changed.

*P.8, L232: To me it looks like the effect is not present over the whole range of the Trans-Antarctic Mountains, i.e. I can't see the effect for the eastern part of the mountain range. Please, clarify.*
**Answer:** We believe this might be because the Antarctic Peninsula is very thin and the eastern side if more difficult to see at this scale, especially since we do not show the ice shelf values. We have added a zoomed-in view of the correlations (both MSWD-SAT and MSWD-SMB) for the AP in supplementary Fig. S9.

*P.8, L236: "this leeward side" → "the leeward side"*
**Answer:** Changed.

***P.8, L.239:*** *"here" → "in this area"*
**Answer:** Changed.

***P.8, L.241:*** *Be specific about the difference between the low level westward winds and upper level winds bringing the warm air masses from the north.*
**Answer:** We now explicitly specify the following page 10: "Berkner Island (BI on Fig. 1, panel f) shows a very distinctive negative MSWD-SAT annual correlation on its eastern side and positive on its western side, which matches the dominant westward wind direction in this area in the lower level of the atmosphere"

***P.8, L.244:*** *Did you test whether the correlation is significantly less negative? Otherwise, avoid using significantly.*
**Answer:** We have removed "significantly", since we meant it more here as "visibly".

***P.9, L.255:*** *"...become weaker or increase". This is not clear, please clarify. Do you mean "become weaker or positive"?*
**Answer:** Yes, we meant "become weaker or positive", we have now changed it in the text.

***P.9, L.256:*** *"bottom panel" → use a), b),... for all the figure panels.*
**Answer:** This has been changed everywhere in the manuscript and supplement.

***P.9, L.268:*** *Instead of "marked on Fig. 2" use either "AE on Fig. 2" (or "marked as AE on Fig. 2"). Same for BG (P.9, L274) and AL (P.9, L.277).*
**Answer:** They have all been changed to e.g.: 'AE' on Fig. 2.

***P.9, L.271:*** *"i.e." instead of "so"*
**Answer:** Changed.

***P.9, L.272:*** *Change "very positive" to "strongly positive"*
**Answer:** Changed.

***P.9, L.277:*** *Does the model confirm especially high precipitation in the Adélie Land?*
**Answer:** Adélie Land is in a coastal regime and therefore show a high snowfall rate, above 500 mm w.e. yr$^{-1}$ and up to 2000 mm w.e. yr$^{-1}$. See Fig 5, top right panel. We have modified this paragraph as follows: "For example, Adélie Land ('AL' on Fig. 2) displays high snowfall rates (above 500 mm w.e. yr−1 based on RACMO2.3 results), but is also known for its record-high katabatics (van den Broeke et al., 2002). This region does not display particularly weak MSWD-SMB or MSWD-SAT annual correlations in RACMO2.3 results."

***P.9, L.282:*** *The lines on Fig. 2 are not magenta but black dashed.*
**Answer:** Thank you for spotting this, it has been changed.

***P.10, L.288:*** *"Perhaps here snowfall input from further north is so high that it dominates the SMB and SAT records." Do the model results confirm this? The models might have the different components of the SMB to check this.*
**Answer:** This was a hypothesis but it is difficult to verify it in this case as we would need daily data to look into the local variability. So we prefer to simply remove this sentence and keep the description

factual.

*P.10, L.290:* *RACMO05? (= RACMO5?)*
**Answer:** Yes, it was a typo and has been corrected everywhere it appeared in the manuscript.

*P.10, L.295:* *Try to avoid generalization. Consider changing "all scales" to "all investigated scales".*
**Answer:** We agree and have added "investigated" in that sentence.

*P.10, L.310:* *Reference figures for iGCMs and RACMO.*
**Answer:** The figures are now referenced accordingly, taking into account the changes in Figs. 1 and 2 (see previous comments).

*P.11, L.318:* *"...an important process, process that..." → "...an important process, i.e. a process that..." or "...an important process, that..."*
**Answer:** We have opted for "...an important process, i.e. a process that...", thank you.

*P.11, L.339:* *instead of "as discussed below" reference Section.*
**Answer:** It is in the same section, so perhaps this phrasing is unnecessary. We have removed it.

*P.11, L.342:* *You mention that the correlations of the cores with an inhomogeneous distribution over the continent are probably not representing the continental or even regional correlation. Then, you mention that the model shows mainly positive correlations where the ice cores are located. Did you consider calculating the correlation of the model only for grid cells for which ice core data exists? How does the comparison compare in this case? Could this result be used to underline the disagreement?*
**Answer:** We have responded to this question in the 2nd comment of this review and therefore refer the reviewer to the earlier response.

*P.11, L.344:* *remove "outlined"*
**Answer:** Removed.

*P.12, L.356:* *reference the 108x108 km results as "not shown"*
**Answer:** We have added "not shown" after "[...] remain at 0.09 and 0.13 respectively".

*P.12, L.366:* *"Results are shown in Fig. 9." No need for a whole sentence, just give reference as (Fig. 9).*
**Answer:** This has been changed to "(only three grid points contain at least five ice cores at the 648 x 648 km grid resolution, Fig. 9)".

*P.12, L.368:* *"all spatial scales": Avoid generalization, i.e. "all investigated spatial scales"*
**Answer:** Modified.

*P.12, L.375-376:* *Please clarify this sentence.*
**Answer:** We have reworded it to: "Examining the ice core-based results, we note that increasing the number of ice core records that are initially averaged results in a visible increase of the SMB-SAT annual correlation, but the trend is weak."

*P.21, Fig.7:* *significantive → significant*
**Answer:** Changed.

***P.21, Fig.7:*** *Maybe change "Large dots indicate..." to "Large gray encircled dots indicate..."*
**Answer:** This has been changed as suggested to "Large gray encircled dots indicate [...]".

***P.22, Fig.8:*** *Why don't you show the standard deviation and percentage of p<0.1 on these graphs? What does the size of the dots display? Provide information in the caption and a legend for sizes.*
**Answer:** The standard deviation and percentage of p< $p_{FDR}$ have been added (see Fig. 8 of the manuscript). The size of the dots displays the number of records averaged over, and we now provide size information in the caption and legend as suggested.

***Suppl., P.8, Fig S9:*** *specify on which level wind speed and direction is shown.*
**Answer:** We have changed the caption to "Mean 10 m wind speed[...]".

[Figure]

Figure 1: SMB-SAT annual correlation as a function of grid spacing for the aggregated records versus mean of the individual annual correlations for the ice cores (dark and light blues, respectively) and RACMO27 simulations (in dark and light green, respectively). Only grid points with at least (top) one and (bottom) five ice cores are kept. Annual correlations are over the 1958–2010 AD time interval.

[Figure]

Figure 2: Annual correlation between SMB and SAT over (left) the 1961–2000 and (right) the 1979–2000 AD time intervals for each of the 3 iCESM1 simulations.

[Figure]

Figure 3: Annual correlation between ice core $\delta^{18}$O records and the weather station SAT records used in the Nicolas and Bromwich (2014) SAT reconstruction for 3 select locations where the weather stations and the ice cores are in close proximity, over the 1960–2010 AD time interval. Correlation values are given on the figure itself. The filled black dots are the locations of the rest of the $\delta^{18}$O ice core records and the empty black dots are the locations of the rest of the weather stations in the Nicolas and Bromwich (2014) SAT reconstruction.

[Figure]

Figure 4: Zoomed view of the AP region: correlation between MSWD and (top) SAT and (bottom) SMB for the AIS calculated from RACMO27 simulations over the 1979–2016 AD time interval. Magenta lines outline the areas with a weak SMB-SAT correlation from Fig. 2 in the main manuscript.

[Figure]

[Figure]

Figure 5: (left) mean SMB in mm w.e. yr$^{-1}$ vs (right) mean snowfall in mm w.e. yr$^{-1}$ in the RACMO27 simulations over the 1979–2016 AD time interval.

[Figure]

[Figure]

Figure 6: Sketch of the averaging of the model and ice core data for Fig. 10 of the manuscript. Example for averaging over 8 x 8 grid cells : the small cells represent the original RACMO27 grid cells, the blue dots the ice core locations, and the red coloring the RACMO27 grid cells used in the average. Sketch (a) is how we calculated the averages shown in Fig. 7 (top panel) of this document: we compare the ice core average correlations (both average correlation and correlation of the averages) to the average RACMO27 correlation over the whole larger (red) grid cell (both average correlation and correlation of the averages as well). Sketch (b) is what we show in Fig. 7 (bottom panel) of this document for comparison: we compare the ice core average correlations to the average RACMO27 correlations using only the original RACMO27 grid cells that contain an ice core instead of the whole larger grid cell. Sketch b is what we use in Fig. 10 of the manuscript to compare ice core and model values locally.

[Figure]

Figure 7: SMB-SAT annual correlation as a function of grid spacing for the aggregated records versus mean of the individual annual correlations for the ice cores (dark and light blues, respectively) and RACMO27 simulations (in dark and light green, respectively). For n the scaling value, RACMO27 correlations use (top panel) the whole n by n grid points, (bottom) only 
[revised manuscript text omitted]

- Figures S1, S2  $\delta^{18}$ and S3 show the annual correlation between SMB and SAT, and SMB and  $\delta^{18}$O for each iGCM model at its original spatial resolution,  and the average over all four iGCMs interpolated onto the RACMO27 grid, over the 1961–2000 AD time intervals  (Fig. S1 and S2) and the 1871–2000 AD time interval (Fig. S3).

- Figure S4 (a,b) shows the spatial distribution of the ice cores used for the SMB-SAT and SMB-$\delta^{18}$O correlations and (c) shows the spatial distribution of the weather stations used in the Nicolas and Bromwich (2014) SAT reconstruction.

- Figure S5 shows the 5-yearly SMB-SAT correlation averaged over the four iGCMs.

- Figure S6 shows the correlation of SMB and SAT at the seasonal timescale.

- Figure S7 shows the correlation of SMB and SAT at the monthly timescale.

- Figure S8 shows the RACMO2.3 (Van Wessem et al., 2018) MSWD calculated as described in the main manuscript, averaged over the entire period (1979–2016 AD).

- Figure S9 shows a zoomed-in view of the Antarctic Peninsula region of the MSWD-SAT and MWSD-SMB annual correlations.

- Figure S10 shows the RACMO2.3 (Van Wessem et al., 2018) 10-m wind strength and direction averaged over the entire period (1979–2016 AD).

- Figure S11 shows the RACMO2.3 (Van Wessem et al., 2018) sublimation and sublimation from snowdrift averaged over the entire period (1979–2016 AD).

- Section 2 provides an evaluation of the iGCMs, with the associated Figs. S12, S13 and S14.

 ECHAM5-MPI-OM

**2   Model evaluation**

Overall, the mean SMB over the Antarctic Ice Sheet (AIS) simulated by the iGCMs is in good agreement with the mean SMB simulated by RACMO27 over the 1979–2000 AD time interval ($R^2$ = 0.49-0.65, maximum bias[1] of $\sim$37 mm w.e. yr$^{-1}$, Fig. S12). We choose to compare the iGCM simulations to RACMO2.3 since the latter includes physics specific to polar regions. As such, RACMO2.3 (and RACMO27 specifically in this case to have a continent-wide field of SMB) is highly appropriate for an Antarctic SMB evaluation. Both iGCMs and RACMO27 show a pattern of high SMB ($>$ 500 mm w.e. yr$^{-1}$) along the coast and low SMB ($<$ 30 mm w.e. yr$^{-1}$) in the interior. We note that ECHAM5-wiso and ECHAM5-MPI/OM show a mean SMB bias of $\sim$30-40 mm w.e. yr$^{-1}$ above that of RACMO27 ($R^2$ = 0.63 and 0.49 respectively), while iCESM1 shows a bias of $\sim$ -11 mm w.e. yr$^{-1}$ with respect to RACMO27 ($R^2$ = 0.65) and the iHadCM3 SMB mean value is very close to that of RACMO27 (bias of $\sim$-2, $R^2$ = 0.58). The biggest differences between the iGCMs and RACMO27 in general occur along the coasts, in particular the West Antarctic coast and the Antarctica Peninsula, reaching up to 500 mm w.e. yr$^{-1}$. In addition, ECHAM5-MPI/OM, which has the coarsest spatial resolution, also shows a positive bias in the interior of $\sim$200 mm w.e. yr$^{-1}$. This agrees with previous studies that have shown that the lower resolution of GCMs induces under-estimations of SMB in coastal areas and in general over-estimations of SMB in the interior (Palerme et al., 2017; Krinner et al., 2007; Agosta et al., 2015). As the iGCM with the lowest spatial resolution of this study, ECHAM5-MPI/OM shows the largest biases (coastal and in the interior). Despite these (known) biases, the simulated SMB by each of the four iGCMs shows minor differences locally with the RACMO27 SMB, expected due to the differing physics involved in each model and range of spatial resolutions.

To evaluate the simulated SAT by the iGCMs, we compare the SAT simulated by the iGCMs for the AIS to that simulated by RACMO27 as well (Fig. S13). The four iGCMs produce the same spatial pattern as RACMO27, that is higher SATs at the coast and the lowest SATs in the interior ($R^2$ = 0.91-0.97). The average SAT over the continent is within 1°C of the RACMO27 average, except for iCESM1 which shows a mean bias of $\sim$5°C. However, if we look at the difference between the iGCM SAT and the RACMO27 SAT (right column of Fig. S13), we see that the differences are spatially variable. iCESM1 underestimates SAT for most of the ice sheet surface while the opposite is true for ECHAM5-wiso. ECHAM5-MPI/OM and iHadCM3 show SAT overestimation around the coast and along the Trans-Antarctic Mountains and SAT underestimation over the west and east Antarctic interiors. Differences remain on the order of $\pm$3°C, with only a few regions, in particular for iCEMS1 and ECHAM5-MPI/OM, where differences reach up to 12°C. Overall, despite the discrepancies, all four iGCMs produce a realistic simulated SAT with lower temperatures in the center of the ice sheet, warmer temperatures over the coastal regions, as expected, and a similar gradient between the interior and the coast for all models, which gives us confidence in using these four iGCMs in our study.

We then evaluate the sea-level pressure (SLP) simulated by the iGCMs by comparing them to the simulated SLP by ERA-Interim (Dee et al., 2011) which allows us to examine the SLP further north of the Antarctic continent compared to RACMO2.3 (Fig. S14). We calculate the average SLP over the ocean below -40°S of latitude. Over that area, the average
* * *
[1]We define bias as the spatial mean value of the iGCM variable over the period considered minus the spatial mean value of the reference model chosen, RACMO27 in this instance. The spatial mean value of each iGCM variable is calculated from the iGCM values after interpolation onto the reference model grid, RACMO27 in this case. This is true for the SAT and SLP evaluation as well, see below.

SLP values for the iGCMs are very close to that of ERA-Interim ($R^2$ = 0.89-0.99), with up to 2 hPa difference except for

55 iHadCM3 which shows a mean SLP bias of ~5.2 hPa lower than ERA-Interim. The same spatial pattern of SLP with the three climatological low-pressure systems over the Southern Ocean is found in all models. The absolute minimum is found in the Amundsen Sea Embayment for iCESM1 and iHadCM3 as in ERA-Interim. For ECHAM5-wiso and ECHAM5-MPI/OM, the absolute minimum is found offshore Princess Elisabeth Land and Enderby Land, respectively, although a weaker minimum is located in the Amundsen Sea Embayment. Looking at the difference between each iGCM simulation and that of ERA-Interim

60 (right column, Fig. S14), we see that differences never exceed ~7 hPa locally, except for iHadCM3 where differences reach up to ~8 hPa. Despite each iGCM's different local bias, all four iGCMs reproduce the same average and spatial pattern of SLP over the ocean as ERA-Interim.

| spacing | number of ice cores |
|---------|---------------------|
| 4 | 6 |
| 4 | 5 |
| 8 | 5 |
| 8 | 10 |
| 8 | 7 |
| 12 | 18 |
| 12 | 7 |
| 16 | 5 |
| 16 | 11 |
| 16 | 17 |
| 20 | 5 |
| 20 | 26 |
| 24 | 8 |
| 24 | 21 |
| 24 | 9 |

**Table S1.** Number of ice cores aggregated as a function of RACMO27 grid spacing for Fig. 10 of the main manuscript, with only grid points that aggregate five or more ice cores retained to have sufficient averaging of the records. E.g. at the grid spacing of 4, two grid points are retained. These are then both included in the gray range of correlation values shown in Fig. 10.

[revised manuscript text omitted]

---

## Editor Decision (ED1)

Technical Corrections:

Dear authors,

Congratulations on your accepted publication. I would like to propose the following technical edits to your paper.

1. The manuscript itself is very well written and the updated version in particular is very clear. However, in reading through it I was a little unclear what the ultimate aim of the project here is. Are you seeking to derive an empirical relationship between SMB and air temperature? And should this be used to interpret ice core records? Or is the aim rather to make projections of SMB change based on temperature projections? I think it would strengthen your paper to add one or two sentences to clarify this at the very start of your paper before you go in to describing the different contributions to the total mass budget.

   At the end of the paper you write:

*"This implies that we must correct for the local processes present in each ice core record so that their spatial representativity is closer to that of the models, or models must increase their spatial resolution to better resolve wind effects, in order to improve our confidence in using SMB as a direct proxy for SAT over the entire AIS."*

So perhaps something anticipating this conclusion that directly links surface air temperature with SMB at the very start would helps

In addition I noticed a few typos, missing references and I suggest a couple of english corrections:

Line 31: surpassing or outpacing but not "outpassing"

Lines 46, 196: there is a reference missing

Line 316: "such **as** the"

Several uses of the word representativity. It's not wrong as such but it's used more to indicate polticial representation in english. The word representativeness is more common for scientific publications.

I may not have caught them all so please do proof read a final time.

Congratulations on a very interesting and well-written paper!